



# The High lAtitude sNowfall Detection and Estimation aLgorithm
# for ATMS (HANDEL-ATMS): a new algorithm for the snowfall
# retrieval at high latitudes
Andrea Camplani[1], Daniele Casella[1], Paolo Sanò[1], Giulia Panegrossi[1]
[1]National Research Council of Italy, Institute of Atmospheric Sciences and Climate (CNR-ISAC), Via del Fosso
del Cavaliere 100, 00133 Rome, Italy
*Correspondence to*: Andrea Camplani (Andrea.Camplani@artov.isac.cnr.it)
**Abstract**. Snowfall detection and quantification are challenging tasks in the Earth system science field. Ground-
based instruments have limited spatial coverage and are scarce or absent at high latitudes. Therefore, the
development of satellite-based snowfall retrieval methods is necessary for the global monitoring of snowfall.
Passive Microwave (PMW) sensors can be exploited for snowfall quantification purposes because their
measurements in the high-frequency channels (> 80 GHz) respond to snowfall microphysics. However, the highly
non-linear PMW multichannel response to snowfall, the weakness of snowfall signature and the contamination by
the background surface emission/scattering signal make snowfall retrieval very difficult. This phenomenon is
particularly evident at high latitudes, where light snowfall events in extremely cold and dry environmental
conditions are predominant. ML techniques have been demonstrated to be very suitable to handle the complex
PMW multichannel relationship to snowfall. Operational microwave sounders on near-polar orbit satellites such
as the Advanced Technology Microwave Sounder (ATMS), and the European MetOp-SG Microwave Sounder in
the future, offer a very good coverage at high latitudes. Moreover, their wide range of channel frequencies (from
23 GHz to 190 GHz), allows for the radiometric characterization of the surface at the time of the overpass along
with the exploitation of the high-frequency channels for snowfall retrieval. The paper describes the High lAtitude
sNow Detection and Estimation aLgorithm for ATMS (HANDEL-ATMS), a new machine learning-based
snowfall retrieval algorithm developed specifically for high latitude environmental conditions and based on the
ATMS observations.
HANDEL-ATMS is based on the use of an observational dataset in the training phase, where each ATMS
multichannel observation is associated with coincident (in time and space) CloudSat Cloud Profiling Radar (CPR)
vertical snow profile and surface snowfall rate. The main novelty of the approach is the radiometric
characterization of the background surface (including snow covered land and sea ice) at the time of the overpass
to derive multi-channel surface emissivities and clear-sky contribution to be used in the snowfall retrieval process.
The snowfall retrieval is based on four different artificial neural networks for snow water path (SWP) and surface
snowfall rate (SSR) detection and retrieval HANDEL-ATMS shows very good detection capabilities - POD =
0.83, FAR = 0.18, and HSS = 0.68 for the SSR detection module. Estimation error statistics show a good
agreement with CPR snowfall products for SSR > $10^{-2}$ mm h$^{-1}$ (RMSE 0.08 mm h$^{-1}$, bias=0,02 mm h$^{-1}$). The
analysis of the results for an independent CPR dataset and of selected snowfall events evidence the unique
capability of HANDEL-ATMS to detect and estimate SWP and SSR also in presence of extreme cold and dry
environmental conditions typical of high latitudes.
# 1 Introduction
Snowfall retrieval is one important topic in the atmospheric science field. On a global scale, snowfall represents
only 5 % of the total global precipitation but it is predominant above 60-70 ° N/S (see *Levizzani et al, 2011*). In
recent years, several studies have highlighted the strong influence of global warming on snowfall distribution and
regimes, especially at high latitudes (see *Liu et al, 2009*, *Liu et al, 2012*, *Bintanja & Selten, 2014*, *Vihma et al,
2015*). However, global snowfall quantification is a challenging topic in weather sciences. Ground-based
instruments such as raingauges or snowgauges provide only punctual measurements which can not fully capture
the spatial variability of precipitation phenomena; moreover, the variability of snowflake shape and density has a
strong influence on their fall speed and trajectories and therefore gauge-based measurements of falling snow result
to be less accurate than for rain (see *Skofronick-Jackson et al, 2015*). Weather radars can provide areal
measurements of precipitation - the rate estimation is based on the conversion of the measured backscattered
radiation to precipitating hydrometeors content - but such operation presents some technical limitations (see *Kidd



*& Huffman, 2011*). Finally, most of the regions where snowfall is predominant - such as Greenland, Siberia,
Canada, and Antarctica - are uninhabited or otherwise sparsely populated areas where weather observation
networks are very scarce or totally absent. Therefore, the development of satellite-based methods for snowfall
retrieval is necessary for global monitoring of snowfall. Passive Microwave (PMW) sensors on board polar
orbiting satellites can be exploited for snowfall detection purposes because the microwave (MW) signal is directly
responsive to the spatial distribution and microphysics properties of precipitation-sized hydrometeors in the
clouds; at the same time, the use of PMW sensors guarantees a high spatial coverage and high temporal resolution
(see *Kidd & Huffman, 2011*).
PMW snowfall detection and quantification is typically based on the ability to interpret the snowfall scattering
signature in the high frequency channels (> 90 GHz), which respond more effectively to ice microphysics and are
less prone to surface effects than low frequency channels, and to distinguish it from the clear-sky (surface and
atmosphere) contribution (e.g., *Panegrossi et al., 2017*). However, several factors make the PMW snowfall signal
ambiguous and the relationship between multichannel measurements and surface snowfall intensity highly non-
linear, especially in extremely cold/dry environmental conditions (see *Panegrossi et al*, 2022). The snowfall
scattering signal is relatively weak and is highly dependent on the complex microphysical properties of snowflakes
(*Kim et al, 2008*, *Kulie et al, 2010*, *Kongoli et al, 2015*), it is often masked by supercooled liquid water emission
signal, and can be contaminated by the extremely variable background surface emissivity (*Liu and Seo, 2013,*
*Takbiri et al., 2019, Rahimi et al, 2017*), especially in cold and dry conditions typical of the high latitude regions
(*Camplani et al., 2021*). In this context, the availability of the last generation microwave radiometers - such as the
conically-scanning radiometer GPM Microwave Imager (GMI) and the cross-track scanning radiometer Advanced
Technology Microwave Sensor (ATMS) - whose channels cover a wide range of frequencies - offers new
possibilities for global snowfall monitoring. The multi-channel PMW observations can be used for both a
radiometric characterization of the background surface - using the low-frequency channels (< 90 GHz) - and for
the detection and the estimation of the snowfall using the high-frequency channels (> 90 GHz) (see *Panegrossi et*
*al, 2022*).
The PMW capability to characterize physically and radiometrically the background surface varies from sea to
land, especially for the identification of cold/frozen surfaces. For what concerns the ocean, sea ice detection using
PMW observations has been a well-documented topic in the remote sensing science field since the 70s. This is
due to the strong contrast between sea ice (≈ 0.9) and open water (≈ 0.5) emissivity values at the MW low-
frequency range (~19 GHz) (see *Comiso, 1983*). Other studies highlighted the ability to discriminate between
different types of ice using a set of low-frequency window channels, because the differences between the
emissivities of the different types of sea ice increase with increasing frequency; in particular, at higher frequencies
(30-50 GHz) the contrast between the emissivity of "new" ice and "old" ice increases, with a decrease of the
emissivity at higher frequencies for "older" sea ice (see *Comiso, 1983, Ulaby et al, 2014*). Moreover, it has been
observed that the simultaneous presence of open water and sea ice causes a decrease in the low-frequency channel
emissivity; the observed emissivity can be considered as a linear combination of the emissivity spectra of sea ice
and open water (see *Ulaby et al, 2014*). For what concerns continental areas, the detection of snow-covered land
surfaces using MW results to be more difficult. In dry conditions, a snowpack acts as a volume scatterer; the
scattering effect is dependent on the grain size and shape and on the depth of the snowpack (see *Clifford, 2010*).
However, the presence of liquid water can mask the scattering signature (see *Mätzler & Hüppi, 1989*). At the same
time, large areas of Greenland and Antarctica could appear as "scatter-free", although these areas throughout the
year are covered by dry snowpacks. Finally, some snow-free areas, such as rocky mountains and cold deserts,
present a scattering signature very similar to that of the snowpack (see *Grody & Basist, 1996*). Therefore, the
detection of snow-covered areas is very complex. A set of several tests, each of which identifies snowpacks
characterized by different physical and radiometric characteristics, may be used.
This paper describes the development of a machine learning-based algorithm for snowfall retrieval (the High
lAtitude sNowfall Detection and Estimation aLgorithm for ATMS, HANDEL-ATMS), exploiting ATMS
radiometer multi-channel measurements and using the CloudSat Cloud Profiling Radar (CPR) snowfall products
as reference. The algorithm has been developed focusing on the typical conditions of high latitude regions - low
humidity, low temperature, presence of snowpack on land or sea ice over ocean, and light snowfall intensity.
The main novelty of the approach is the exploitation of the ATMS wide range of channels (from 22 GHz to 183
GHz) to obìtain the radiometric characterization of the background surface at the time of the overpass. The derived



surface emissivities are used to infer the clear-sky contribution to the measured TBs in the high-frequency
channels in the snowfall retrieval process. Moreover, the algorithm is based on the exploitation of an observational
dataset where each ATMS multichannel observation is associated with coincident (in time and space) CloudSat
CPR vertical snow profile and surface snowfall rate (hereafter ATMS-CPR coincidence dataset).
Several snowfall retrieval algorithms for cross-track scanning radiometers have evolved in the last 20 years
starting from the Advanced Microwave Sounder Unit-B (AMSU-B) (*Kongoli et al, 2003, Skofronick-Jackson et*
*al, 2004, Noh et al., 2009, Liu and Seo 2013*), and Microwave Humidity Sounder (MHS) (see *Liu & Seo*, 2013,
*Edel et al*, 2020), and evolving to ATMS (*Kongoli et al*, 2015, *Meng et al*, 2017, *Kongoli et al*, 2018, *You et al*,
2022, *Sanò et al*, 2022). Some of them are based on radiative transfer simulations of observed snowfall events
(*Kongoli et al*, 2003, *Skofronick-Jackson et al*, 2004, *Kim et al*, 2008), or on in-situ data (see *Kongoli et al, 2015,*
*Meng et al, 2017, Kongoli et al, 2018*), others on CPR observations (*Edel et al, 2020, You et al, 2022, Sanò et al,*
*2022*), or a combination of them (*Noh et al, 2009, Liu & Seo, 2013*).  In the last five years, there has been an
increasing use of machine learning (ML) approaches trained on CPR-based coincidence datasets. These
approaches have proven to be very effective for snowfall retrieval. On one side, ML techniques are suitable to
handle the complex, nonlinear PMW multichannel response to snowfall (e.g., *Rysman et al., 2018, Edel et al.,*
*2020, Sanò et al. 2022*). On the other side, the use of CPR-based datasets overcomes some of the limitations
deriving from the assumptions to be made in cloud-radiation model simulations (e. g., the microphysics scheme,
the emissivity of the background surface, scattering properties of ice hydrometeors), which are particularly
problematic for snowfall estimation. However, some limitations of the radar product used as reference and issues
related to the spatial and temporal matching between the CPR and the PMW radiometer measurements introduces
some uncertainty.
For what concerns ATMS, the ML-based Snow retrievaL ALgorithm fOr gpM–Cross Track (SLALOM-CT)
(*Sanò et al., 2022*) has been developed within the EUMETSAT Satellite Application Facility for Hydrology (H
SAF) in preparation for the launch of the EPS-SG Microwave Sounder (MWS). Similarly to HANDEL-ATMS, it
is trained on a ATMS-CPR coincidence dataset. SLALOM-CT is the evolution for cross-track scanning
radiometers of the Snow retrievaL ALgorithm fOr GMI (SLALOM) (*Rysman et al, 2018, Rysman et al, 2019*)
which was the first ML algorithm for snowfall detection and retrieval for GMI trained and tested on GMI-CPR
coincident observations made available in the NASA GPM-CloudSat coincidence dataset (*Turk et al., 2021a*).
One of the novelties in the SLALOM (SLALOM-CT) approach is the use of the GMI (ATMS) low-frequency
channels to better constrain the snowfall retrieval to the characteristics of the surface at the time of the overpass
(*Turk et al., 2021b*). SLALOM-CT is based on a modular scheme, i.e., four separate modules are used for snowfall
detection, supercooled water layer detection, snow water path (SWP) and surface snowfall rate (SSR) estimate.
The predictor set is composed of the ATMS TBs and some environmental variables ($T_{2m}$, TPW, and principal
components derived from temperature and humidity profiles).
However, none of the algorithms mentioned here were trained specifically for the extreme conditions typical of
high latitudes. The present work has the aim to develop an algorithm for snowfall detection and estimation by
exploiting the large frequency range typical of the last generation radiometers and to obtain a radiometric
characterization of the background surface at the time of the satellite overpass in order to highlight the complex
relationship between upwelling radiation and snowfall signature, which makes the detection very difficult in the
typical conditions of the high latitudes.
This article is organized as follows: Section 2 provides background information on ATMS and CPR, on the
methodology used to build the coincidence dataset and on the machine learning approaches used to develop the
algorithm. In Section 3 the algorithm structure is described. In Section 4 the overall performance scores are
reported and analyzed; a case study is analyzed and a comparison with SLALOM-CT is reported. Section 5 is
dedicated to the summary of the main results and to the conclusions.
**2. Instruments and methods**
**2.1 Advanced Technology Microwave Sounder (ATMS)**
ATMS is a total power cross-track scanning radiometer within 52.7° off the nadir direction. It has a total of 22
channels with the first 16 channels primarily used for temperature sounding from the surface to about 1 hPa (45
km) and the remaining channels used for water vapor sounding in the troposphere from the surface to about 200
hPa (10 km). There are two receiving antennas: one serving channels 1–15 below 60 GHz, and the other for
channels above 60 GHz. The beamwidth changes with frequency and is 5.2° for channels 1–2 (23.8–31.4 GHz),



2.2° for channels 3–16 (50.3–57.29 and 88.2 GHz), and 1.1° for channels 17–22 (165.5–183.3 GHz). The
corresponding nadir resolutions are 74.78, 31.64, and 15.82 km, respectively. The outmost field of view (FOV)
sizes are 323.1 km × 141.8 km (cross-track × along-track), 136.7 km × 60.0 km, and 68.4 km × 30.0 km,
respectively. The ATMS can be considered the evolution of the three main previous cross-track scanning
radiometers: Advanced Microwave Sounding Unit-A (AMSU-A), Advanced Microwave Sounding Unit-B
(AMSU-B), and Microwave Humidity Sounder (MHS). Seventeen ATMS channels (channels 1–3, 5–15, 17, 20,
and 22) have the same frequencies as its two predecessors AMSU, two ATMS channels (channels 16 and 18) have
slightly different frequencies from AMSU channels, and three new channels (channels 4, 19 and 21) have been
added to ATMS (see *Weng et al, 2012*). ATMS is currently carried by three near-polar orbiting satellites, Suomi
National Polar-orbiting Partnership (SNPP) NOAA-20, and NOAA-21 providing global coverage including polar
regions. Moreover, each satellite revisiting time is equal to 12 hours at the equator, but drops to 100 minutes over
the polar regions, ensuring a very high temporal resolution for the research area of interest in this work. Moreover,
the operational nature of the mission guarantees observations for the next decades. It is worth noticing that the
polarization of ATMS channels is not defined as vertical or horizontal, but as "Quasi-Vertical" or "Quasi-
Horizontal". The "Quasi" prefix is used to indicate that ATMS (and any other cross-track scanner) measures
vertical or horizontal polarization only when looking at nadir and a mixture of V and H polarization for off-nadir
scan angles.

**2.2 Cloud Profiling Radar (CPR)**

The CPR is a 94 GHz nadir-looking radar onboard CloudSat. CloudSat was launched on April 28, 2006; the W-
band (94 GHz) Cloud Profiling Radar (CPR) operations began on June 2, 2006. CPR has been acquiring the first-
ever continuous global time series of vertical cloud structures and vertical profiles of cloud liquid and ice water
content with a 485-m vertical resolution and a 1.4-km antenna 3-dB footprint. The reference CloudSat snowfall
product is the 2C-Snow-Profile (2CSP) product (Version 5 is used in this work). It provides estimates of snowfall
characteristics for each observed profile. In particular, it provides an estimate of the Snow Water Path (SWP), i.
e., the total snow water content integrated over the atmospheric column, and of the Surface Snowfall Rate (SSR)
(see *Stephens et al, 2008*). SWP is estimated also when there is no snowfall at the ground level; therefore, the
presence of SWP is not always linked to the presence of SSR, especially in warmer near-surface conditions (see
*Wood & L'Ecuyer, 2018*). 2CSP has several limitations, such as the contamination of the signal in the lowest 1000
- 1500 m of the profile due to ground-clutter, the underestimation of the heavy snowfall events, due to attenuation
of the radar signal in these conditions, and the limited temporal sampling (although it is higher in the polar
regions), and the day-only operation mode since 2011, which limits its use during the winter seasons (see *Milani*
*and Wood, 2021, Panegrossi et al, 2022*). However, 2CSP has been demonstrated to be more accurate than GPM
Dual-frequency Precipitation Radar (DPR) snowfall products (see *Casella et al, 2017*) and in good agreement
with estimates obtained by ground-based radars (e.g., *Mroz et al, 2021*), although it is affected by underestimation
for medium-heavy snowfall events. Moreover, the polar orbit and the W-band high sensitivity make CPR suitable
for snowfall monitoring at higher latitudes (as demonstrated in several studies, *Kulie et al, 2016, Milani et al,*
*2018*) typically characterized by light/moderate intensity (*Beranghi et al, 2016*).

**2.3 ATMS-CPR Coincidence Dataset**

The present study is based on a coincidence dataset between CPR and ATMS observations between January 2014
and August 2016. The same dataset has been used for the development of SLALOM-CT (*Sanò et al*, 2022). Each
coincidence comes from observations from CloudSat CPR and ATMS - onboard SNPP - within a maximum 15-
minute time window. Moreover, the elements in the dataset have been selected by removing all corrupted data
and by applying an additional filter based on the minimum distance between CPR and ATMS IFOV center which
(22 km).  The zonal distribution of the coincidences is due to the orbital geometry of CloudSat and SNPP, which
are both sun-synchronous with a relatively small difference in the satellite height (i. e., about 689 km and 833 km
for CloudSat and SNPP respectively). Therefore, the coincidence dataset is built from longer orbit fragments
(often semi-orbits) and by a very large number of elements near the poles. There is an asymmetry in the CPR
sampling between the Northern and the Southern hemisphere that can be observed in the dataset due to the CPR
daytime-only mode operation since 2011, which influences mostly the acquisitions in the Southern Polar region.
The database has been built considering the horizontal resolution of the high-frequency channels of ATMS. The
CPR snowfall product used as reference is the 2CSP (v.5). Some model-derived variables have been added to the
dataset to be used as ancillary variables. Both 2D and 3D environmental variables have been obtained from the



European Center Medium Weather Forecast (ECMWF). In particular, they are obtained from the CPR ECMWF-
AUX product where the set of ancillary ECMWF atmospheric state variable data is associated with each CloudSat
CPR bin (the product is described by *Partain, 2022*). Moreover, a cloud-cover fraction index, which indicates the
fraction of CPR observations where cloud is observed on the total CPR observations within each ATMS pixel, is
added to the dataset.
Information about the presence of supercooled water is added in the coincidence dataset to be used towards the
correct interpretation of the snowfall signal in presence of supercooled water layers. The supercooled water
information has been extracted from the DARDAR product (see DARDAR). DARDAR, which stands for
raDAR+liDAR, combines CPR radar and Cloud-Aerosol Lidar with Orthogonal Polarization (CALIOP) lidar
observations, onboard Cloud-Aerosol Lidar and Infrared Pathfinder Satellite Observations (CALIPSO) satellite,
and estimates both the cloud water phase and the ice water content and ice particle effective radius (see *Battaglia*
*& Delanoë, 2013*, *Ceccaldi et al, 2013*). In particular, the coincidence dataset includes an index indicating the
presence of supercooled liquid water within each ATMS pixel, calculated as the fraction of DARDAR
observations where supercooled water within and on the top of the cloud is observed to the total DARDAR
observations within each pixel.
The association of ATMS TBs and CPR products has been done by averaging the CPR snow products with a
Gaussian function approximating the ATMS high-frequency antenna pattern (varying with the scan angle). It is
worth noting, however, that the ATMS IFOV is under-sampled by the narrow swath of the CPR (see *Sanò et al.,*
*2022* for details).
**2.4 Machine Learning approaches**
The algorithm is based on different machine-learning (ML) techniques. These techniques are widely applied in
Earth observation because of their ability to approximate, to an arbitrary degree of accuracy, complex nonlinear,
and imperfectly known functions. A fundamental characteristic of these techniques is that the training process
eliminates the need for a well-defined physical or numerical model that describes the relationships between the
input values and output results, allowing the identification of these relationships during the learning phase (see
*Sanò et al*, 2022). Moreover, clustering techniques have been used to characterize from a radiometric point of
view the background surface. In particular, an unsupervised clustering technique has been used to identify
emissivity clusters with small internal variability, and a supervised clustering technique has been used to identify
an emissivity spectrum based on other parameters.
**2.4.1 Artificial Neural Networks**
An Artificial Neural Network (ANN) is an information-processing system inspired by the functioning of
biological neural networks. It is composed of neurons, i. e., elements where the information is processed using an
activation function, and the connecting links between the neurons, where a weight multiplies the deriving from
the upstream signal. In particular, the HANDEL-ATMS snowfall detection and estimation modules have been
developed using feedforward multilayer neural network architectures, i. e., a neural network architecture where
the neurons are arranged in layers; each neuron belonging to a layer receives, as input to its transfer function, a
weighted sum of the outputs of the previous layer. This architecture, which is defined by the number of layers,
the number of neurons for each layer, and the transfer function of each neuron, has to be designed beforehand.
The weights of connection links and the bias values for each layer are estimated with a training process, based on
the Levenberg–Marquardt algorithm (see *Sanò et al, 2015*).
**2.4.2 Self Organizing Maps**
The unsupervised clustering method used for the background surface classification is the Self Organizing Map
(SOM) method (see *Kohonen, 2012*). The characteristic of this method is to assume a topological structure among
the cluster units: the maps can be represented as a neuron network where each neuron represents a cluster. Similar
to the k-means clustering method, the neuron is associated with an input vector by minimizing a distance
measurement; however, not only the weight vector of the winning neuron is updated, but also the weight vectors
of all the neurons which are considered topologically close (see *Faussett, 2006*). Therefore, classes that are close
to each other from a topological point of view can be considered similar also from a physical and radiometric
point of view (see *Munchak et al, 2020*). SOMs have been already used to make a classification of the background
surface by creating clusters based on emissivity values (see *Prigent et al, 2001*, *Cordisco et al, 2006*, *Prigent et*
*al, 2008*, *Munchak et al, 2020*).





### 2.4.3 Linear Discriminant Analysis

Several supervised clustering methods have been tested in this study, such as the linear discriminant analysis, the quadratic discriminant analysis, the classification tree, and the nearest neighbor method. The final choice came down to linear discriminant analysis (LDA) because this method guarantees satisfactory accuracy in the results with a difference between the performances of the training and the test phase which is not too significant, and a computational effort which is not too high. Discriminant Analyses are classification methods based on the assumption that each observation is a realization of a normal distribution - if there is a single predictor - or a multivariate normal distribution - if it is based on more than one predictor. In particular, LDA assumes that clusters have a common covariance analysis; therefore, the decision boundary between the clusters results to be linear relationships (see *Hastie et al, 2009*).

### 3 Algorithm description

The configuration of the HANDEL-ATMS is summarized in the Flowchart in Figure 1. The process begins with the classification of the background surface using the PMW Empirical cold Surface Classification Algorithm (PESCA, *Camplani et al, 2021*); then, the surface emissivity spectra are derived through refinement process based on LDA and these are used to estimate clear-sky simulated TB ($TB_{sim}$) using the ECMWF-AUX atmospheric temperature and water vapor profiles. Then, the differences between the $TB_{sim}$ and the ATMS observed TB ($TB_{obs}$) are evaluated ($\Delta TB_{obs-sim} = TB_{obs} - TB_{sim}$). Four ANNs are then applied to a predictor set consisting of ATMS $TB_{obs}$, $\Delta TB_{obs-sim}$, a surface classification flag, and other environmental and ancillary parameters. Finally, the pixels classified with the presence of snowfall by the detection module, are used in the estimation modules while for no-snowfall flagged pixels the snowfall rate value is set to 0 mm/h. In the following sections the main blocks of the algorithm are described in detail.

### 3.1 Surface Classification and emissivity spectra estimation

#### 3.1.1 PESCA Design and Performances

The classification and radiometric characterization of the background surface at the time of the satellite overpass is based on PESCA exploiting ATMS low-frequency channels (*Camplani et al., 2021*). The algorithm discriminates between frozen and unfrozen surfaces (sea ice and open water, snow-covered land and snow-free land), and identifies 10 surface classes (4 over ocean, 5 over land, 1 for coast). The algorithm has been tuned against the NASA AutoSnow product (see *Romanov, 2019*), which gives daily maps of sea ice and snow cover. For each ATMS observation, a flag reporting the AutoSnow class percentage (sea ice, open water, snow-covered land, snow-free land) has been calculated; then, a threshold has been applied to discriminate between sea ice and open water pixels (sea ice AutoSnow class > 10 %) and between snow-covered and snow-free land pixels (snow-covered land AutoSnow class > 50 %). ATMS pixels have been classified into land, ocean, and coast pixels using a land-sea mask.

The land module discriminates between snow-free land and snow-covered land and identifies four different snow cover classes (Perennial, Winter Polar, Thin, and Deep Dry). It is based on a decision tree that makes use of a limited number of inputs (the ratio $TB_{23QV}/TB_{31QV}$ - **ratio**, the difference between $TB_{23QV}$ and $TB_{88QV}$ or Scattering Index - **SI**, 23 GHz pseudo-emissivity (i. e. the ratio between an observed brightness temperature (TB) and a near-surface temperature value) - **pem23**). The module has been described by *Camplani et al, 2021*.

For what concerns the ocean module, a simple relationship to distinguish between sea ice and open water observations has been identified. In Figure 2 a Cartesian plane where the x-axis represents 23 GHz observed TB and the y-axis represents the near-surface temperature ($T_{2m}$) is shown. - In the figure each point represents a pseudo-emissivity value, and the color describes the mean AutoSnow sea ice percentage within each bin (see Figure 2, left panel). It is possible to observe that open water (0 % of sea ice, blue) and sea ice (100 % of sea ice, red) are characterized by very different pseudo-emissivities. A transition area between open water and sea ice pseudo-emissivity values can be observed: these values characterize IFOVs where both open water and sea ice are present. The simple relationship for sea ice identification is reported in the left panel as a green line where the condition for sea ice identification is defined by Equation 1.

$$TB_{23QV} > T_{2m} - 96\,K$$

(1)

The analysis of the two low-frequency pseudo-emissivity values has been used to obtain information about sea ice characteristics downstream of the sea ice/open water identification. This is possible because sea ice surface emissivity for low-frequency channels can be approximated by the pseudo-emissivity, because the interaction





between the MW radiation and the atmosphere in the MW low-frequency channels is not significant, especially
in cold and dry conditions. For that reason, the 23 GHz pseudo-emissivity (**pem$_{23}$,**) and the 31 GHz pseudo-
emissivity (**pem$_{31}$**) have been used. In Figure 3 (top panel) it is possible to observe that there are sea ice classified
observations characterized by the contemporary presence of open water and sea ice above the bisector of the plane
and in correspondence with low emissivity values. In the center panel, where the color represents sea ice
occurrences, it is evident the presence of a cluster, in correspondence with high pseudo-emissivity, with two
"tails": the one above the bisector, the other below it. This behavior has been used to identify 3 different sea ice
classes (New Sea Ice, Broken Sea Ice, and Multilayer Sea Ice). This algorithm is based on a Nearest Neighbour
Method based on a set of points that define the areas of interest for each sea ice class. In Figure 3 (bottom panel)
a classification representation is reported: the markers represent the points on which the Nearest Neighbor method
is based. The names of the classes have been chosen by analyzing the physical properties of the classes and by
comparing the estimated emissivity spectra with previous works (*Hewison & English ,1999*, *Munchak et al, 2020*).
PESCA's upper working limits for $T_{2m}$ and atmospheric total precipitable water (TPW) have been established to
280 K and 10 mm, respectively (see *Camplani et al, 2021* for details). Moreover, the land module does not work
in the high elevation areas outside the polar regions (surface elevation > 2500 m for latitude < 67 ° N/S) because
the ATMS low spatial resolution does not allow for depicting the small-scale snow-cover variability that
characterizes the orographic regions. Within these well-defined limits, the PESCA manages to optimally
discriminate between sea ice, open water, snow-free land and snow-covered land. An analysis carried out using
the ATMS-CPR coincidence dataset highlights that the presence of cloud cover does not influence the overall
PESCA performances (not shown). The statistical scores (POD, FAR, HSS) of PESCA identification of sea ice
and snow cover (using AutoSnow as reference) are summarized in Table 1. In the defined environmental
conditions PESCA manages to optimally detect the presence of a frozen background (sea ice over the ocean, snow
covered land over the continental part) at the time of the satellite overpass. It is important to underline that the
difference between the HSS values is due to the different number of correct negative observations, which has a
strong influence on HSS values.
An analysis of the physical characteristics of the PESCA classes has been conducted by considering the mean $T_{2m}$,
the geographical and seasonal distribution associated with each class. For what concerns land class characteristics
and properties, refer to *Camplani et al., 2021*. For what concerns sea ice, the New Sea Ice class, which is detected
during the winter, at high latitudes, and for low temperatures represents the sea ice that forms during the winter.
The Broken Sea Ice class, which is predominant in the lower latitudes and whose occurrence increases during the
spring, represents the co-presence of sea ice and water typical of the intermediate seasons and in presence of
melting phenomena. The Multilayer Sea Ice class, which is detected only at high latitudes, for very low
temperatures, and with a constant occurrence during the year, represents the ice typical of those regions where
extremely cold conditions allow its presence during the whole year.

### 3.1.2 PESCA emissivity spectra estimation

The emissivity spectra of each class have been estimated by applying the PESCA algorithm to the cloud-free (0%
CPR cloud mask fraction, i.e., clear sky) ATMS observations in the ATMS-CPR dataset satisfying PESCA
working limits.  The ATMS clear-sky TBs measured for each PESCA surface class have been used as input to an
inverse radiative transfer model. The emissivity spectra have been estimated by calculating the mean and the
standard deviation of the emissivity values for each class (excluding the values lower than the 10th percentile and
higher than the 90th percentile). The emissivity spectra dependence on the ATMS viewing angle for polarized
surfaces has been neglected because an analysis of such dependence in the ATMS-CPR coincidence dataset has
shown that it is significant only for larger viewing angles (tot for >40 °). This is due to the fact that cross-track
scanning radiometers measure a signal (off-nadir) which derives from a mixture between the two polarizations
(e.g., quasi-vertical, QV, and quasi-horizontal, QH). As a consequence, although the emissivities of polarized
surfaces, such as open water surfaces, are strongly influenced by the viewing angle, for the cross-track scanning
radiometers the emissivity variation is compensated by the effect of the mixture of the two polarizations (see also
*Felde & Pickle*, 1995, *Prigent et al*, 2000, *Mathew et al*, 2008, *Prigent et al,* 2017).
The estimated spectra are shown in Figure 4 and Figure 5 for the land and ocean classes, respectively. It is possible
to observe that the classes are well-characterized from a radiometric point of view, showing distinct behavior of
the emissivity spectra (e.g., the mean values). However, all the classes present significant standard deviations at
high frequency, and some classes - such as the snow classes and the Broken Sea Ice class - present a high value



of standard deviation also at low frequency. The coast observations have been also considered as a class, however
its spectrum is not shown in Figures 4-5.
The RMSE between simulated clear-sky TBs - based on the mean emissivity values estimated for each class - and
the coincident observed clear-sky TBs appears to be too high to implement a robust signal analysis (>10 K). For
this reason, a refinement process for the emissivity spectra estimation based on machine learning techniques has
been developed downstream of the PESCA classification.
The refinement process has been based on a combination of an unsupervised classification technique (SOM) and
a supervised technique (LDA). The unsupervised classification identifies clusters characterized by the minimum
inner variability from a radiometric point of view. The supervised technique, instead, has the goal to identify the
previously obtained clusters, and the associated emissivity spectra, by using only input variables that are not
affected by the presence of clouds. The final emissivity spectra are estimated as the mean emissivity for each
frequency within each cluster identified by the supervised technique.  Therefore, as first step, the emissivity value
set has been clusterized in order to minimize the emissivity variability in each cluster by arranging the retrieved
emissivity values for six ATMS channels (23.8 GHz, 31.4 GHz, 50.3 GHz, 88.2 GHz, 165.5 GHz, and 183.31±7
GHz) in a one-dimensional SOM architecture. Then, an LDA model has been trained using the previously obtained
clusters as reference and using the PESCA input parameters (**pem$_{23}$**, **pem$_{31}$**, **ratio** and **SI**), some environmental
parameters (**TPW**, **T$_{2m}$**, surface pressure - **P$_{surf}$**) and ancillary variables (latitude - **lat**, Julian day - **jd**, altitude -
**DEM**, the maximum solar height during the day - **H$_{sun}$**) as input. The use of the LDA is necessary to associate an
emissivity spectrum to all the observations which are classified by PESCA, independently from the presence of
clouds. It is worth noticing that the whole predictor set of the LDA has resulted to be redundant; therefore, a subset
of the predictors has been selected for each class. The accuracy of the LDA classification is given by the ratio
between the number of hits (observations where LDA identifies the associated SOM class) and the total number
of observations; it can be considered as an indicator of the effectiveness of the LDA model in rebuilding the SOM
results.
The evaluation of the refinement process is based on the comparison between the simulated clear-skyTBs and the
observed clear-sky TBs for each PESCA surface class. An emissivity spectrum, (calculated as the mean of the
emissivity values for each cluster), together with ECMWF temperature and water vapor profiles, is used as input
in the RTM to simulate the clear-sky TBs. For each PESCA surface class, the number of clusters that
simultaneously lowers the errors (RMSE) between the simulated and observed clear-sky TBs at high frequency
(without lowering the classification accuracy too much) is chosen.
In Table 2 the number of clusters, the predictors selected, the accuracy, RMSE and percentage normalized root
mean squared error (NRMSE$_{\%}$) (*Gareth et al, 2013*) estimated on the test dataset, are reported for the 165.5 GHz
channel. NRMSE$_{\%}$ is defined by Equation 2.
$$NRMSE_{\%} = (\frac{RMSE}{\sigma} * 100)$$

394  (2)

where σ represents the standard deviation of the measured clear-sky TBs dataset in each PESCA class. It can be
considered an indicator of the effectiveness of the refinement process.
For some classes, such as the Ocean class, the refinement process leads to very low RMSE values (≈ 2 K). For
other classes, such as Deep Dry Snow and Broken Sea Ice, RMSE remains > 5 K even with a high number of
clusters, although there is a significant reduction compared to the initial variance in each class (NRMSE$_{\%}$ < 50).
This is due to the variability of snow-covered background within each class; in the worst scenario, the limited
number of predictors are insufficient to infer the emissivity spectrum at high frequency. Overall, the refinement
process allows to obtain a general improvement of the accuracy of the emissivity estimation for the PESCA
classes; however, for some classes, the high-frequency channel uncertainty remains significant.
The emissivity spectra obtained by PESCA refinement are used as inputs of the RTM to obtain clear sky simulated
TBs (TB$_{sim}$) to be compared to the actual observations (TB$_{obs}$). The comparison between TB$_{sim}$ with TB$_{obs}$ allows
to highlight and interpret the MW signal in presence of snowfall.
In Figure 6, the snowfall signal is represented as a function of the SWP for the 165.5 GHz channel and for different
PESCA classes. The red line and shaded areas represent the mean values and standard deviations of the difference
between the TB$_{obs}$ and the TB$_{sim}$ ($\Delta$TB$_{obs-sim}$ =TB$_{obs}$ -TB$_{sim}$) for SWP bins calculated for observations where 2CSP
SWP > 0 kg m$^{-2}$. The blue lines represent the uncertainty due to surface emissivity variability for each PESCA.
They are centered on the estimated bias for each class (close to 0 K) and the dashed lines correspond to the



standard deviation of $\Delta TB_{obs-sim}$ in clear sky conditions. A clear scattering signal ($\Delta TB_{obs-sim} < 0$) is observed over
all the classes considered for intense snowfall events (SWP > 1 kg m$^{-2}$). For lower SWP values, the signal is more
ambiguous and changes with the background surface. While over Land there is a clear scattering signal for SWP
> 0.1 kg m$^{-2}$, over the Perennial Snow class a scattering signal can be observed only for SWP > 0.5 kg m$^{-2}$ . For
SWP < 0.1 kg m$^{-2}$, the mean $\Delta TB_{obs-sim}$ for snowfall observations is less than its standard deviation in clear sky.
This is due mainly to the emissivity variability for each surface class, and to the error introduced by the use of
model-derived temperature and water vapor profiles in the RT simulations. However, while for the Land class the
mean $\Delta TB_{obs-sim} < 0$ K can be explained as a predominant scattering effect for all SWP values, for the Perennial
Snow class the mean $\Delta TB_{obs-sim} > 0$ K can be interpreted as a predominant emission signal with respect to the
radiatively cold background (Figure 5). The Thin Snow class shows an intermediate behavior: for SWP < 0.1 kg
m$^{-2}$ the red shaded area within the RMSE limits (blue lines) of the RT simulations denotes the difficulty in
interpreting the signal, while a clear scattering signal can be observed for SWP > 0.3 kg m$^{-2}$.

**3.2 ANN Design for snowfall retrieval**

The snowfall detection and estimation modules have been based on ANNs. Four ANNs have been developed: two
for the detection of SWP and SSR and two for the SWP and SSR estimate. The performance of more than 50
architectures have been tested, by varying the number of layers, the number of neurons for each layer, and the
activation functions. The final architecture, for all modules, is composed of four layers: an input layer with a
neurons number equal to the predictor number, and a hyperbolic tangent function as the activation function, a first
hidden layer (60 neurons), and hyperbolic tangent function, a second hidden layer (30 neurons), with a logarithmic
tangent function. At the same time, several predictor sets have been tested combining in different ways ATMS
$TB_{obs}$, $\Delta TB_{obs-sim}$, PESCA surface class, ATMS angle of view, ancillary information (surface elevation from a
Digital Elevation Model), and model-derived environmental variables ($T_{2m}$, TPW, and freezing level height). In
Table 3 the statistical scores of the algorithm performance for the SSR detection module obtained for different
predictor sets are reported. It is possible to see that the best performance is obtained when the predictor set is
composed of ATMS $TB_{obs}$ and $\Delta TB_{obs-sim}$, (besides PESCA surface flag, the pixel elevation and the cosine of the
viewing angle). In particular, it is possible to observe an improvement of the detection capabilities with respect to
a predictor set composed of ATMS $TB_{obs}$ and environmental parameters. This indicates that the computation of
the multi-channel clear-sky TBs at the time of the overpass through the estimation of the PESCA surface class
emissivity spectra and its deviation from the measured TBs, derived from the previous surface radiometric
characterization obtained by PESCA, plays a fundamental role in snowfall retrieval. It provides essential
information to the ANN to be able to exploit the subtle snowfall-related signal in ATMS measurements. This is
the most innovative aspect of HANDEL-ATMS.
The final set of predictors is composed by the 16 ATMS $TB_{obs}$, by the $\Delta TB_{obs-sim}$ set for the 16 ATMS channels,
from the PESCA classification flag, the pixel elevation (obtained from a DEM) and the cosine of the view angle.

**4. Results**

**4.1 HANDEL-ATMS Performances**

In Table 4 the statistical scores of  HANDEL-ATMS detection module performances are reported in terms of
POD, FAR and HSS. It is possible to observe good detection capabilities both for SWP and SSR modules (POD
> 0.8, FAR < 0.2).
In Figure 7 and in Figure 8 the dependence of HANDEL-ATMS snowfall detection statistical scores on TPW and
on $T_{2m}$ is reported. In both figures, it is possible to observe that the SWP detection capabilities improve (with an
increase of POD and HSS and a decrease of FAR) with increasing humidity and temperature. This is due to the
combined effect of a stronger scattering signal associated with more intense snowfall events - linked to moister
and warmer environmental conditions - and to the lower transmissivity of the atmosphere which masks the
background surface signal, reducing its impact and the uncertainties linked to its variability. On the other hand,
colder and drier conditions are usually linked to background surface types characterized by high radiometric
variability such as Perennial Snow and Winter Polar Snow classes, which cause uncertainty in emissivity
estimation. It is possible to observe that in Figure 7 SSR detection capabilities show a maximum HSS value for
TPW between 3 mm and 5 mm, and then there is a slight decrease due to the decrease of POD. A similar situation
can be observed in Figure 8, where HANDEL-ATMS SSR HSS reaches a maximum between 250 K and 275 K
and then decreases and it is lower than for SWP. This is due to the fact that PMW measurements respond mostly
to the snow in the atmospheric column and in moister/warmer conditions the presence of snow in the atmosphere





is not always linked to surface snowfall. In both cases, it is worth noting that also considering very dry (TPW ≈ 2
mm) or very cold ($T_{2m}$ ≈ 240 K) conditions, HANDEL-ATMS shows good detection capabilities, in spite of the
uncertainties linked to the modeling of the background surface and the weakness of the signal in such conditions.
Moreover, it is worth noticing that, also considering very low SWP and SSR values (SWP ≈ 0.001 kg m$^{-2}$ , SSR
≈ 0.001 mm h$^{-1}$), HANDEL-ATMS manages to detect around 60 % of the snowfall events. Similar considerations
can be done also for the different background surfaces. The detection capabilities are influenced both by the typical
environmental conditions of each PESCA class and by the uncertainties linked to the emissivity estimation. In
Table 6 the statistical scores of the algorithm performance by considering each PESCA class for both the SWP
and the SSR detection module are reported. It can be observed that, also considering specifically the classes where
the detection is more problematic both for the uncertainties linked to the emissivity retrieval (see Table 2), for the
extremely dry and cold environmental conditions, and for the low intensity of the snowfall events,   such as
Perennial Snow or Winter Polar Snow, HANDEL-ATMS has good detection capabilities (POD and FAR values
greater than 0.7 and less than 0.25, respectively, for both SWP and SSR). These results provide evidence that
HANDEL-ATMS can be used to analyze snowfall occurrence in the polar regions.
The error statistics of the two estimation modules are reported in Table 5 in terms of bias, RMSE and the
coefficient of determination $R^2$ , which is defined by Equation 3.
$$R^2 = 1 - \frac{RMSE^2}{std^2}$$

481 (3)
It is worth noticing that the biases are negligible for both modules while RMSE values are comparable to the light
events recorded in the dataset. Moreover, RMSE and $R^2$ values are respectively higher and lower for the SSR
module than for the SWP module; this is due to the fact that the PMW signature is mainly related to the presence
of snow in the atmosphere rather than to the surface snowfall rate. In Figure 9 the density scatterplots between the
SWP and SSR values retrieved by HANDEL-ATMS and the 2CSP corresponding values are reported. For both
modules, an overestimation can be observed for very light snowfall (SWP < $10^{-2}$ kg m$^{-2}$ and SSR < $10^{-2}$ mm h$^{-1}$),
while there is a very good agreement for higher SWP and SSR values.
. Generally, it can be observed that, although HANDEL-ATMS is able to detect extremely light snowfall events,
it does not have the sensitivity to correctly estimate their intensity.

**491 4.2 A Case Study: Greenland-2016/04/24**

The case study reported corresponds to the observation of a moderately light snowfall event over the central part
of Greenland that occurred on 24 April 2016. ATMS overpass is between 14:51:23 U.T.C. and 14:57:47 U.T.C.,
while the CPR overpass is between 15:05:25 U.T.C. and 15:11:45 U.T.C., with a time difference of 14 minutes
and 2 seconds. This event presents several characteristics typical of high latitudes, such as light snowfall rate, dry
and cold atmospheric conditions, and presence of a frozen background surface, a typical case of interest for the
application of HANDEL-ATMS.
In Figure 10 PESCA classification is reported. The entire territory of Greenland, except for a narrow area on the
southwestern coast, is identified as a snow-covered surface; PESCA identifies the Perennial Snow class in the
central part of Greenland and along CloudSat track and the Polar Winter Snow class near the northern shoreline.
CloudSat overpasses the central part of the island, and CPR track is along the central part of the ATMS swath.
In Figure 11 a synopsis of the event along the CPR track is reported: the environmental parameters, $T_{2m}$ and TPW,
the 2CSP SWP and SSR values, the cross-section of CPR reflectivity, with the DARDAR supercooled water
information superimposed (in magenta). Moreover, the PESCA surface classification, and the TBs of the main
ATMS high-frequency channels along the CloudSat track are reported. The event is characterized by dry
conditions (TPW < 5 mm) and $T_{2m}$ below 273 K, except over the coast. CPR observes a cloud system linked to
the snowfall event between 68 °N and 76 °N; DARDAR detects the presence of a supercooled water layer at the
cloud top between 68 °N and 72 °N and indicates the presence of supercooled droplets embedded in the deeper
cloud associated to the more intense snowfall. 2CSP detects a light snowfall event in the inner part of the island
and a more intense event, with a peak of intensity between 72 °N and 76 °N, near the shoreline.  For what concerns
the associated ATMS observations, an increase of the 88 GHz and 165 GHz TBs is observed in coincidence with
the supercooled water layer; on the other hand, only a slight decrease of 165.5 and 183.3+7 GHz TBs can be
observed in coincidence with the snowfall intensity peak.



In figure 12 the maps of the $TB_{obs}$ at 165.5 GHz (top panel) and the $\Delta TB_{obs-sim}$ at 165.5 GHz (bottom panel) are
reported. In the top panel, it is possible to observe that, despite the snowfall event, there is not a clear TB scattering
signal in the area where 2CSP detects snowfall (70 °N -76 °N, 40 °W -70 °W); instead a slight increase in the TBs
can be observed in the area where DARDAR detects the supercooled water layer. The simulation of the clear-sky
TBs ($TB_{sim}$) allows to observe an emission signal ($\Delta TB_{obs-sim} > 0$) over the central part of the ATMS swath due to
the combined effect of supercooled liquid water emission (as evidenced by DARDAR supercooled water layers)
and radiatively cold surface background. Only near the shoreline, the $TB_{obs}$ is slightly lower than the $TB_{sim}$
($\Delta TB_{obs-sim} < 0$). In Figure 13 the results of the HANDEL-ATMS four modules are reported. It is worth noting
that both detection modules find snowfall in the central region of Greenland and near the northern coast. The
estimated intensity of this event is generally light (SWP < 0.1 kg m$^{-2}$ and SSR < 0.1 mm h$^{-1}$) except over the
western coast, where SWP reaches 0.5 kg m$^{-2}$ and SSR reaches 1 mm h$^{-1}$. It is worth noticing that HANDEL-
ATMS detects snowfall also where there is an emission signal ($\Delta TB_{obs-sim} > 0$).
Finally, a comparison between the HANDEL-ATMS and the 2CSP is reported in Figure 14. It is worth noting that
there is a substantial agreement on the snowfall detection of the two products. It can be observed that HANDEL-
ATMS tends to overestimate both SWP and SSR in presence of very light snowfall (2CSP SWP < 0.05 kg m$^{-2}$
and SSR <0. 1 mm h$^{-1}$, between 68 °N and 72 °N), consistently with what shown in Fig. 9; on the other hand,
there is a good agreement between 72 °N and 76 °N, where snowfall intensity increases.
The analysis of this case study demonstrates that the algorithm can interpret the ambiguity of the
emission/scattering signal associated with snowfall described in Section 4.1 and so efficiently detect and quantify
snowfall even in extreme conditions.

**4.3 Comparison with SLALOM-CT**

SLALOM-CT has been introduced in Section 1. It presents some similarities with HANDEL-ATMS: it is based
on an ANN approach and uses CPR-2CSP product as reference. On the other hand, substantial differences have
to be highlighted: SLALOM-CT was designed to operate on a global scale, while HANDEL-ATMS has been
developed specifically for the extreme conditions typical of high latitudes. Moreover, the predictor sets are
different: in addition to TB observations, SLALOM-CT relies on several model derived environmental
parameters, while HANDEL-ATMS relies on differences between simulated clear-sky TBs and observed TBs
($\Delta TB_{obs-sim}$), and therefore on the estimation of the background surface emissivity at the time of the overpass, as
described in Section 3.
In Table 7 a comparison between the statistical scores of the detection performances of the two algorithms is
reported for different environmental conditions. The comparison has been carried out considering the same
observations of the ATMS-CPR coincidence dataset. It can be observed that, as in colder and drier conditions, the
differences between the two algorithm performances increase: HANDEL-ATMS shows better snowfall detection
capabilities than SLALOM-CT. Considering the working limits of HANDEL-ATMS, POD increases by 2 % and
FAR decreases by 8 %; however, if only very cold/dry conditions are considered ($T_{2m} < 250$ K, TPW < 5 mm),
POD increases by 7 % and FAR decreases by 16 %; for extremely dry/cold conditions ($T_{2m} < 240$ K, TPW < 3
mm), typical of the inner part of Greenland and Antarctica, POD increases by 18 % and FAR decreases by 16 %.

**5 Conclusions and Future Perspectives**

In this paper a new snowfall retrieval algorithm, the High lAtitude sNow Detection and Estimation aLgorithm for
ATMS (HANDEL-ATMS), is described. The algorithm is based on machine learning techniques, and it has been
trained against CPR snowfall products. It has been developed specifically for the extreme conditions typical of
high latitude regions. The driving and innovative principle in the algorithm development is the exploitation of
the full range of ATMS channel frequencies to characterize the frozen background surface radiative properties at
the time of the overpass to be able to better isolate and interpret the snowfall-related contribution to the measured
multi-channel upwelling radiation. This approach is proven to be effective for snowfall detection and
quantification at high latitudes, particularly in presence of a frozen (snow-covered land or sea ice) background,
also compared to other state-of-the art machine learning based methods.
HANDEL-ATMS can detect snowfall at high latitudes in good agreement with CPR. The estimation modules tend
to overestimate the intensity of light snowfall events (SWP < 10$^{-2}$ kg m$^{-2}$) but it shows good accuracy for more
intense snowfall events (SWP > 10$^{-2}$ kg m$^{-2}$, SWP < 1 kg m$^{-2}$). It is worth noting, however, that the uncertainty
associated with the surface emissivity estimation in some conditions affects the capabilities of HANDEL-ATMS
to correctly interpret the snowfall signature. Such uncertainty, related to the difficulty in correctly modeling the





566 intrinsic variability of snow cover surface emissivity, propagates in the radiative transfer simulation of the clear-
567 sky TBs used as input in the algorithm. Despite these limitations, it is worth noticing that the development of an
568 algorithm capable of retrieving snowfall at high-latitudes conditions with good accuracy is an important
569 development in the climate science field. The possibility to exploit a big amount of data guaranteed by near-polar
570 operational satellites carrying ATMS radiometers allows obtaining snowfall estimates characterized by a full
571 coverage of the polar areas and a high temporal resolution. Moreover, the future European MetOp Second
572 Generation (MetOp-SG) mission, with the launch of the Sat-A Microwave Sounder (MWS), with characteristics
573 very similar to ATMS, will provide another instrument to improve global snowfall monitoring. The HANDEL
574 methodology will be also adapted to be able to exploit MWS measurements in the future. The possibility to exploit
575 a wide range of microwave channels allows obtaining a characterization of the background surface at the time of
576 the overpass. This element is fundamental to obtain a characterization of the snowfall signature (especially for the
577 extreme environmental conditions typical of high latitudes), and an accurate snowfall retrieval. The capability to
578 estimate snowfall events with a high temporal resolution is ancillary to the development of a continuous snowfall
579 monitoring system over the high latitude areas and to analyze the snowfall climatology in these areas. This
580 research could have important impacts in climate change studies; snowfall is predominant over rain in the high-
581 latitude areas, and it has been proven that climate change has a strong impact on snowfall regime in these areas.
582 Future research activities will tackle some open issues. The estimation of the surface emissivity and the simulated
583 clear-sky multi-channel TBs needs to be further improved, either by considering other predictor sets or by using
584 a different technique for the emissivity spectra refinement process, or by using more advanced radiative transfer
585 models. Another important aspect is the quantification of the error linked to the background emissivity estimation
586 on the snowfall detection capabilities. This would be also useful for the development of modules for mountainous
587 areas, which have not been analyzed in this study. Moreover, the effect on the algorithm snowfall detection
588 capabilities of the uncertainties linked to the model-derived environmental variables (e.g., temperature and water
589 vapor profile), which are used in the clear-sky TB simulations, should be investigated. The use of the ATMS water
590 vapor (183 GHz band) and temperature (50 GHz band) sounding channels to characterize the atmospheric
591 conditions at the time of the overpass in order to avoid the use of model-derived data is another subject of future
592 research. Moreover, the possible development of a separate supercooled liquid water detection module could also
593 be evaluated, similarly to what is done in other PMW snowfall detection and estimation algorithms (*Rysman et*
594 *al., 2018*, *Sanò et al., 2022*). Such information can be exploited to improve snowfall detection and estimation
595 capabilities since the emission by the cloud droplets in dry conditions tends to mask the snowfall scattering signal
596 (see *Panegrossi et al, 2017*, *Panegrossi et al, 2022*), and adds larger uncertainties in the CPR snowfall products
597 (*Battaglia & Panegrossi, 2021*). Finally, since the algorithm has been developed only for specific environmental
598 conditions typical of high latitudes (dry and cold atmosphere) an integration with other approaches, such as
599 SLALOM-CT, designed for global estimation of snowfall, could be considered in the future to improve global
600 snowfall monitoring based on ATMS and future cross-track scanning radiometers.
601

602 **Data availability**
603 ATMS data are provided by the NOAA CLASS facility www.avl.class.noaa.gov/ (last access 4 april 2023), CPR
604 data are distributed by the CloudSat data processing center https://www.cloudsat.cira.colostate.edu/ (last access
605 4 april 2023), DARDAR data are available from the ICARE FTP server of the University of Lille (ftp.icare.univ-
606 lille1.fr, last access 4 april 2023) and ECMWF operational forecasts are distributed by ECMWF through the
607 MARS facility via the ECGATE cluster. AutoSonw data are provided by the NOAA Satellite and Information
608 Service https://satepsanone.nesdis.noaa.gov/northern_hemisphere_multisensor.html (last access 4 april 2023).

609 **Author Contribution**
610 Conceptualization, A.C., P.S., D.C.; methodology, A.C., P.S., D.C.; software, A.C.; validation, A.C.; formal
611 analysis, A.C.; investigation, A.C., P.S., D.C., G.P.; data curation, A.C. and D.C.; writing—original draft
612 preparation, A.C.; writing—review and editing, A.C., P.S., D.C., and G.P.; visualization, A.C.; supervision, G.P.;
613 project administration, G.P.; funding acquisition, G.P. All authors have read and agreed to the published version
614 of the manuscript.

615 **Competing Interests**
616 The authors declare no conflict of interest. The funders had no role in the design of the study; in the collection,
617 analyses, or interpretation of data; in the writing of the manuscript, or in the decision to publish the results.



**Acknowledgements**

This work was carried out under the RainCast study (ESA Contract No. 4000125959/18/NL/NA) and by the EUMETSAT Satellite Application Facility for Operational Hydrology and Water management (H SAF) Fourth Continuous and Operations Phase (CDOP-4). Andrea Camplani was supported by the Ph.D. program in Infrastructures, Transport Systems and Geomatics at the Department of Civil, Constructional, and Environmental Engineering at Sapienza University of Rome. The authors would like to thank EUMETSAT and the NASA Precipitation Measurement Mission (PMM) Research Program for supporting scientific collaborations between H SAF and GPM, and the PMM Science Team. The authors wish to express their sincere gratitude to Joe Turk (NASA JPL) and Alessandro Battaglia are warmly acknowledged for useful interactions and discussions during the algorithm development and validation and to Mattia Crespi for the scientific support to Andrea Camplani during the Ph.D. program.

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



**Figures**



**Figure 1: HANDEL-ATMS workflow diagram (read text for details)**

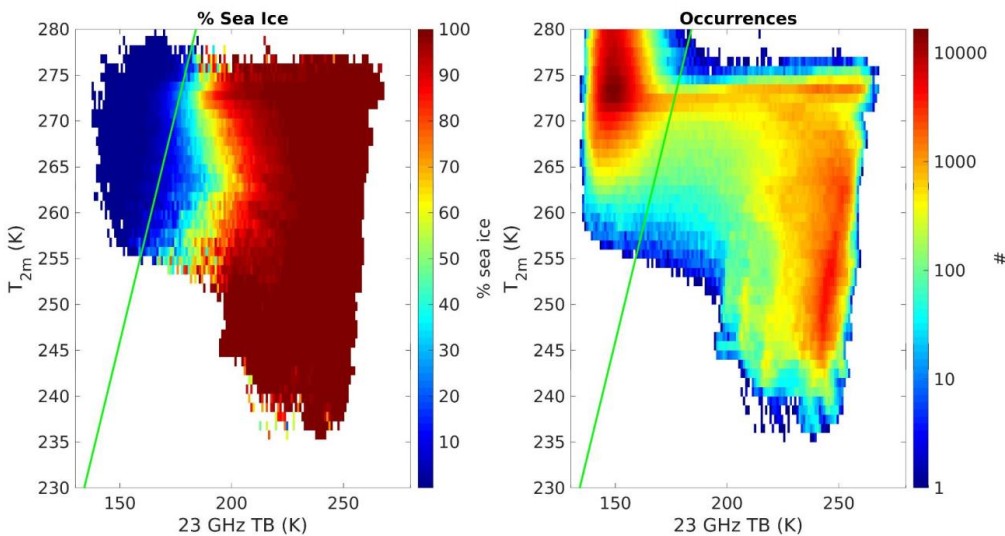

**Figure 2: Sea Ice Detection: 23 TB-$T_{2m}$ Plan. The color represents the mean AutoSnow sea ice percentage within each bin (left) and the observation occurrence (right).**

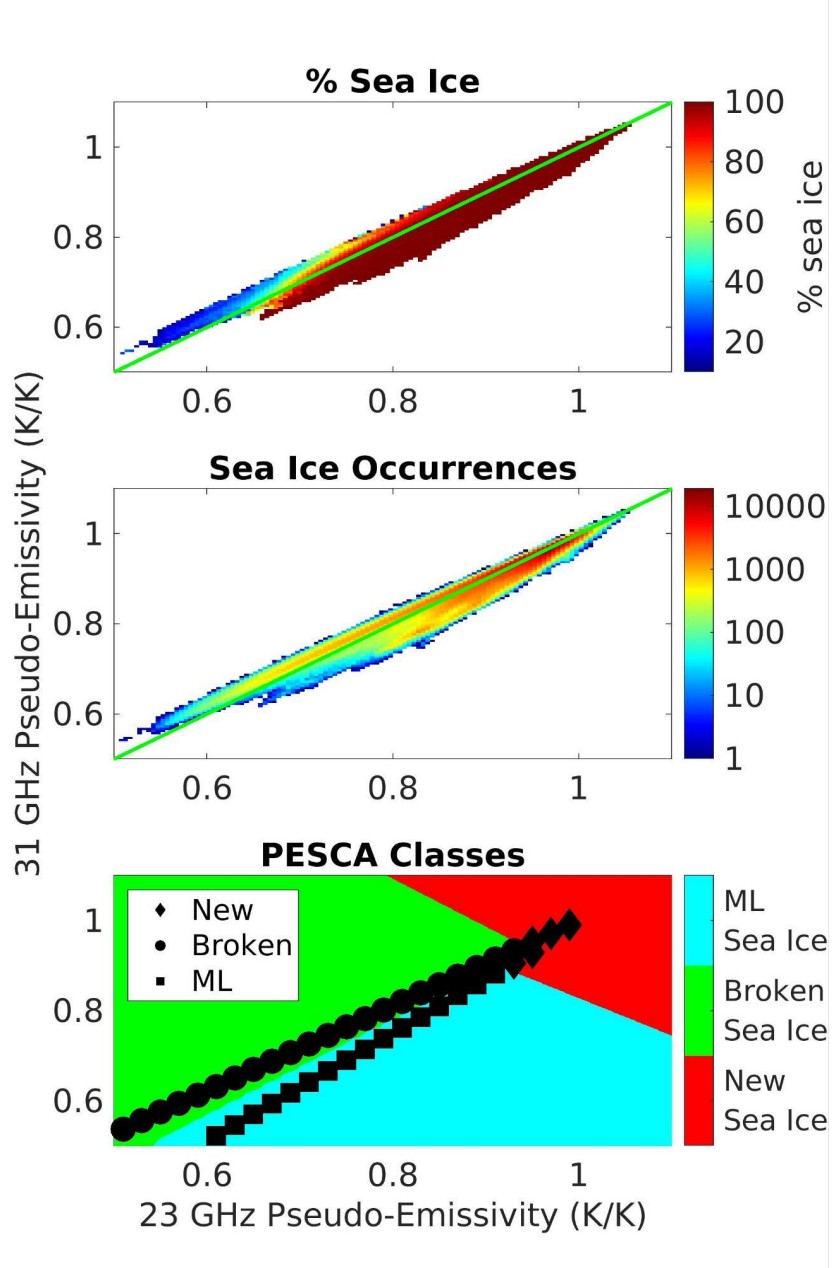

**Figure 3: Sea Ice Detection: relationship between 31 GHz Pseudo-Emissivity (y-axis) and 23 GHz Pseudo-Emissivity**
**(x-axis). The color represents the mean AutoSnow sea ice percentage within each bin (top) and the observation**
**occurrence (center) and the PESCA classification with the Nearest Neighbor markers (bottom).**



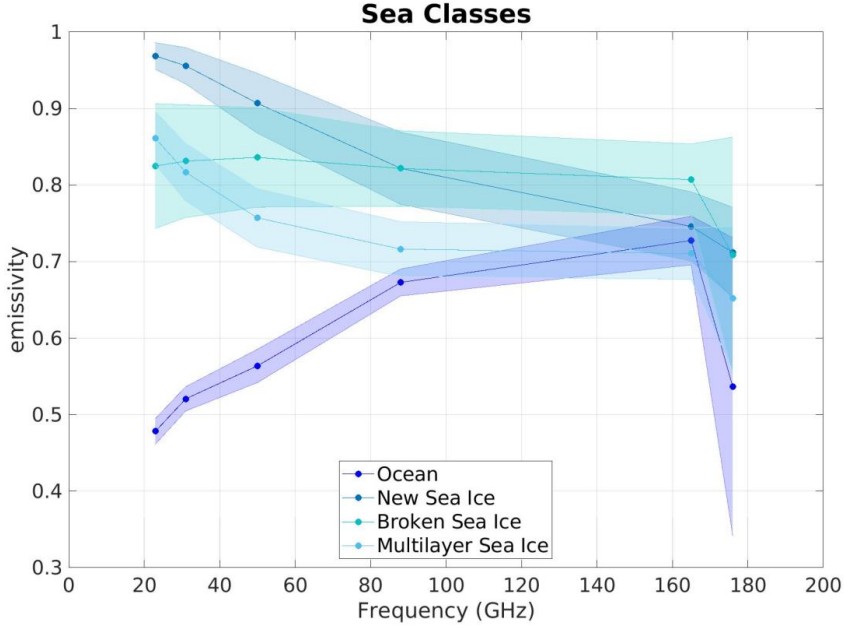

**Figure 4: Emissivity Spectra for PESCA Sea Classes. The continuous lines represent the mean values of the emissivity**
**while the shaded areas represent the standard deviation calculated at the ATMS reference frequencies (23.8 GHz, 31.4**
**GHz, 50.3 GHz, 88.2 GHz, 165.5 GHz, and 183.3 ±7 GHz) represented by the dots.**

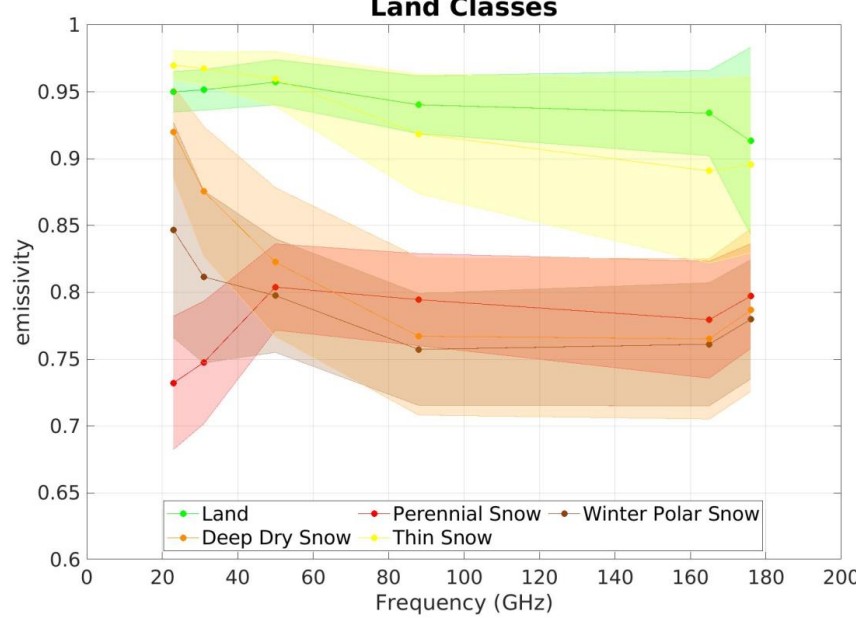

**Figure 5: Same as Figure 4 but for PESCA Land Classes.**



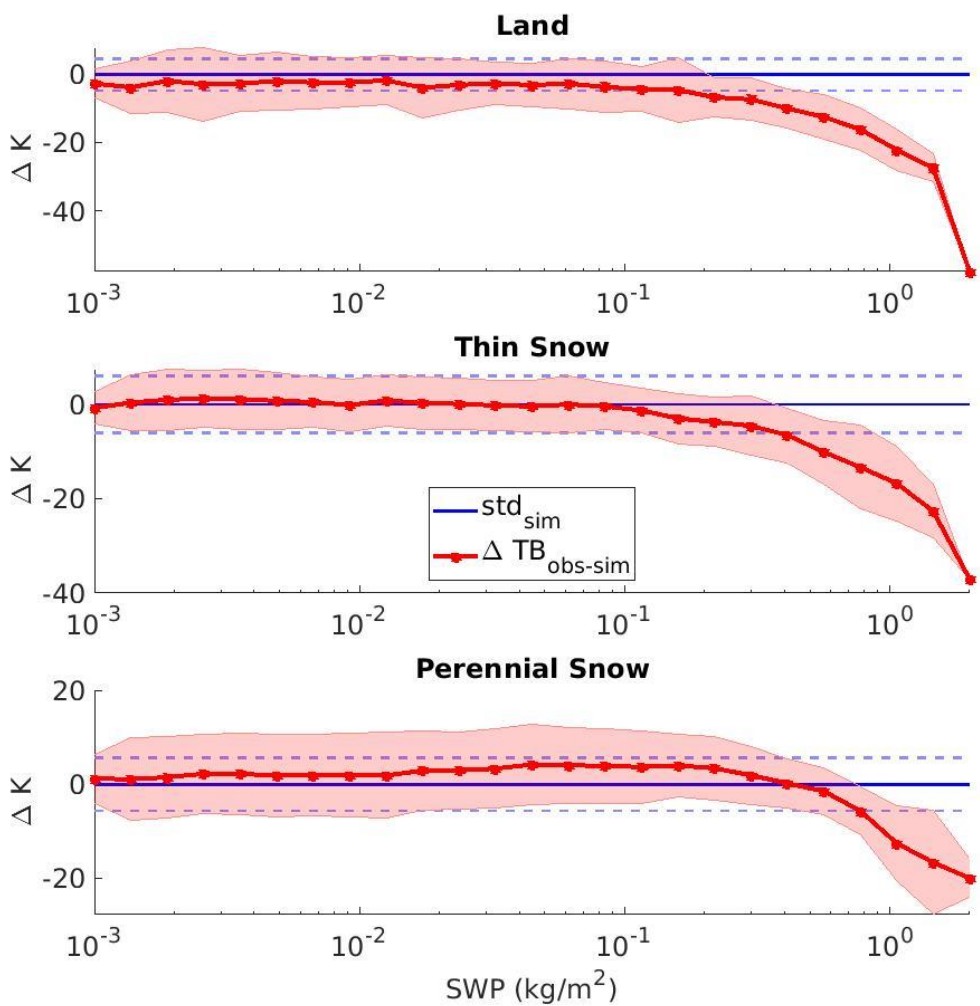

**Figure 6: 165.5 GHz Snowfall Signature as a function of SWP for three Land surface Classes. The red line**
**and shaded areas represent the mean values and standard deviations of ΔTB$_{obs-sim}$ (i.e., the snowfall**
**signature) while the blue lines are centered on the estimated bias and standard deviation of ΔTB$_{obs-sim}$ in**
**clear sky conditions for the corresponding PESCA surface class.**

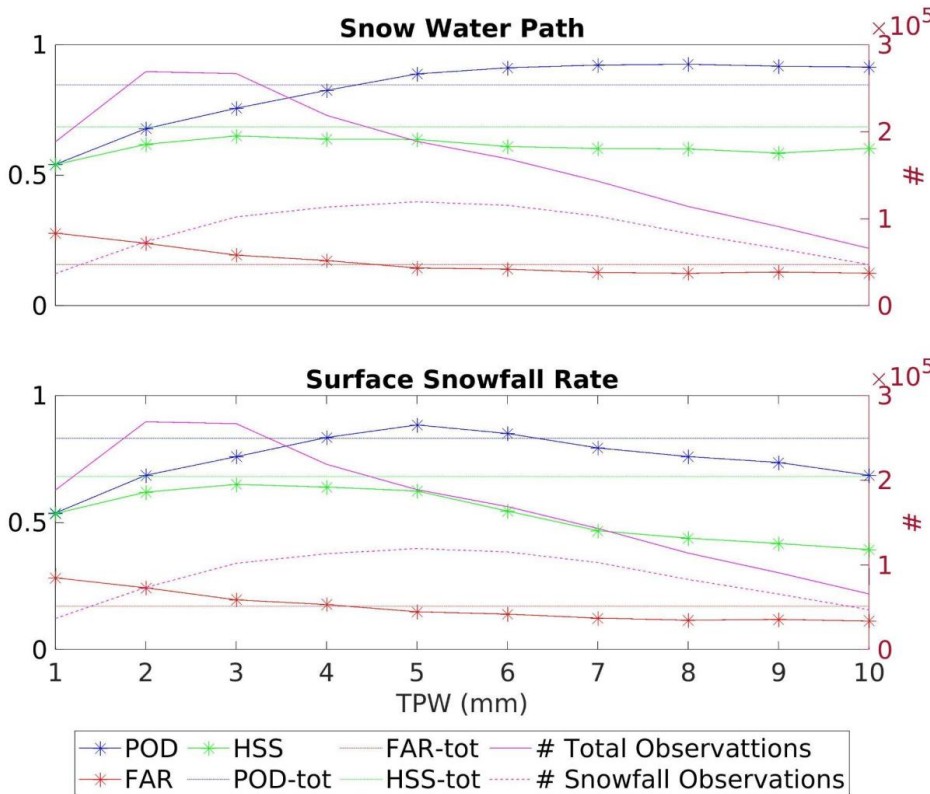

**Figure 7 HANDEL-ATMS SWP and SSR Detection Performances for different bins of TPW. The left y-axis**
**reports POD, FAR and HSS vales, while the right y-axis reports the total number and snowfall observations**
**in the  dataset.POD-tot, FAR-tot and HSS-tot (dotted lines) represent the statistical scores estimated on the**
**total dataset (values reported in Table 2).**





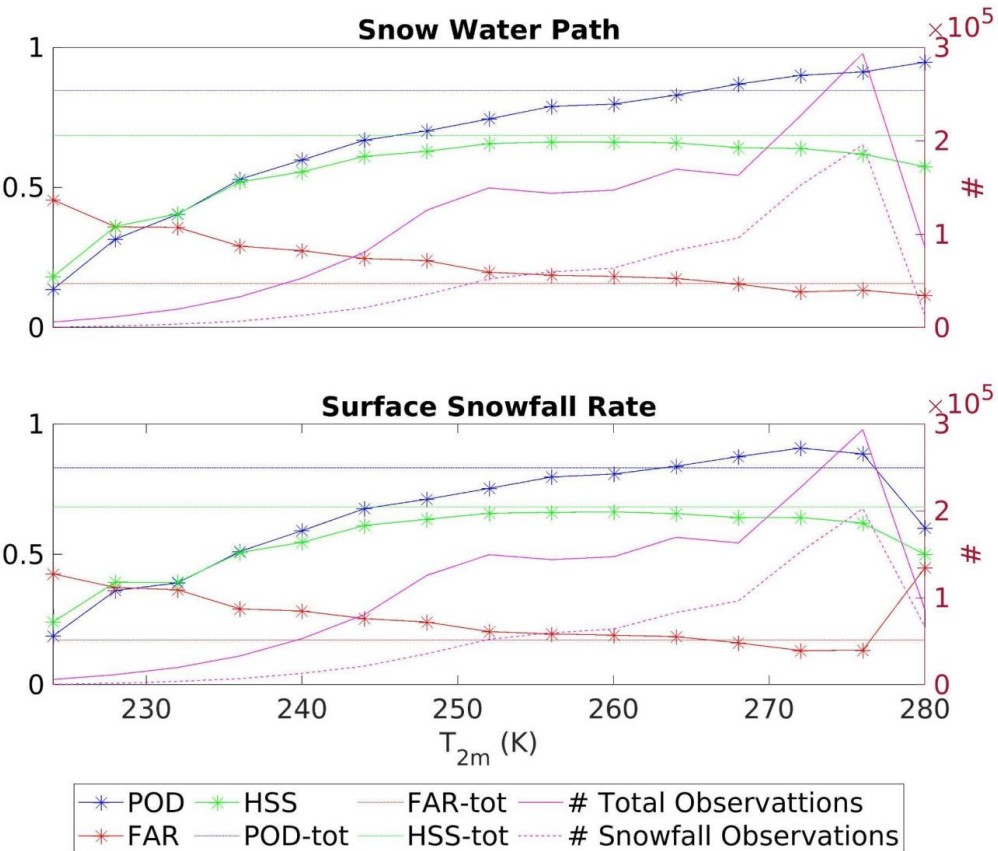

**Figure 8: Same as Figure 7 but for $T_{2m}$ bins.**

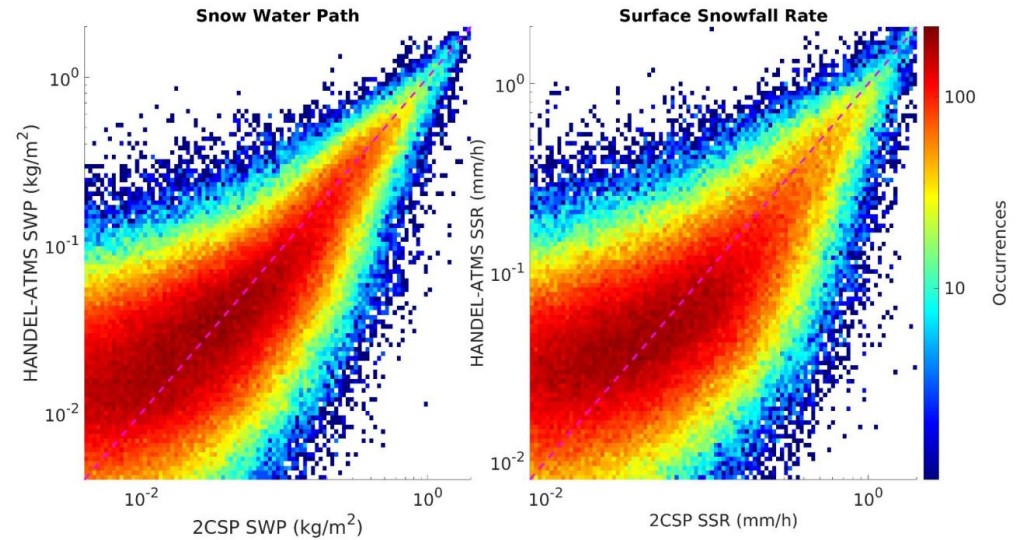

**Figure 9: 2D Histogram reporting HANDEL-ATMS SWP (left) and SSR (right) estimation and 2CSP. The colorbar represents the observation number for each HANDEL-ATMS/2CSP bin.**

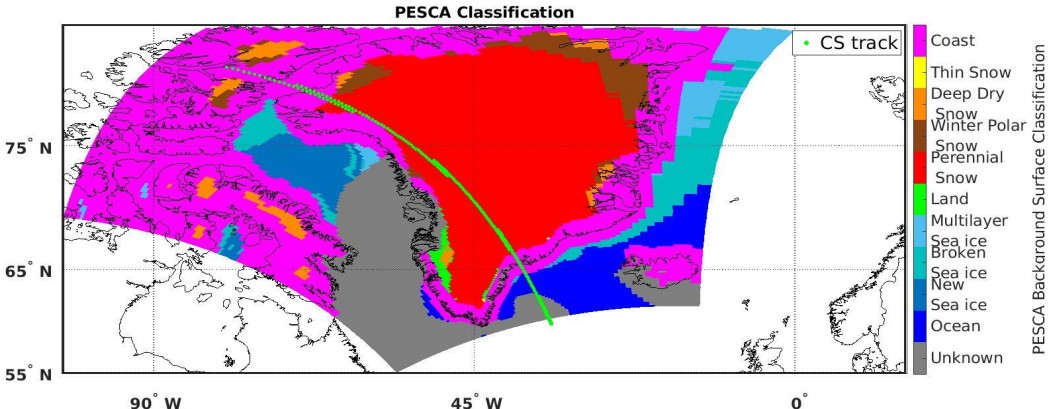

**Figure 10: Greenland - 2016/04/24 - PESCA Background Surface Classification.**



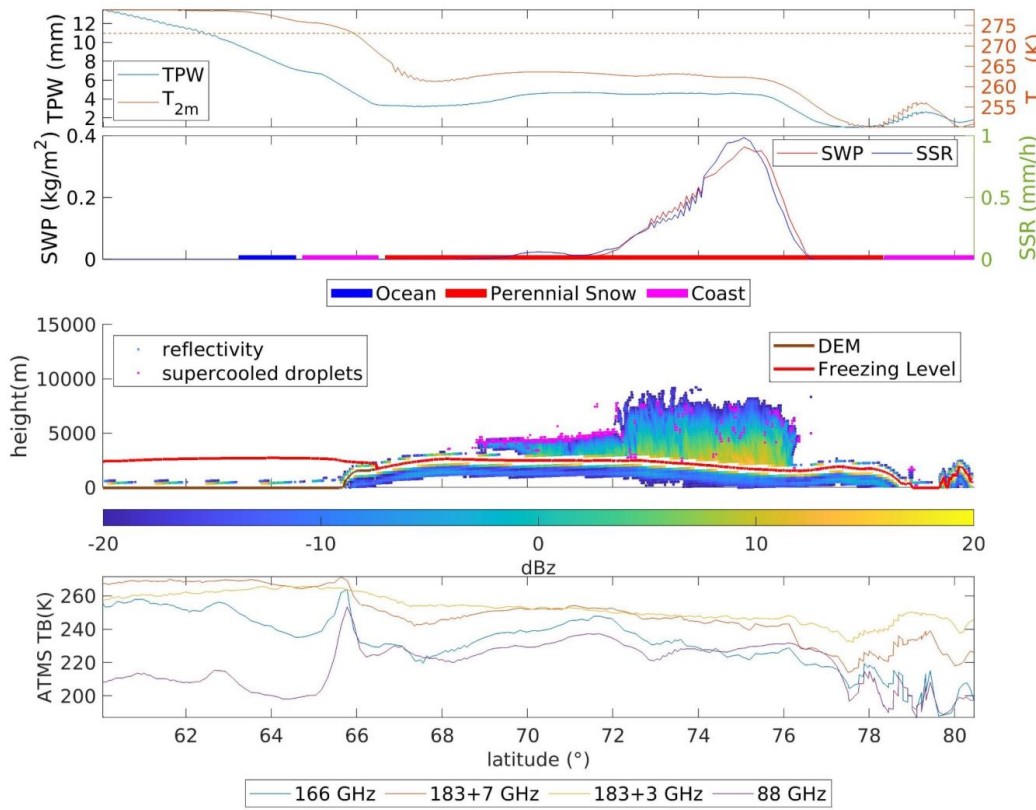

**Figure 11: Greenland - 2016/04/24 - Synopsis along CloudSat Track. The first panel shows the ECMWF TPW and $T_{2m}$ values along the CloudSat track. In the second panel, the 2CSP SWP (left) and the SSR (right) values are reported, besides the PESCA classification along CloudSat track. In the third panel, the CPR reflectivity (values are reported in the colorbar below), the supercooled water droplets detected by DARDAR (magenta points) are shown. Also the Digital Elevation Model (brown line) and the ECMWF Freezing Level (red line) along CloudSat track are reported. In the bottom panel the observed TBs of the main high-frequency channels (88 GHz, 166 GHz, 183+3 GHz, 183+7 GHz) along CloudSat track are shown.**

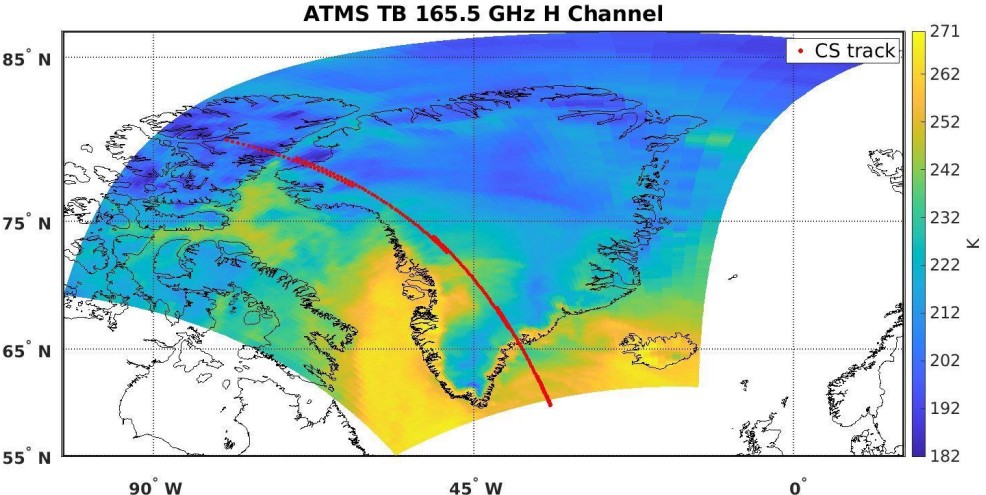

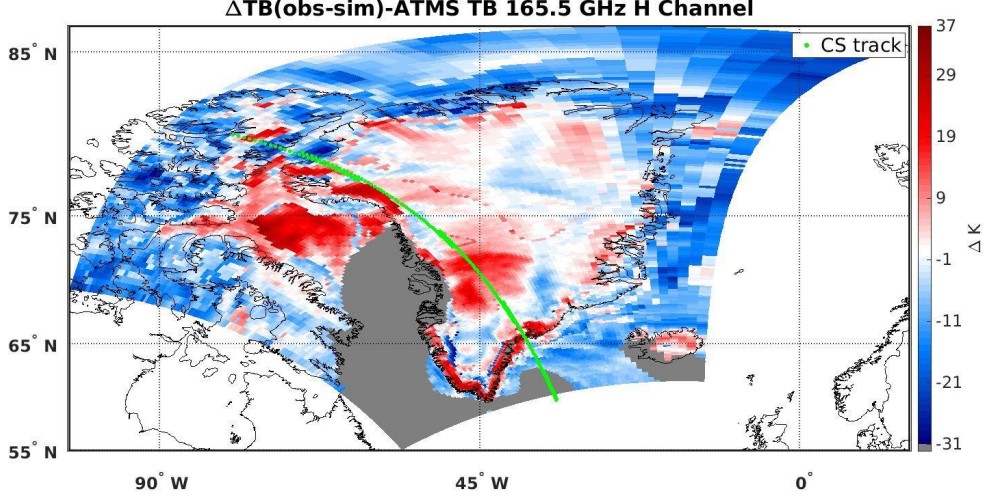

Figure 12: Greenland - 2016/04/24 - 165 GHz Channel measured TB (TB$_{obs}$) (top panel) and the deviation of
TB$_{obs}$ from the simulated clear-sky TBs ( $\Delta\mathbf{TB_{obs-sim}}$) (bottom panel)



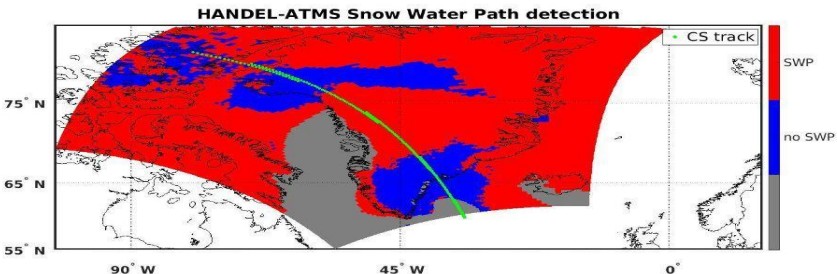

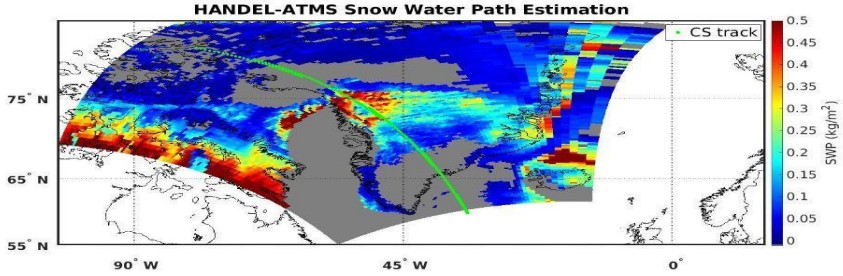

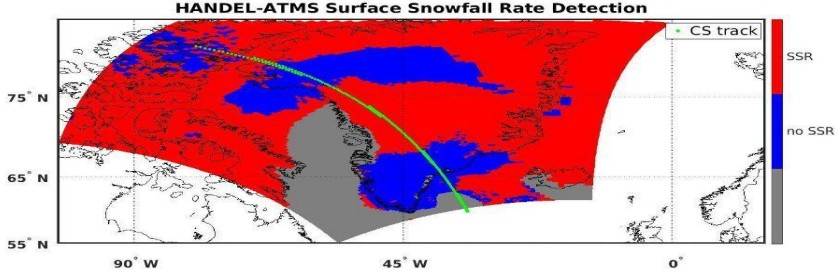

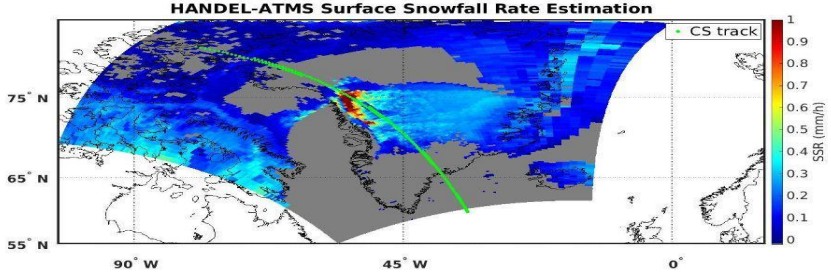

**Figure 13: Greenland - 2016/04/24 - Maps of the HANDEL-ATMS module's output: the SWP detection mask (top panel), the estimated SWP (kg m⁻²) (second panel), the SSR detection mask (third panel), the estimated SSR (mm h⁻¹) (bottom panel).**





| Class | n clusters | accuracy | 165.5 GHz RMSE (K) | 165.5 GHz NRMSE$_\%$ | Predictor Set |
|---|---|---|---|---|---|
| Ocean | 2 | 0.9 | 3.37 | 44 | $P_{surf}$ - TPW - $T_{2m}$ |
| New Sea Ice | 3 | 0.74 | 4.52 | 48 | SI - $T_{2m}$ - $P_{surf}$- ratio - jd - $pem_{23}$ |
| Broken Sea Ice | 16 | 0.56 | 5.34 | 41 | $pem_{23}$ - TPW - SI - $P_{surf}$ |
| Multilayer Sea Ice | 9 | 0.53 | 4.38 | 34 | $pem_{31}$ - SI - TPW - $T_{2m}$ - $pem_{23}$ - $P_{surf}$ |
| Land | 2 | 0.87 | 4.57 | 52 | DEM - jd - TPW |
| Perennial Snow | 8 | 0.65 | 5.98 | 54 | $pem_{23}$ - jd - SI - $pem_{31}$ - lat |
| Winter Polar Snow | 5 | 0.76 | 5.87 | 37 | $pem_{31}$ -SI - lat -$H_{sol}$ - $pem_{31}$ - jd |
| Deep Dry Snow | 15 | 0.34 | 6.77 | 45 | SI - $pem_{31}$ - ratio |
| Thin Snow | 3 | 0.78 | 6.03 | 39 | SI -ratio - lat |
| Coast | 13 | 0.43 | 6.80 | 44 | SI - $pem_{23}$ - $pem_{31}$ - DEM - $T_{2m}$ |

**Table 2: Classification Refinement - Parameters.**

| Predictor Set | POD | FAR | HSS |
|---|---|---|---|
| $\Delta TB_{obs-sim}$ | 0.75 | 0.29 | 0.48 |
| $TB_{obs}$ | 0.81 | 0.18 | 0.65 |
| $TB_{obs}$+environmental var | 0.82 | 0.17 | 0.68 |
| $TB_{obs}$+$\Delta TB_{obs-sim}$ | 0.84 | 0.16 | 0.69 |

**Table 3: HANDEL-ATMS SSR Detection Performance: Statistical scores for different Predictor Sets**

| | POD | FAR | HSS |
|---|---|---|---|
| SWP | 0.85 | 0.15 | 0.70 |
| SSR | 0.84 | 0.16 | 0.69 |

**Table 4: HANDEL-ATMS detection Performance - SWP and SSR Detection Modules Statistical Scores**

| | RMSE | bias | $R^2$ |
|---|---|---|---|
| SWP (kg m$^{-2}$) | 0.047 | 0.001 | 0.72 |
| SSR (mm h$^{-1}$) | 0.079 | 0.002 | 0.61 |

**Table 5: HANDEL-ATMS Estimation Performance - SWP and SSR Estimation Module Error Statistics ù**






| | SWP | | SSR | |
|---|---|---|---|---|
| PESCA Class | POD | FAR | POD | FAR |
| Ocean | 0.95 | 0.11 | 0.91 | 0.12 |
| New Sea Ice | 0.78 | 0.19 | 0.79 | 0.20 |
| Broken Sea Ice | 0.83 | 0.18 | 0.85 | 0.19 |
| Multilayer Sea Ice | 0.81 | 0.18 | 0.81 | 0.18 |
| Land | 0.76 | 0.16 | 0.79 | 0.20 |
| Perennial Snow | 0.77 | 0.21 | 0.72 | 0.21 |
| Winter Polar Snow | 0.73 | 0.22 | 0.74 | 0.23 |
| Deep Dry Snow | 0.83 | 0.16 | 0.84 | 0.17 |
| Thin Snow | 0.88 | 0.16 | 0.88 | 0.18 |
| Coast | 0.80 | 0.18 | 0.80 | 0.19 |

**Table 6: Comparison between HANDEL-ATMS detection Performances for the Different PESCA surface classesl.**

| | POD | | FAR | |
|---|---|---|---|---|
| | SLALOM-CT | HANDEL-ATMS | SLALOM-CT | HANDEL-ATMS |
| TPW<10 mm $T_{2m}$<280 K (*) | 0.82 | 0.84 | 0.19 | 0.16 |
| TPW<5 mm $T_{2m}$<250 K | 0.64 | 0.68 | 0.28 | 0.23 |
| TPW<3 mm $T_{2m}$<240 K | 0.45 | 0.54 | 0.33 | 0.28 |

**Table 7: Comparison between HANDEL-ATMS and SLALOM-CT detection Performances for Different**
**Environmental Conditions (* HANDEL-ATMS working limits).**