# Peer review of "The High lAtitude sNowfall Detection and Estimation aLgorithm"

_Atmospheric Measurement Techniques, 2023_

## Author Comment (AC1)

**Reviewer 1**

We would like to thank Reviewer #1 for his/her review of our paper and the important comments and suggestions provided. Please, find below our responses to the Reviewer's comments and the details on how we address them in the new version of the manuscript.

**1.1) Line 27: This approach has been used before so it's not accurate to call it innovative. Zhao and Weng (2002, http://www.jstor.org/stable/26184983) retrieved ice cloud parameters by isolating ice scattering signature. The latter is derived from observed high frequency TBs and simulated cloud base (i.e. clear-sky) TBs. They calculated the over land cloud base high frequency TBs from low frequencies with the assumption that low frequency measurements are less affected by cloud scattering. Please modify the manuscript accordingly and cite Zhao and Weng's paper.**

Thanks to the reviewer for the very useful suggestion. The HANDEL-ATMS approach is indeed very similar to Zhao&Weng's approach. However, it is also worth noticing some important differences:

1) the Zhao&Weng Algorithm screens out all possible "scattering surfaces" including snow cover and sea ice, that are the kind of surfaces where HANDEL-ATMS is focused on.
2) the Simulated clear-sky TB estimated by Zhao&Wheng is obtained by an empirical relationship between AMSU-A 23 and 31 GHz and 89 and 150 GHz clear-sky TB; in our work, an emissivity spectrum has been estimated for the ATMS channels downstream a background surface classification and the differences between the observed signal and the simulated one for 16 different channels have been used as input of a neural network approach

Moreover in the Abstract we stated:

*The main novelty of the approach is the radiometric characterization of the background surface (including snow covered land and sea ice) at the time of the overpass to derive multi-channel surface emissivities and clear-sky contribution to be used in the snowfall retrieval process.*

The statement in parenthesis, in our opinion, is sufficient to restrict the novelty of the approach to some background surfaces. Therefore we would like to keep the abstract as it is. However, we recognize the importance of the Zhao&Weng approach and the similarities between that work and HANDEL-ATMS and we modified the Introduction (lines 99-121):

From:

*The main novelty of the approach is the exploitation of the ATMS wide range of channels (from 22 GHz to 183 GHz) to obìtain the radiometric characterization of the background surface at the time of the overpass. The derived surface emissivities are used to infer the clear-sky contribution to the measured TBs in the high frequency channels in the snowfall retrieval process. Moreover, the algorithm is based on the exploitation of an observational dataset where each ATMS multichannel observation is associated with coincident (in time and space) CloudSat CPR vertical snow profile and surface snowfall rate (hereafter ATMS-CPR coincidence dataset). Several snowfall retrieval algorithms for cross-track scanning radiometers have evolved in the last 20 years starting from the Advanced Microwave Sounder Unit-B (AMSU-B) (Kongoli et al, 2003, Skofronick-Jackson et al, 2004, Noh et al., 2009, Liu and Seo 2013), and Microwave Humidity Sounder (MHS) (see Liu & Seo, 2013, Edel et al, 2020), and evolving to ATMS (Kongoli et al, 2015, Meng et al, 2017, Kongoli et al, 2018, You et al, 2022, Sanò et al, 2022). Some of them are based on radiative transfer simulations of observed snowfall events (Kongoli et al, 2003, Skofronick-Jackson et al, 2004, Kim et al, 2008), or on in-situ data (see Kongoli et al, 2015, Meng et al, 2017, Kongoli et al, 2018), others on CPR observations (Edel et al, 2020, You et al, 2022, Sanò et al, 2022), or a combination of them (Noh et al, 2009, Liu & Seo, 2013).*

to:

*The main novelty of the approach is the exploitation of the ATMS wide range of channels (from 22 GHz to 183 GHz) to obìtain the radiometric characterization of the background surface at the time of the overpass. The derived surface emissivities are used to infer the clear-sky contribution to the measured TBs in the high frequency*

*channels in the snowfall retrieval process. This approach is similar to the work of Zhao and Weng, 2002, for*
*AMSU observations limited to non-scattering surfaces (i.e., ocean and vegetated land), however the application*
*to surfaces with a very complex and time-varying emissivity (such as snow cover and sea ice) required a far-away*
*more advanced algorithm taking advantage of machine learning techniques. Moreover, the algorithm is based*
*on the exploitation of an observational dataset where each ATMS multichannel observation is associated with*
*coincident (in time and space) CloudSat CPR vertical snow profile and surface snowfall rate (hereafter ATMS-*
*CPR coincidence dataset).*
*Several snowfall retrieval algorithms for cross-track scanning radiometers have evolved in the last 20 years*
*starting from the Advanced Microwave Sounder Unit-B (AMSU-B) (Zhao and Weng 2002, Kongoli et al, 2003,*
*Skofronick-Jackson et al, 2004, Noh et al, 2009, Liu and Seo 2013), and Microwave Humidity Sounder (MHS)*
*(see Liu & Seo, 2013, Edel et al, 2020), and evolving to ATMS (Kongoli et al, 2015, Meng et al, 2017, Kongoli et*
*al, 2018, You et al, 2022, Sanò et al, 2022). Some of them are based on radiative transfer simulations of observed*
*snowfall events (Kongoli et al, 2003, Skofronick-Jackson et al, 2004, Kim et al, 2008), or on in-situ data (see*
*Kongoli et al, 2015, Meng et al, 2017, Kongoli et al, 2018), others on CPR observations (Edel et al, 2020, You et*
*al, 2022, Sanò et al, 2022), or a combination of them (Noh et al, 2009, Liu & Seo, 2013).*
The following reference has been added to the text (Line 810):
*Zhao, L., & Weng, F.: Retrieval of ice cloud parameters using the Advanced Microwave Sounding Unit. Journal*
*of Applied Meteorology and Climatology, 41(4), 384-395, https://www.jstor.org/stable/26184983, 2002.*
Reference:
Zhao, L., & Weng, F.: Retrieval of ice cloud parameters using the Advanced Microwave Sounding Unit. *Journal*
*of Applied Meteorology and Climatology*, *41*(4), 384-395, https://www.jstor.org/stable/26184983, 2002.
**1.2) Line 67: replace with "new" or "latest"**
Thanks to the reviewer for the suggestion. The text has been modified
From:
*the availability of the last generation microwave radiometers*
to:
*the availability of the latest generation microwave radiometers*
**1.3) Line 89: Contrary to what's stated here, Greenland and Antarctica show scattering year-round in**
**window and water vapor sounding channels, and even in the low temperature sounding channels.**
Thanks to the reviewer for the comment. Greenland and Anctartica have been defined as scatter-free by
*Grody&Basist, 1996*. For what concerns our paper, the intention was to underline the absence of a significant
difference between the emissivities at 23 GHz and at 31 GHz, typical of the snowcover over Greenland and
Antarctic plateau (see Camplani et al, 2021), without referring to higher frequencies, as opposed to deep dry snow
at lower latitudes where this difference is evident. So we agree that the term "scatter-free" can be misleading if
we also consider high-frequency channels. Therefore, the text has been changed
from:
*At the same time, large areas of Greenland and Antarctica could appear as "scatter-free", although these areas*
*throughout the year are covered by dry snowpacks.*
to:
*At the same time, large areas of Greenland and Antarctica, although these areas are covered by dry snowpacks*
*throughout the year, do not show a significant difference between the two ATMS low frequency channels.*
References:

Grody, N. C., & Basist, A. N.: Global identification of snowcover using SSM/I measurements. *IEEE Transactions*
*on geoscience and remote sensing*, *34*(1), 237-249, DOI: 10.1109/36.481908, 1996.
Camplani, A., Casella, D., Sanò, P., & Panegrossi, G.: The Passive microwave Empirical cold Surface
Classification Algorithm (PESCA): Application to GMI and ATMS. *Journal of Hydrometeorology*, *22*(7), 1727-
1744,https://doi.org/10.1175/JHM-D-20-0260.1, 2021.

**1.4) Lines 116-119: While 2CSP is a well-recognized product and is not derived from radiative transfer**
**modeling, it does include assumptions about snow microphysics, and uses optimal estimation to retrieve**
**these parameters. The algorithm also uses a simplified radar reflectivity equation. Refer to the 2CSP ATBD**
**at                 https://www.cloudsat.cira.colostate.edu/cloudsat-static/info/dl/2c-snow-profile/2C-SNOW-**
**PROFILE_PDICD.P1_R05.rev0_.pdf. Please modify the text here accordingly.**

Thanks to the reviewer for the clarification. In the text, we wanted to highlight the issues inherent in using a
dataset based on simulations (cloud-resolving model and radiative transfer) with respect to one based on
coincident observations.  The text has been changed from:
*On the other side, the use of CPR-based datasets overcomes some of the limitations deriving from the assumptions*
*to be made in cloud-radiation model simulations (e. g., the microphysics scheme, the emissivity of the background*
*surface, scattering properties of ice hydrometeors), which are particularly problematic for snowfall estimation.*
*However, some limitations of the radar product used as reference and issues related to the spatial and temporal*
*matching between the CPR and the PMW radiometer measurements introduces some uncertainty.*
to:
*On the other hand, the use of CPR-based datasets overcomes some of the limitations deriving from the use of*
*cloud-radiation model simulations, which are particularly challenging for snowfall events. However, some*
*limitations of the radar product used as a reference and issues related to the spatial and temporal matching*
*between the CPR and the PMW radiometer measurements introduce some uncertainty. Moreover, the 2CSP*
*product is based on assumptions on snow microphysics, uses optimal estimation to retrieve snow parameters ,*
*and uses a simplified radar reflectivity equation and is affected by CloudSat CPR limitations as outlined in*
*Battaglia & Panegrossi, 2020.*
Reference:
Battaglia, A., & Panegrossi, G.: What can we learn from the CloudSat radiometric mode observations of snowfall
over the ice-free ocean?. Remote Sensing, 12(20), 3285, https://doi.org/10.3390/rs12203285, 2020.

**1.5) Line 181: How is the underestimation of heavy snowfall handled in training and validating the SWP**
**and SSR models?**

Thanks to the reviewer for the question.  The aim of the algorithm is to reproduce the 2C-Snow Profile product
snowfall climatology, which is the only global radar product obtained from satellites. So, the underestimation has
not been corrected .

The following statement has been added to the text (line 223):

*Moreover, it is worth noting that CPR 2CSP product limitations for snowfall detection and estimation (see Section*
*2.2) affect the algorithm snowfall retrieval capabilities.*

**1.6) Line 273:  Do the ANNs use environmental parameters? What are they?**

Thanks to the reviewer for the question. The final version of the algorithm does not use environmental parameters
as input of the ANNs, but only some ancillary parameters (Digital Elevation Model (DEM), radiometer viewing
angle). So the text has been modified
from
*Four ANNs are then applied to a predictor set consisting of ATMS $TB_{obs}$, $\Delta TB_{obs-sim}$, a surface classification*
*flag, and other environmental and ancillary parameters.*
to:
*Four ANNs are then applied to a predictor set consisting of ATMS $TB_{obs}$, $\Delta TB_{obs-sim}$, a surface classification*
*flag, and other ancillary parameters (elevation and ATMS viewing angle for the final version).*
**1.7) Lines 191-192: Add the info on the dataset's geographic area. Was the data filtered for high latitudes**
**given the focus of this study?**
Thanks to the reviewer for the suggestion and for the question. The data have been not filtered based on a
geographic criteria. However, the data selection is based on temperature (T2m<280 K) and water vapor content
(TPW<10 mm) and on elevation (see lines 320-321 and *Camplani et al, 2021*); As a consequence, the majority of
the observations selected are obtained over high latitude areas. A statement about the dataset composition has
been added (see answer to Comment 1.21).
Reference:
Camplani, A., Casella, D., Sanò, P., & Panegrossi, G.: The Passive microwave Empirical cold Surface
Classification Algorithm (PESCA): Application to GMI and ATMS. *Journal of Hydrometeorology*, 22(7), 1727-
1744,https://doi.org/10.1175/JHM-D-20-0260.1, 2021.
**1.8) Lines 193-194: With a 15-min time window, the snow mass that ATMS detects in the atmosphere most**
**likely is higher than the near-surface snow (SSR) observed by CPR (refer to You et al., doi:**
**10.1029/2019GL083426). This adds uncertainties to the SSR (and to a lesser degree to SWP). Suggest the**
**authors run an experiment where ATMS data is collocated with CPR snowfall rate with a certain time lag**
**(30-minute?), and compare the retrieved ATMS snowfall rate with what is presented in this manuscript.**
Thanks to the reviewer for the suggestion. The suggested experiment is extremely interesting, and we want to take
it into account for future works. However, the selection of coincident observations and the making of a coincidence
dataset is a computationally and time consuming process, so we do not have the possibility to face this problem
during the revision phase. The following statement have been added to the conclusions (line 597):
*Moreover, recent studies have highlighted that TBs correlate more strongly with lagged surface precipitation*
*(with a time lag of 30-60 min for snowfall) than the simultaneous precipitation rate ( see You et al, 2019) .*
*Therefore, an analysis based on a coincident dataset characterized by different time lags will be carried out. The*
*results of this analysis will be compared with HANDEL-ATMS performances in order to identify a way to exploit*
*this information to improve SSR detection and estimation.*
The following reference has been added to the text (Line 806):
*You, Y., Meng, H., Dong, J., & Rudlosky, S.: Time-lag correlation between passive microwave measurements and*
*surface precipitation and its impact on precipitation retrieval evaluation. Geophysical Research Letters, 46(14),*
*8415-8423, doi: 10.1029/2019GL083426, 2019.*
Reference:

You, Y., Meng, H., Dong, J., & Rudlosky, S.: Time-lag correlation between passive microwave measurements and surface precipitation and its impact on precipitation retrieval evaluation. *Geophysical Research Letters*, *46*(14), 8415-8423, doi: 10.1029/2019GL083426, 2019.

**1.9) Line 282: Is there any noticeable discontinuity in the retrieved SWP and SSR between the different surface classes? Please add some discussion in the appropriate section.**

Thanks to the reviewer for the comment. As it is possible to observe by the case study reported, discontinuities in the SWP/SSR retrieval are not observed in correspondence with the surface class change. Also for other case studies analyzed it has not been observed any discontinuity in snowfall retrievals in correspondence with a surface class change. In the following plots the statistical scores (POD, FAR and HSS) are reported as a function of the class. It is possible to observe that there are not very large differences. Also the error statistics do not show any significant difference between the various surface classes (see the answer to 1.23, Figure 9). So, the following statement has been added in the section dedicated to the case study (line 525):
*Discontinuities in snowfall retrievals are not observed in correspondence with surface class changes.*

**1.10) Line 283: replace NASA with NOAA**

Thanks to the reviewer for the correction. The text has been modified from:
*the NASA AutoSnow product*
to:
*the NOAA AutoSnow product*

**1.11) Line 290: While this is outside the scope of this study, is it possible to improve snow cover classification using ML approach? I'd like to get the authors' comments on it.**

Thanks to the reviewer for the question. In *Camplani et al, 2021* a comparison between the PESCA performances and the performance obtained with a RobustBoost approach (Machine Learning ensemble method) has been carried out. The results show that the performances obtained with this ML approach are very similar to those obtained by using PESCA. However, the leading idea of PESCA is to use a simple and not too computationally demanding method to obtain a surface classification ancillary to the snowfall retrieval by exploiting the radiometer low-frequency channels. Indeed, in our opinion, the use of ML approaches for the prediction of the surface emissivity for snow cover surfaces is very promising. In particular, it could be of great benefit for the exploitation of the heterogeneous observations from the radiometer constellation. In this context, we are presently working in how the future measurements of CIMR radiometer, with an unprecedented spatial resolution, but no high frequency channels, can be exploited for improving the snowfall and IWP estimates of other radiometers equipped with high frequency channels, such as EPS-SG MWI, ICI, MWS the ATMS and AWS-STERNA. We sincerely thank the reviewer for this comment, and we would be pleased to further discuss this topic when the revision of this manuscript will be completed.

Reference:

Camplani, A., Casella, D., Sanò, P., & Panegrossi, G.: The Passive microwave Empirical cold Surface Classification Algorithm (PESCA): Application to GMI and ATMS. *Journal of Hydrometeorology*, *22*(7), 1727-1744, https://doi.org/10.1175/JHM-D-20-0260.1, 2021.

**1.12) Line 327: give explicit definitions of POD, FAR, and HSS even though they are well known.**

Thanks to the reviewer for the suggestion. The text has been modified from:

*The statistical scores (POD, FAR, HSS) of PESCA identification of sea ice and snow cover (using AutoSnow as reference) are summarized in Table 1.*

to:

*The statistical scores of PESCA identification of sea ice and snow cover (using AutoSnow as the reference) are summarized in Table 1. In particular, the Probability of Detection (POD), the False Alarm Ratio (FAR), and the Heidke Skill Score (HSS) are reported. POD, FAR, and HSS are defined by equations 2,3 and 4.*

$$POD = \frac{h}{h+m}$$

*(2)*

$$FAR = \frac{f}{f+h}$$

*(3)*

$$HSS = \frac{2(h*cn - f*m)}{(h+m)*(m+cn) + (h+f)(f+cn)}$$

*(4)*

*where h represents the hits, f represents the false alarms, m represents the misses and cn represents the correct negatives*

**1.13) Line 346: Give reference to the radiative transfer model, or add some information about the model.**

Thanks to the reviewer for the suggestion. The simulations are based on a plane-parallel approximation (see *Ulaby, 2014)* and the gas absorption model is described by *Rosenkranz, 1998*. The text has been modified (see answer to Comment 1.15).

The following reference has been added to the text (Line 806):

*Rosenkranz, P. W., Water vapor microwave continuum absorption: A comparison of measurements and models. Radio Science, 33(4), 919-928. https://doi.org/10.1029/98RS01182, 1998.*

References:

Ulaby, F., & Long, D., Microwave radar and radiometric remote sensing, 1st Edition, the Univ. of Michigan Press, ISBN: 978-0-472-11935-6, 2014.

Rosenkranz, P. W., Water vapor microwave continuum absorption: A comparison of measurements and models. *Radio Science*, 33(4), 919-928. https://doi.org/10.1029/98RS01182, 1998.

**1.14) Line 350: Is the polarization effect on emissivity also neglected between viewing angles of 40 degree and 52.7 degree (the max ATMS viewing angle)? Need to state it if it's the case.**

Thanks to the reviewer for the question. The polarization effect is less than 0.05 between 0 ° and 52.7 °, so it has not been considered. In the plot below the dependence of the ocean emissivity on viewing angle at 89 GHz (top) and the differences between the emissivity at nadir and the emissivity at a certain angle (bottom) are reported based on the FASTEM model (see *Prigent et al, 2017*) . It is possible to observe that, while the V and H emissivity show a variation up to 0.15, the QV and QH emissivity variation is lower than 0.05 for scan angles < 52 °.

The text has been modified from:

*The emissivity spectra dependence on the ATMS viewing angle for polarized surfaces has been neglected because an analysis of such dependence in the ATMS-CPR coincidence dataset has shown that it is significant only for larger viewing angles (tot for >40 °). This is due to the fact that cross-track scanning radiometers measure a signal (off-nadir) which derives from a mixture between the two polarizations (e.g., quasi-vertical, QV, and quasi-horizontal, QH). As a consequence, although the emissivities of polarized surfaces, such as open water surfaces, are strongly influenced by the viewing angle, for the cross-track scanning radiometers the emissivity variation is compensated by the effect of the mixture of the two polarization (see also Felde & Pickle, 1995, Prigent et al 2000, Mathew et al, 2008, Prigent et al, 2017).*

to:

*The emissivity spectra dependence on the ATMS viewing angle for polarized surfaces has been neglected because*
*an analysis of such dependence in the ATMS-CPR coincidence dataset has shown that it is not significant for*
*ATMS viewing angles (emissivity difference smaller than 0.05 for angles up to 52.7 °). This is due to the fact that*
*cross-track scanning radiometers measure a signal (off-nadir) which derives from a mixture between the two*
*polarizations (e.g., quasi-vertical, QV, and quasi-horizontal, QH). As a consequence, although the emissivities of*
*polarized surfaces, such as open water surfaces, are strongly influenced by the viewing angle, for the cross-track*
*scanning radiometers the emissivity variation is compensated by the effect of the mixture of the two polarization*
*(see also Felde & Pickle, 1995, Prigent et al, 2000, Mathew et al, 2008, Prigent et al, 2017).*

[Figure]

[Figure]

Reference:
Prigent, C., Aires, F., Wang, D., Fox, S., & Harlow, C.: Sea-surface emissivity parametrization from microwaves to millimetre waves. *Quarterly Journal of the Royal Meteorological Society*, *143*(702), 596-605, https://doi.org/10.1002/qj.2953, 2017.

**1.15) Line 362: Reference for the RTM?**

Thanks to the reviewer for the suggestion. The text has been modified from:
*The RMSE between simulated clear-sky TBs - based on the mean emissivity values estimated for each class - and the coincident observed clear-sky TBs appears to be too high to implement a robust signal analysis (>10 K).*
to:
*The clear-sky radiative transfer model simulations are based on the mean emissivity values estimated for each class, and simulated by using the plane-parallel approximation (Ulaby & Long, 2014) and the Rosenkrantz gas absorption model (Rosenkrantz, 1998) - The RMSE between simulated clear-sky TBs and the coincident observed clear-sky TBs appears to be too high to implement a robust signal analysis (>10 K).*

References:

Rosenkranz, P. W., Water vapor microwave continuum absorption: A comparison of measurements and models. *Radio Science*, *33*(4), 919-928. https://doi.org/10.1029/98RS01182, 1998.

Ulaby, F., & Long, D., Microwave radar and radiometric remote sensing, 1st Edition, the Univ. of Michigan Press, ISBN: 978-0-472-11935-6, 2014.

**1.16) Line 397, the RMSE for ocean is 3.37 K in Table 2.**

Thanks to the reviewer for the observation. The text has been modified from:
very *low RMSE values (≈ 2 K)*
to:
*low RMSE values (< 4 K)*

**1.17) Line 403: Since high frequencies are more important for snowfall retrieval, need to discuss the impact of the significant uncertainties at these channels to retrieve SWP and SSR.**

Thanks to the reviewer for the suggestion. In Figure 9 (see answer to Comment 1.23) the statistical scores for each PESCA class are reported. It is possible to observe that the worst scores are obtained for classes characterized by high uncertainties in the clear-sky TB simulations (Perennial Snow, Winter Polar Snow). However, it is also worth noting that these classes are mostly associated with environmental conditions (very dry and cold, with very light snowfall events, see *Camplani et al, 2021*) which make it difficult both to obtain a more accurate clear emissivity estimation and to retrieve snowfall. At the same time, it can be observed that classes characterized by the highest uncertainties on the emissivity estimate (Deep Dry Snow and Broken Sea Ice), show statistical scores which are coherent with the general scores of the algorithm. So it is clear that the uncertainties on emissivity estimation have less influence than other factors, such as the environmental conditions.

The text has been modified (line 471)

from:
*In Table 6 the statistical scores of the algorithm performance by considering each PESCA class for both the SWP and the SSR detection module are reported. It can be observed that, also considering specifically the classes where the detection is more problematic, both for the uncertainties linked to the emissivity retrieval (see Table 2), for the extremely dry and cold environmental conditions, and for the low intensity of the snowfall events, such as Perennial Snow or Winter Polar Snow, HANDEL-ATMS has good detection capabilities (POD and FAR values greater than 0.7 and less than 0.25, respectively, for both SWP and SSR). These results provide evidence that HANDEL-ATMS can be used to analyze snowfall occurrence in the polar regions.*
to:

*In Figure 9 the statistical scores of the algorithm performance by considering each PESCA class for both the*
*SWP and the SSR detection module are reported. It can be observed that, also considering specifically the classes*
*associated to extremely dry and cold environmental conditions such as Perennial Snow or Winter Polar Snow*
*(see Camplani et al, 2021) (where the detection is more problematic due to the uncertainties in the emissivity*
*retrieval (see Table 2) , and to the low snowfall intensity), , HANDEL-ATMS has good detection capabilities (POD*
*and FAR values greater than 0.7 and less than 0.25, respectively, for both SWP and SSR). On the other hand, it*
*is possible to observe also that for surface classes characterized by the highest emission estimation uncertainties,*
*such as Deep Dry Snow, the statistical scores are coherent with the general scores and better than those obtained*
*in presence of extremely dry/cold environmental conditions. So, it is possible to conclude that the extremely*
*cold/dry environmental conditions - have more influence on the detection than the uncertainties on clear sky*
*emissivity estimation. Generally, these results provide evidence that HANDEL-ATMS can be used to analyze*
*snowfall occurrence in the polar regions.*

Reference:

Camplani, A., Casella, D., Sanò, P., & Panegrossi, G.: The Passive microwave Empirical cold Surface
Classification Algorithm (PESCA): Application to GMI and ATMS. *Journal of Hydrometeorology*, 22(7), 1727-
1744,https://doi.org/10.1175/JHM-D-20-0260.1, 2021.

**1.18) Line 430: Logarithmic tangent function is not a common activation function. Please add a reference**
**or explain what it is.**

Thanks to the reviewer for this comment. It was  a typo, the activation function is a sigmoid.  We used hyperbolic
tangent and sigmoid functions, which are indeed very common activation functions. The choice of the activation
functions has been performed by trial and testing.

The manuscript has been modified from:
*The final architecture, for all modules, is composed of four layers: an input layer with a neurons number equal*
*to the predictor number, and a hyperbolic tangent function as the activation function, a first hidden layer (60*
*neurons), and hyperbolic tangent function, a second hidden layer (30 neurons), with a logarithmic tangent*
*function.*
to:
*The final architecture, for all modules, is composed of four layers: an input layer with a neurons number equal*
*to the predictor number, and a hyperbolic tangent function as the activation function, a first hidden layer (60*
*neurons), and hyperbolic tangent function, a second hidden layer (30 neurons), with a sigmoid function.*

**1.19) Lines 435-436: Did the predictor set including TB_obs, TB_obs-TB_sim, and environmental variables**
**give better result than the set only included the first two? If not, why? Is it because TB_sim also used the**
**environmental variables being tested?**
Thanks to the reviewer for the question. The NNs that use both the $\Delta_{obs-sim}$ and the environmental parameters show
detection scores almost equal to those obtained by using only   $\Delta_{obs-sim}$. This is because the information about
environmental conditions is already used as input in the clear-sky TB simulations The following statement has
been added to the text (line 438):
*On the contrary, the simultaneous use of both the $\Delta TB_{obs-sim}$ and the environmental parameters show scores almost*
*equal to that obtained by using only $\Delta TB_{obs-sim}$ .*

**1.20) Lines 444: Which 16 ATMS channels and how are they selected?**

Thanks to the reviewer for the suggestion.  The sixteen channels are ATMS channels 1-9, 16-22. The ATMS 10-
15 channels peak above the tropopause, so we did not take them into account in the development of HANDEL-

ATMS. Figure below shows the temperature weighting functions for a standard atmosphere in clear sky
conditions.

[Figure]

The text has been modified from:

*16 ATMS TB_{obs}*

to:

*1-9, 16-22 ATMS channels TB_{obs} (the 10-15 ATMS channels have not been considered because their weighting*
*function peaks above the tropopause).*

**1.21) Section 4.1: Some details about the validation data should be provided. Is the data from selected**
**snowfall events used or from a time period? How many events were included and their geographic areas?**
**How many data points were in the dataset etc.? The information is important because it provides the**
**context for the performance metrics.**

Thanks to the reviewer for the suggestion. The following section has been added to the text of section 2.3 (line
223):

*In this work, the dataset has been filtered based on humidity (TPW < 10 mm) and temperature (T_{2m} <280 K) and*
*elevation conditions (the working limits of the PESCA algorithm, see Camplani et al, 2021) leading to a good*
*representation of the higher latitudes with 80 % of the dataset elements located above 60°N/S . The dataset is*
*made of 2,14\*10 [6] elements, including 1,07\*10 [6] elements with falling snow (2CSP SWP > 0 kg m^{-2}) and 9,99\*10*
*[5] with snowfall at the surface (2CSP SSR > 0 mm h^{-1}) . The training and test phases have been conducted by*
*splitting randomly the dataset, with ⅓ of the elements in the training and ⅔ of the elements in the test dataset.*

Reference:

Camplani, A., Casella, D., Sanò, P., & Panegrossi, G.: The Passive microwave Empirical cold Surface Classification Algorithm (PESCA): Application to GMI and ATMS. *Journal of Hydrometeorology*, *22*(7), 1727-1744,https://doi.org/10.1175/JHM-D-20-0260.1, 2021.

**1.22) Line 451: A large percentage of the snowfall appears to fall when T_2m is around the freezing point or higher. Snowfall under such conditions generally has different characteristics from snowfall in high latitudes which is the focus of this study. Add some discussion about the data distribution and its impact on the new snowfall algorithm.**

Thanks to the reviewer for the suggestion. Generally, the SWP detection shows better performances in moister and warmer conditions than in colder/drier situations for two main reasons: 1) the atmosphere is less transparent 2) these conditions are usually associated with more intense events. However, in these conditions there can be a mismatch between the presence of falling snow in the atmosphere and the presence of snowfall at the surface; therefore, the SSR detection statistical scores show a maximum around 273 K and 5 mm and then decrease. From Figure 8, it is possible to observe that the maximum number of observations and of snowfall elements in the dataset is around 273 K, where the best performances are obtained. However, it is worth noticing that HANDEL shows very good results also in very dry and very cold conditions. We believe that this is the main achievement of this work, since the main objective of this study is to show that HANDEL is able to detect and retrieve snow also in extreme conditions typical of the higher latitudes. We think that this is the added value of this study. In order to highlight this aspect, we have added a new figure showing the variability of the estimation statistical scores and the mean SWP and SSR with TPW (see answer to Comment 1.25).

**1.23) Line 471: Add HSS to Table 6.**

Thanks to the reviewer for the suggestion.
We have deleted Table 6 and we have added Figure 9, where the POD, FAR, HSS, the observation occurrences and the snowfall observation occurrences (SWP, SSR>0) are reported.

[Figure]

               **Figure 9: Same as Figure 7 but for PESCA surface classes.**

**1.24) Table 5: Since the goal of this study is to retrieve snowfall in high latitude, it'd be informative to**
**analyze how well the statistics represent the cold, dry and light snowfall versus the warm, moist, and heavier**
**snowfall. Please add some quantitative analysis to show the performance of the snowfall representative of**
**high latitude conditions.**

Thanks to the reviewer for the suggestion. The dependence of the detection scores on the environmental conditions
has been reported in Figure 7 and in Figure 8. The presence of a less transparent atmosphere and the presence of
high SWP values generates a more intense signal. We have decided to add one Figure in the manuscript showing
the variability of the snowfall estimation statistical scores, as well as SWP and SSR, with TPW (see answer to
Comment 1.25).

**1.25) Line 487: Typically, high latitude snowfall is rather light. Does this result mean that the snowfall**
**retrieval in high latitude is generally overestimated? Add some discussion here.**

Thanks to the reviewer for the comment. From Figure 9 it is possible to observe that the algorithm tends to
overestimate light snowfall, while there is a better agreement for more intense snowfall. Very light snowfall events
are linked to the dry /cold environmental conditions typical of high latitude areas, where more intense snowfall
events are typical of moister conditions. We state that "Generally, it can be observed that, although HANDEL-
ATMS is able to detect extremely light snowfall events, it does not have the sensitivity to correctly estimate their
intensity." The final part of Section 4.1 has been largely modified (see below)

We decided to add the following Figure to the paper in order to answer 1.22, 1.24 and 1.25.

[Figure]

*Figure 11: HANDEL-ATMS SWP and SSR Detection Performances for different bins of TPW. The left y-axis reports RMSE absolute values and the mean intensity value for each 1-mm TPW bin, while the relative bias, calculated as the ratio between the bias and the SWP/SSR mean value for each bin.*

The text has been modified to comment the Figure 11 (Line 488)

from:

*. Generally, it can be observed that, although HANDEL-ATMS is able to detect extremely light snowfall events, it does not have the sensitivity to correctly estimate their intensity.*

to:

*Figure 11 shows the dependence of HANDEL-ATMS snowfall estimation error statistics, as well of SWP and SSR, on TPW. The curves represent the mean SWP or SSR computed for each 1-mm TPW bin, the RMSE and the relative bias (the ratio between the bias and the SWP/SSR mean value for each bin). TPW and snowfall intensity are strongly correlated. An increase of the absolute RMSE can be observed as TPW increases, and it is larger than the SWP/SSR mean value for TPW < 8 mm. A similar behavior can be observed by analyzing the dependence of*

*HANDEL-ATMS snowfall estimation error statistics on $T_{2m}$ (not shown).   A very moderate overestimation is*
*observed for TPW < 8 mm and for lower SWP and SSR values (< 0.1 mm/h), with relative bias around 5%, (up*
*to 8% only for extremely low TPW values and very low number of observations (see Figure 7)), while*
*underestimation (relative bias up to -5%)  is observed for higher TPW values and higher SWP and SSR  values.*
*Generally, light snowfall events are linked to the very cold/dry environmental conditions typical of high-latitude*
*regions. So, the algorithm manages to detect also the very light snowfall typical of high latitudes, but tends to*
*slightly overestimate snowfall intensity in such conditions.  It can be concluded that HANDEL-ATMS has good*
*detection capabilities (also for extremely light snowfall) but it shows some limitations in  correctly estimating its*
*intensity, with slight overestimation of the very light snowfall typical of high latitudes.*

**1.26) Lines 555-558: See the comment on line 27.**

Thanks to the reviewer for the suggestion. The text has been modified from:

*The  driving and innovative principle in the algorithm development is the exploitation of the full range of ATMS*
*channel frequencies to characterize the frozen background surface radiative properties at the time of the overpass*
*to be able to better isolate and interpret the snowfall-related contribution to the measured multi-channel upwelling*
*radiation.*
to

*The driving and innovative principle in the algorithm development is the exploitation of the full range of ATMS*
*channel frequencies to characterize the frozen background surface radiative properties at the time of the overpass*
*to be able to better isolate and interpret the snowfall-related contribution to the measured multi-channel upwelling*
*radiation. A similar approach has been used by Zhao &Weng, 2002; however, their application was limited to*
*non-scattering surfaces and was based on empirical relationships.*

Reference:

Zhao, L., & Weng, F.: Retrieval of ice cloud parameters using the Advanced Microwave Sounding Unit. *Journal*
*of Applied Meteorology and Climatology*, *41*(4), 384-395, https://www.jstor.org/stable/26184983, 2002.

---

## Author Comment (AC2)

Reviewer 2

We would like to thank Reviewer #2 for his/her review of our paper and the important comments and suggestions
provided. Please, find below our responses to the Reviewer's comments and the details on how we address them
in the new version of the manuscript

**General comments.**

**The text is a bit hard to follow. It is highly recommended that the authors make an effort to shorten it and**
**make the language and the message more succinct. The quality of the figures can be significantly improved**
**as well. There are a few important points that need to be cleared in the next revision.**

Thanks to the reviewer for the suggestion. We have shortened the manuscript and tried to make the message more
succinct. We have also improved figures 2, 6, 7, 8, 11, and 14 (now Figures 13 and 16  because  new Figures 9
and 11 have been added to address some comments by Reviewer 1)  and the captions have been modified
accordingly.

Figure 2:

[Figure]

The caption has been changed from:

*Figure 2: Sea Ice Detection: 23 TB-$T_{2m}$ Plan. The color represents the mean AutoSnow sea ice percentage within each bin*
*(left) and the observation occurrence (right).*
to
*Figure 2: Sea Ice Detection: 23 TB-$T_{2m}$ Plan. The color represents the mean AutoSnow sea ice percentage within each bin*
*(left) and the observation occurrence (right). The green (left) and red (right) lines represent the discriminant Equation*
*between sea ice and ocean.*

For Figure 6, see answer to Comment 2.20.

Figure 7:

[Figure]

The caption has been changed from:

*Figure 7 HANDEL-ATMS SWP and SSR Detection Performances for different bins of TPW. The left y-axis*
*reports POD, FAR and HSS vales, while the right y-axis reports the total number and snowfall observations in*
*the dataset. POD-tot, FAR-tot and HSS-tot (dotted lines) represent the statistical scores estimated on the total*
*dataset (values reported in Table 2).*
to
*Figure 7: HANDEL-ATMS SWP and SSR Detection Performances for different bins of TPW. The left y-axis*
*reports POD, FAR and HSS vales, while the right y-axis reports the total number and snowfall observations in*
*the  dataset.*

Figure 8:

[Figure]

The caption has not been changed

[Figure]

The caption has been changed from:

*Figure 11: Greenland - 2016/04/24 - Synopsis along CloudSat Track. The first panel shows the ECMWF TPW and T$_{2m}$ values along the CloudSat track. In the second panel, the 2CSP SWP (left) and the SSR (right) values are reported, besides the PESCA classification along CloudSat track. In the third panel, the CPR reflectivity (values are reported in the colorbar below), the supercooled water droplets detected by DARDAR (magenta points) are shown. Also the Digital Elevation Model (brown line) and the ECMWF Freezing Level (red line) along CloudSat track are reported. In the bottom panel the observed TBs of the main high-frequency channels (88 GHz, 166 GHz, 183+3 GHz, 183+7 GHz) along CloudSat track are shown.*

to

*Figure 13: Greenland - 2016/04/24 - Synopsis along CloudSat Track. The first panel shows the ECMWF TPW and T$_{2m}$ values along the CloudSat track. In the second panel, the 2CSP SWP (left) and the SSR (right) values are reported, besides the PESCA classification along CloudSat track. In the third panel, the CPR reflectivity (values are reported in the colorbar on the right), the supercooled water droplets detected by DARDAR (magenta points) are shown. Also the Digital Elevation Model (brown line) and the ECMWF Freezing Level (red line) along CloudSat track are reported. In the bottom panel the observed TBs of the main high-frequency channels (88 GHz, 166 GHz, 183+3 GHz, 183+7 GHz) along CloudSat track are shown.*

Figure 14/16:

[Figure]

________________________________________________________________

The caption has not been changed

For the new Figures 9 and 11, see answers to Comments 2.5 and 2.18.

**2.1) The explanation of the inverse radiative transfer modeling is missing. Such an inversion can be**
**significantly underconstrained and add additional uncertainty to the results.**

Thanks to the reviewer for the comment. The model used is a plane-parallel approximation (see *Ulaby&Long,*
*2014*); the gas absorption model is that described by *Rosenkranz, 1998*. In particular, the emissivity has been
calculated by inverting the radiative transfer equation

$$TB = T_{up} + (1 - \varepsilon) * T_{down} * e^{-\tau} + \varepsilon * T_{skin} * e^{-\tau}$$

to

$$\varepsilon = \frac{TB - T_{up} - T_{down} * e^{-\tau}}{e^{-\tau} * (T_{skin} - T_{down})}$$

where $T_{up}$ represents atmospheric upward emission, $T_{down}$ represents the atmospheric downward emission, $\tau$
represents the atmospheric optical thickness, $\varepsilon$ represents the emissivity, $T_{skin}$ represents the skin temperature and
TB the ATMS observed TB. $T_{up}$, $T_{down}$, and $\tau$ are obtained by applying the Rosenkranz model using ECMWF-
AUX temperature and water vapour profiles, $T_{skin}$ is obtained from ECMWF-AUX product.
References:

Rosenkranz, P. W., Water vapor microwave continuum absorption: A comparison of measurements and models.
*Radio Science*, *33*(4), 919-928. https://doi.org/10.1029/98RS01182, 1998.

Ulaby, F., & Long, D., Microwave radar and radiometric remote sensing, 1st Edition, the Univ. of Michigan Press,
ISBN: 978-0-472-11935-6, 2014.

**2.2) Please clarify upfront whether the estimated values of surface emissivities are used dynamically or**
**statistically in the algorithm. Do they change in time or not?**

Thanks to the reviewer for the comment. The emissivity values are retrieved for each pixel using the low-frequency TBs and environmental parameters at the time of the overpass; therefore, the emissivities are used dynamically. So the text has been changed:

Line 27:

from:

*Moreover, their wide range of channel frequencies (from 23 GHz to 190 GHz), allows for the radiometric characterization of the surface at the time of the overpass along with the exploitation of the high-frequency channels for snowfall retrieval.*
to:
*Moreover, their wide range of channel frequencies (from 23 GHz to 190 GHz), allows for the dynamic radiometric characterization of the surface at the time of the overpass along with the exploitation of the high-frequency channels for snowfall retrieval.*

Line 136:
from:
*The present work has the aim to develop an algorithm for snowfall detection and estimation by exploiting the large frequency range typical of the last generation radiometers and to obtain a radiometric characterization of the background surface at the time of the satellite overpass in order to highlight the complex relationship between upwelling radiation and snowfall signature, which makes the detection very difficult in the typical conditions of the high latitudes.*
to:
*The present work has the aim to develop an algorithm for snowfall detection and estimation by exploiting the large frequency range typical of the last generation radiometers and to obtain a dynamic radiometric characterization of the background surface at the time of the satellite overpass in order to highlight the complex relationship between upwelling radiation and snowfall signature, which makes the detection very difficult in the typical conditions of the high latitudes.*

**2.3) It will be helpful if the authors clarify why we need land surface classification for the algorithm. For example, there are multiple products for the detection of the presence of snow and sea ice dynamics using optical bands (every 30 minutes). These optical products can be more accurate than microwave classification schemes, in terms of the presence or absence of frozen surfaces. Why we should not use them?**

Thanks to the reviewer for the question. There are indeed multiple products for snow-cover and sea ice detection. However, PESCA aim is to obtain information ancillary to the snowfall retrieval at the time of the overpass, by exploiting the same instruments and the same type of data which will be used downstream for snowfall retrieval (see *Camplani et al, 2021*). We are more interested in the emissivity spectrum in the microwave than in very accurate and high-resolution snow and sea ice detection. Moreover, products based on optical observations are unreliable in presence of clouds, while our goal is to use them to retrieve cloud properties. To our knowledge, the only product available every 30 min comes from geostationary satellites that show several limitations in observing high latitudes.

References:
Camplani, A., Casella, D., Sanò, P., & Panegrossi, G.: The Passive microwave Empirical cold Surface Classification Algorithm (PESCA): Application to GMI and ATMS. *Journal of Hydrometeorology*, 22(7), 1727-1744,https://doi.org/10.1175/JHM-D-20-0260.1, 2021.

**2.4) From a methodological standpoint, the explanations of neural networks need to be improved. A the same time, the use of linear discriminant analysis seems outdated in light of the new deep-learning classification models.**

Thanks to the reviewer for the comment. We know that deep-learning classification models are more effective
than models based on other machine learning approaches, such as linear discriminant analysis. However, our goal
was to obtain a classification scheme preliminary to the snowfall retrieval modules, and so we have chosen to use
methods which are simple and not too computationally and time consuming.

**2.5) While the paper focuses on different land surface types and sea ice ages, it is unclear how statistically**
**significant the presented results are in Table 7. The number of training and testing samples needs to be**
**clarified.**

Thanks to the reviewer for the suggestion. We believe that the reviewer is referring to Table 6. We have replaced
it with Figure 9. In the two plots the statistical scores for each class, the total observation number and the snowfall
observation number for the test phase are reported. For what concerns the number of training and testing samples,
see answer to Comment 2.6.

[Figure]

**Figure 9: Same as Figure 7 but for  PESCA surface classes.**

The violet continuous and dashed line represents the total class occurrence and the snowfall occurrence for each
class respectively. So, it is possible to observe that also the less populated classes, such as Thin Snow, are
characterized by about $3*10^4$ total observations and $1*10^4$ snowfall observations. So the statistics can be
considered statistically significant. This Figure has been added to the manuscript.

**2.6) It would benefit the paper if the authors provide the entire confusion matrix of the detection of snowfall,**
**including, recall, precision, and accuracy.**

Thanks to the reviewer for the suggestion. Here the confusion matrices and the precision, recall and accuracy
values are reported.

| SWP detection - Confusion Matrix | | |
|---|---|---|
| **HANDEL/2CSP** | **YES** | **NO** |
| **YES** | **606711** | **106407** |
| **NO** | **106541** | **581671** |

precision=0.85

recall=0.85

accuracy=0,84

| SSR detection - Confusion Matrix | | |
|---|---|---|
| **HANDEL/2CSP** | **YES** | **NO** |
| **YES** | **541688** | **102542** |
| **NO** | **113615** | **643485** |

precision=0.82

recall=0.84

accuracy=0,84

The total number of observations is $1,40*10^6$, which corresponds to about ⅔ of the total observations number. A
similar proportion can be observed for the SWP and SSR observations. The following statement has been added
to the text (line 223):

*In this work, the dataset has been filtered based on humidity (TPW < 10 mm) and temperature ($T_{2m}$ <280 K)*
*conditions (the working limits of the PESCA algorithm, see Camplani et al, 2021)  leading to a good*
*representation of the higher latitudes with 80 % of the dataset elements located above 60°N/S. are . The dataset*
*is made of $2,14*10^6$ elements, including $1,07*10^6$ elements with falling snow (2CSP SWP > kg/m2) and $9,99*10$*
*$^5$ with snowfall at the surface (2CSP SSR > 0 mm/h) . The training and test phases have been conducted by splitting*
*randomly the dataset, with ⅓ of the elements in the training and ⅔ of the elements in the test dataset.*

Therefore, data about the dataset dimension, the training and test phase and the snowfall have been added to the
text. We would prefer not to add the confusion matrices to the text in order to avoid further lengthening the
manuscript. We think that the information about the dataset, joined with the statistical scores, shows   a comprehensive picture of the study. At the same time, the recall gives the same information of POD, and precision can be considered the complementary value to 1 of the FAR. The information linked to the accuracy can be misleading: so we would prefer to keep in the text only the information about POD, FAR and HSS.

**Detail comments:**

**2.7) Section 2.4 is long and has some generic explanations about for example neutral networks, which is not necessary at this time. It is recommended to shorten the text.**

Agreed. The text has been shortened (see answer to Comment 2.8).

**2.8) The explanation of the architecture of the neural network is weak. First of all the networks use the Levenberg-Marquardt algorithm which is extremely old and is not being used in modern training of deep neural networks. Unlike algorithms like Adam, it is prone to get stuck in local minima and suffer from the vanishing gradient problem.**

We agree with the reviewer that the LM algorithm is outdated and it is not being used in deep neural network training. Our point here is that our networks are shallow, as written in section 3.2 of the manuscript:

*The snowfall detection and estimation modules have been based on ANNs. Four ANNs have been developed: two for the detection of SWP and SSR and two for the SWP and SSR estimate. The performance of more than 50 architectures have been tested, by varying the number of layers, the number of neurons for each layer, and the activation functions. The final architecture, for all modules, is composed of four layers: an input layer with a neurons number equal to the predictor number, and a hyperbolic tangent function as the activation function, a first hidden layer (60 neurons), and hyperbolic tangent function, a second hidden layer (30 neurons), with a logarithmic tangent function.*

Therefore, the neural networks described in this paper are composed of less than 150 weights. These networks fall into the category of feed forward, or multilayer perceptron networks, or shallow neural networks. The LM optimizer is prone to several issues when the depth of the network grows (i.e. if the number of weights to be trained is higher than about 500, see *Yu & Wilamowski, 2018*), such as gradient vanishing, however it has been proven to be a very accurate optimizer for shallow neural networks. The use of the LM optimizer forces the choice of the error function, that needs to be the mean squared error, in regression problems, and may result slower than other optimizers, however it has proven to reach higher accuracy in many problems, even in very recent papers, in particular we followed the *Hagan&Menhaj, 1994* implementation of the LM algorithm that has been cited in about 700 papers after 2022 (see the google scholar link to recent citation of this paper). Moreover, we did test the impact of the choice of the optimizer for one of the neural networks module of the HANDEL-ATMS algorithm, and the results confirmed the use of the LM optimizer as an optimal choice for the complexity of the networks that we are training and for the size of the dataset that we are using. In particular the LM optimizer resulted to be more accurate but slower than other optimizers (including the Conjugate-gradient, gradient descend with momentum and Adam optimizers).

About the first point raised by the reviewer "The explanation of the architecture of the neural network is weak", we believe that He/She is referring to section 2.4.1, that was intended as a brief introduction, and that has been modified from:

*2.4.1 Artificial Neural Networks*

*An Artificial Neural Network (ANN) is an information-processing system inspired by the functioning of biological neural networks. It is composed of neurons, i. e., elements where the information is processed using an activation function, and the connecting links between the neurons, where a weight multiplies the deriving from the upstream signal. In particular, the HANDEL-ATMS snowfall detection and estimation modules have been developed using feedforward multilayer neural network architectures, i. e., a neural network architecture where the neurons are arranged in layers; each neuron belonging to a layer receives, as input to its transfer function, a weighted sum of the outputs of the previous layer. This architecture, which is defined by the number of layers, the number of neurons for each layer, and the transfer function of each neuron, has to be designed beforehand. The weights of connection links and the bias values for each layer are estimated with a training process, based on the Levenberg–Marquardt algorithm (see Sanò et al, 2015)*

to:

*2.4.1 Artificial Neural Networks*

*The HANDEL-ATMS snowfall detection and estimation modules have been developed using feedforward multilayer neural network architectures, i. e., a neural network architecture where the neurons are arranged in layers. This architecture, which is defined by the number of layers, the number of neurons for each layer, and the transfer function of each neuron, has to be designed beforehand. The weights of connection links and the bias values for each layer are estimated with a training process, based on the Levenberg–Marquardt algorithm (see Sanò et al, 2015). The specific networks architecture, and the training and optimization procedure of the HANDEL-ATMS algorithm are described in detail in section 3.2.*

References:

Yu, H., & Wilamowski, B. M.: Levenberg–marquardt training. In *Intelligent systems* (pp. 12-1), CRC Press, ISBN 9781315218427, 2018.

Hagan, M. T., & Menhaj, M. B.: Training feedforward networks with the Marquardt algorithm, *IEEE transactions on Neural Networks*, *5*(6), 989-993, DOI: 10.1109/72.329697, 1994.

**2.9) Line 424–445 It is unclear how the detection and estimation networks are implemented. What are the cost functions? This must be clarified.**

Thanks to the reviewer for the suggestion. The cost function is a sum of squares error (SSE) given by the following equation:

$$E = \frac{1}{n}\sum_{i=1}^{n} (y_i - t_i)^2$$

where y represents the output of the neural networks, and t represents the reference truth value. The characteristics of this Neural network approach have been largely described by *Sanò et al, 2015*, doi:10.5194/amt-8-837-2015). So, a reference to this paper has been added (line 431):
*(for more information about the Neural Network characteristics, see Sanò et al, 2015)*

References:

Sanò, P., Panegrossi, G., Casella, D., Di Paola, F., Milani, L., Mugnai, A., Petracca, M., & Dietrich, S.: The Passive microwave Neural network Precipitation Retrieval (PNPR) algorithm for AMSU/MHS observations: description and application to European case studies. *Atmospheric Measurement Techniques*, *8*(2), 837-857, https://doi.org/10.5194/amt-8-837-2015, 2015.

**2.10) Line 345-346: It is not well-described how the inverse radiative transfer model is used. What is the forward RT model?**

Thanks to the reviewer for the question. The simulations are based on a plane-parallel approximation (see *Ulaby,*
*2014)* and the gas absorption model is described by *Rosenkranz, 1998*. The text has been modified (see answer to
Comment 1.15).

The text has been modified from:
*The RMSE between simulated clear-sky TBs - based on the mean emissivity values estimated for each class - and*
*the coincident observed clear-sky TBs appears to be too high to implement a robust signal analysis (>10 K).*
to:
*The clear-sky radiative transfer model simulations are based on the mean emissivity values estimated for each*
*class, and simulated by using the plane-parallel approximation (Ulaby & Long, 2014) and the Rosenkrantz gas*
*absorption model (Rosenkrantz, 1998) - The RMSE between simulated clear-sky TBs and the coincident observed*
*clear-sky TBs appears to be too high to implement a robust signal analysis (>10 K).*

The following reference has been added to the text (Line 756):

*Rosenkranz, P. W., Water vapor microwave continuum absorption: A comparison of measurements and models.*
*Radio Science, 33(4), 919-928. https://doi.org/10.1029/98RS01182, 1998.*

References:

Rosenkranz, P. W., Water vapor microwave continuum absorption: A comparison of measurements and models.
*Radio Science*, *33*(4), 919-928. https://doi.org/10.1029/98RS01182, 1998.

Ulaby, F., & Long, D., Microwave radar and radiometric remote sensing, 1st Edition, the Univ. of Michigan Press,
ISBN: 978-0-472-11935-6, 2014.

**2.11) Lines 362-365: How emissivity is used for calculating the simulated TBs? It seems recursive to use the**
**observations to estimate the emissivity and then use it for retrievals. Please clarify whether the used**
**emissivities are dynamic or static.**

Thanks to the reviewer for the comment. The emissivity values are retrieved for each pixel and are used to estimate
the simulated TBs. Only low-frequency channels are used to classify the observations (by using PESCA) and to
retrieve an emissivity spectrum for the observations. Then, this spectrum has been used to estimate the TBs for
all ATMS channels. So the process is not recursive. The emissivities are used dynamically because they have been
calculated for each observation (see answer to Comment 2.2).

**2.12) Table 3: The parameters mentioned in the table are different than the ones mentioned in the text in**
**lines 435-437.**

Thanks to the reviewer for the comment. The Table has been changed:

from:

| Predictor Set | POD | FAR | HSS |
|---|---|---|---|
| $\Delta TB_{obs-sim}$ | 0.75 | 0.29 | 0.48 |
| $TB_{obs}$ | 0.81 | 0.18 | 0.65 |
| $TB_{obs}$+environmental var | 0.82 | 0.17 | 0.68 |
| $TB_{obs}+\Delta TB_{obs-sim}$ | 0.84 | 0.16 | 0.69 |

**Table 3: HANDEL-ATMS SSR Detection Performance: Statistical scores for different Predictor Sets**

to:

| Predictor Set | POD | FAR | HSS |
|---|---|---|---|
| $\Delta TB_{obs-sim}$+ ancillary parameters | 0.75 | 0.29 | 0.48 |
| $TB_{obs}$+ ancillary parameters | 0.81 | 0.18 | 0.65 |
| $TB_{obs}$+environmental variables+ ancillary parameters | 0.82 | 0.17 | 0.68 |
| $TB_{obs}+\Delta TB_{obs-sim}$+ ancillary parameters | 0.84 | 0.16 | 0.69 |

**Table 3: HANDEL-ATMS SSR Detection Performance: Statistical scores for different Predictor Sets**

Minor comments:

**2.13) Line 273: It is better to mention all the variables that have been used for training the network here.**

Thanks to the reviewer for the suggestion. The text has been changed
from:
*Four ANNs are then applied to a predictor set consisting of ATMS $TB_{obs}$, $\Delta TB_{obs-sim}$, a surface classification*
*flag, and other environmental and ancillary parameters.*
to:
*Four ANNs are then applied to a predictor set consisting of ATMS $TB_{obs}$, $\Delta TB_{obs-sim}$, a surface classification*
*flag, and other ancillary parameters (elevation and ATMS viewing angle for the final version).*

**2.14) line 203-204: list of environmental and ancillary parameters is not presented in the dataset.**

Thanks to the reviewer for the comment. The text has been changed
from:
*Some model-derived variables have been added to the dataset to be used as ancillary variables.*

to:

*Some model-derived variables, specifically Total Precipitable Water (TPW), the 2-m Temperature ($T_{2m}$), the Skin*

*Temperature, the freezing level height and the temperature and humidity profiles, have been added to the dataset*

*to be used as ancillary parameters.*

**2.15) Line 356: "…for ocean and land respectively."**

Thanks to the reviewer for the correction.

The text has been changed from:

*The estimated spectra are shown in Figure 4 and Figure 5 for the land and ocean classes, respectively.*

to:

*The estimated spectra are shown in Figure 4 and Figure 5 for ocean and land respectively.*

**2.16) Line 387: What is the used atmospheric radiative transfer model? Please spell out RTM.**

Thanks to the reviewer for the comment. The model used is that described by *Rosenkranz, 1998*. The text has been modified from:

*An emissivity spectrum, (calculated as the mean of the emissivity values for each cluster), together with ECMWF*

*temperature and water vapor profiles, is used as input in the RTM to simulate the clear-sky TBs.*

to

*An emissivity spectrum, (calculated as the mean of the emissivity values for each cluster), together with ECMWF*

*temperature and water vapor profiles, is used as input in the radiative transfer model (RTM) (see Ulaby & Long*

*,2014, Rosenkrantz, 1998) to simulate the clear-sky TBs.*

References:

Rosenkranz, P. W., Water vapor microwave continuum absorption: A comparison of measurements and models.

*Radio Science*, *33*(4), 919-928. https://doi.org/10.1029/98RS01182, 1998.

Ulaby, F., & Long, D., Microwave radar and radiometric remote sensing, 1st Edition, the Univ. of Michigan Press,

ISBN: 978-0-472-11935-6, 2014.

**2.17) Table 2: What is the accuracy represented here? The accuracy of PESCA for surface classification?**

Thanks to the reviewer for the comment. The accuracy represented here is the ratio between the number of observations where both SOM and LDA identify the same cluster and the total observations of the class.

**2.18) Line 489: Remove the dot at the beginning of the sentence.**

Thanks to the reviewer for the correction. The text has been largely modified  to address some comments by

Reviewer 1.

from:

. *Generally, it can be observed that, although HANDEL-ATMS is able to detect extremely light snowfall events, it*
*does not have the sensitivity to correctly estimate their intensity.*

to:
*Figure 11 shows the dependence of HANDEL-ATMS snowfall estimation error statistics, as well of SWP and SSR,*
*on TPW . The curves represent the mean SWP or SSR computed for each 1-mm TPW bin, the RMSE and the*
*relative bias (the ratio between the bias and the SWP/SSR mean value for each bin). TPW and snowfall intensity*
*are strongly correlated. An increase of the absolute RMSE can be observed as TPW increases, and it is larger*
*than the SWP/SSR mean value for TPW < 8 mm. A similar behavior can be observed by analyzing the dependence*
*of HANDEL-ATMS snowfall estimation error statistics on $T_{2m}$ (not shown). A very moderate overestimation is*
*observed for TPW < 8 mm and for lower SWP and SSR values (< 0.1 mm/h), with relative bias around 5%, (up*
*to 8% only for extremely low TPW values and very low number of observations (see Figure 7)), while*
*underestimation (relative bias up to -5%) is observed for higher TPW values and higher SWP and SSR values.*
*So, it can be concluded that HANDEL-ATMS has good detection capabilities (also for extremely light snowfall)*
*but it shows some limitations in correctly estimating its intensity, with slight overestimation of the very light*
*snowfall typical of high latitudes.*

[Figure]

*Figure 11: HANDEL-ATMS SWP and SSR Detection Performances for different bins of TPW. The left y-axis*
*reports RMSE absolute values and the mean intensity value for each 1-mm TPW bin, while the relative bias,*
*calculated as the ratio between the bias and the SWP/SSR mean value for each bin.*

**2.19) Figure 1: The inputs of PESCA mentioned in this figure are not aligned with the original paper. For**
**example, there exists no explanation for the low-frequency ratio and scattering coefficients.**

Thanks to the reviewer for the comment. Indeed, there is not a direct mention of the PESCA input parameters;
however, these parameters are derived from the inputs cited in the box (low-frequency ratio is a ratio between two
$TB_{obs}$, the scattering index is a difference between two $TB_{obs}$, $pem_{LF}$ is a ratio between a $TB_{obs}$ and $T_{2m}$, see
*Camplani et al, 2021*). We wanted to highlight that we use the same inputs in more than one module - e. g., TBs
are used both for surface classification and snowfall detection and estimate. The same definition of the input
variables of PESCA can be found in the paper in section 3.1.1.

References:
Camplani, A., Casella, D., Sanò, P., & Panegrossi, G.: The Passive microwave Empirical cold Surface
Classification Algorithm (PESCA): Application to GMI and ATMS. *Journal of Hydrometeorology*, *22*(7), 1727-
1744,https://doi.org/10.1175/JHM-D-20-0260.1, 2021.

**2.20) Figure 6: No results are presented over sea ice.**

Thanks to the reviewer for the comment.
Figure 6 has been modified, with two new subplots related to two PESCA classes (Ocean and New sea Ice).

[Figure]

The following statement has been added to the text (line 423):
*For what concerns ocean and new sea ice classes, a clear scattering signal is visible only for high SWP values*
*(> 1 kg m⁻²) while for low SWP values a significant emission signal is observed. The ubiquitous presence of*
*supercooled water layers in snowing clouds (Wang et al, 2013, Battaglia & Panegrossi 2020), especially over*
*oceans (Battaglia & Delanoe, 2013), generates an emission effect which is particularly significant over*
*radiatively cold surfaces (such as Ocean and New Sea Ice at high frequency, see Figure 4), and can mask or*
*overcome the weak scattering signal generated by falling snow especially in light snowfall events. It is also*
*important to underline that the DARDAR product identifies only supercooled water layers at the cloud top*
*(Panegrossi et al., 2017), while it has been shown that the impact of supercooled water layers embedded in the*
*clouds can be very significant on the measured TBs at MW high frequency window channels (Battaglia &*
*Panegrossi, 2020, Panegrossi et al., 2022) . It is very likely that the emission effect observed over ocean and sea*
*ice is generated by supercooled liquid layers which are not identified by the DARDAR product.*
Figure 6 caption has been modified accordingly
from:
**Figure 6: 165.5 GHz Snowfall Signature as a function of SWP for three Land surface Classes. The red line and**
**shaded areas represent the mean values and standard deviations of ΔTB_obs-sim (i.e., the snowfall signature)**
**while the blue lines are centered on the estimated bias and standard deviation of ΔTB_obs-sim in clear sky**
**conditions for the corresponding PESCA surface class.**

to:
*Figure 6: 165.5 GHz Snowfall Signature as a function of SWP for five PESCA surface classes. The red line and*
*shaded areas represent the mean values and standard deviations of $\Delta TB_{obs-sim}$ (i.e., the snowfall signature)*
*while the blue lines are centered on the estimated bias and standard deviation of $\Delta TB_{obs-sim}$ in clear sky*
*conditions for the corresponding PESCA surface class.*
The following reference has been added to the text (Line 798):
*Wang, Y., Liu, G., Seo, E. K., & Fu, Y.: Liquid water in snowing clouds: Implications for satellite remote sensing*
*of snowfall. Atmospheric research, 131, 60-72, https://doi.org/10.1016/j.atmosres.2012.06.008,2013.*
References:

Battaglia, A., & Delanoë, J.: Synergies and complementarities of CloudSat-CALIPSO snow observations. *Journal*
*of Geophysical Research: Atmospheres*, *118*(2), 721-731. https://doi.org/10.1029/2012JD018092, 2013.
Battaglia, A., & Panegrossi, G.: What can we learn from the CloudSat radiometric mode observations of snowfall
over the ice-free ocean?. Remote Sensing, 12(20), 3285, https://doi.org/10.3390/rs12203285, 2020.
Panegrossi, G., Rysman, J. F., Casella, D., Marra, A. C., Sanò, P., & Kulie, M. S.: CloudSat-based assessment of
GPM Microwave Imager snowfall observation capabilities. *Remote Sensing*, *9*(12), 1263,
https://doi.org/10.3390/rs9121263, 2017.
Panegrossi, G., Casella, D., Sanò, P., Camplani, A., & Battaglia, A.: Recent advances and challenges in satellite-
based snowfall detection and estimation. *Precipitation Science*, 333-376, https://doi.org/10.1016/B978-0-12-
822973-6.00015-9, 2022.
Wang, Y., Liu, G., Seo, E. K., & Fu, Y.: Liquid water in snowing clouds: Implications for satellite remote sensing
of snowfall. *Atmospheric research*, *131*, 60-72, https://doi.org/10.1016/j.atmosres.2012.06.008,2013.

**2.21) Figure 10: Please mention that the shown green dots denote the CPR overpass.**
Thanks to the reviewer for the suggestion. The caption of Figures 10 12, and 13 (now Figures 12, 14, and 15)  has
been changed
Figure 10/12:
from:
*Figure 10:  Greenland - 2016/04/24 - PESCA Background Surface Classification.*
to:
*Figure 12:  Greenland - 2016/04/24 - PESCA Background Surface Classification. The green dotted line*
*represents the CloudSat track.*

Figure 12/14:

from:

*Figure 12: Greenland - 2016/04/24 - 165 GHz Channel measured TB (TB$_{obs}$) (top panel) and the deviation of*
*TB$_{obs}$ from the simulated clear-sky TBs ( $\Delta$TB$_{obs-sim}$) (bottom panel)*
to:
*Figure 14: Greenland - 2016/04/24 - 165 GHz Channel measured TB (TB$_{obs}$) (top panel) and the deviation of*
*TB$_{obs}$ from the simulated clear-sky TBs ( $\Delta$TB$_{obs-sim}$) (bottom panel). The red dotted line (top*
*panel) and the green dotted line (bottom panel)  represent the CloudSat track.*

Figure 13/15:

from:

*Figure 13:  Greenland - 2016/04/24 - Maps of the HANDEL-ATMS module's output: the SWP detection mask*
*(top panel), the estimated SWP (kg m$^{-2}$) (second panel), the SSR detection mask (third panel), the estimated SSR*
*(mm h$^{-1}$) (bottom panel).*
to:
*Figure 15:  Greenland - 2016/04/24 - Maps of the HANDEL-ATMS module's output: the SWP detection mask*
*(top panel), the estimated SWP (kg m$^{-2}$) (second panel), the SSR detection mask (third panel), the estimated SSR*
*(mm h$^{-1}$) (bottom panel). The green dotted lines (bottom panel)  represent the CloudSat track.*

---

## Referee Report (RR1)

This work by Camplani et al. presented a new snowfall detection and intensity estimation technique. The results are very encouraging. I have several minor comments.

Minors:

1. From line 567, you explained why the newly designed method (HANDEL) performs better than SLALOM. Two reasons are provided for the better performance from HANDLE, including (1) regional database vs. global database; and (2) environmental parameters from model vs. from TBs. Which factor do you think is more important for the better performance from HANDLE?

2. Figure 3 and the corresponding texts: can you explain in detail how you define the "pseudo-emissivity". Some of these emissivity values are greater than 1.

3. Figure 4. It seems that the emissivity at 165 GHz is too low over ocean. See below, it can be as low as 0.35 for 165 GHz. Can you please check.

[Figure]

885

4. Figure 7 bottom panel. It shows that the HSS is smaller for TPW being 10 mm, compared with TPW being 4 mm. Specifically, the HSS decreases from about 0.6 to about 0.4. I am surprised by this result. Can you explain why? In contrast for snow water path (top panel), the HSS remains about 0.6.

[Figure]

5. Fig. 10, As a comparison, can you provide a similar two-panel plots from SLALOM-CT?

6. Are these results for all ATMSs (i.e., NPP, NOAA20, and NOAA21)?

---

## Author Response (AR2)

**Author's response**

**Public justification (visible to the public if the article is accepted and published**):
**The referees have all suggested public subject to minor revisions or technical corrections, however, the number of suggested revisions is quite large. Since each of the referees has indicated a willingness to provide further review, I have decided to "reconsider after major revisions", although the authors should be aware that there is no foreseen impediment to publication at this time; the decision is only to allow the referees to provide one additional review.**

We would like to thank the editor. Please, find below our responses to the Reviewers' comments and the details on how we address them in the new version of the manuscript.

The following main changes have been set with respect to the previous manuscript version, in order to answer the reviewers' comments:

Table 2 and Figure 9 have been added and the numbers of the subsequent figures and tables have changed accordingly.

The following references have been added:

*Delanoë, J., and R. J. Hogan: Combined CloudSat-CALIPSO-MODIS retrievals of the properties of ice clouds. J. Geophys. Res., 115, D00H29, doi:10.1029/2009JD012346, 2010.*

*Kidd, C., Becker, A., Huffman, G. J., Muller, C. L., Joe, P., Skofronick-Jackson, G., & Kirschbaum, D. B.: So, how much of the Earth's surface is covered by rain gauges?. Bulletin of the American Meteorological Society, 98(1), 69-78, https://doi.org/10.1175/BAMS-D-14-00283.1, 2017.*

Moreover, Figures 6, 8, 10 (11 in the revised manuscript), 11 (12 in the revised manuscript), 13 (14 in the revised manuscript), 15 (16 in the revised manuscript), and 16 (17 in the revised manuscript) have been modified.

Reviewer 1
**I think the paper can be published after some clarifications about the used RT model.**
We would like to thank Reviewer #1 for his/her review of our paper and the important comments and suggestions
provided. Please, find below our responses to the Reviewer's comments and the details on how we address them
in the new version of the manuscript.
**1.1) It appears that the used model is a zeroth-order approximation of the radiative transfer equation,**
**which can only be applied to a weakly scattering medium. The atmospheric attenuation is modeled by a**
**one-way transmissivity parameter, which may be highly uncertain for high-frequency channels when ice**
**and snow particles strongly scatter the upwelling emission. This class of models is applicable largely to low-**
**frequency channels with minimal atmospheric scattering. This caveat needs to be acknowledged. Further**
**details about the RT model seem to be necessary. The authors might consider addressing this in the**
**appendix.**
Thanks to the reviewer for the comment. We totally agree with the reviewer about the high uncertainty of the
modeling of the scattering effect for high-frequency channels. However, in this work we apply the RT model in
clear sky conditions, i.e. we consider only absorption and emission from atmospheric gasses and surface emission
in the RT, which can be considered scatter-free for the microwave frequency range. Therefore, a comparison
between the clear-sky simulated signal and the observed ones is performed, in order to highlight the snowfall
signature.
In lines 103-104 (lines 93-94 in the revised manuscript) the following statement is reported:
*The derived surface emissivities are used to infer the clear-sky contribution to the measured brightness*
*temperatures (TBs) in the high-frequency channels in the snowfall retrieval process.*
To make the text clearer, the term "*simulated TBs*" and the acronym "*TB$_{sim}$*" have been replaced with "*clear-sky*
*simulated TBs*".

Reviewer 2

**This work by Camplani et al. presented a new snowfall detection and intensity estimation technique. The results are very encouraging. I have several minor comments.**

We would like to thank Reviewer #2 for his/her review of our paper and the important comments and suggestions provided. Please, find below our responses to the Reviewer's comments and the details on how we address them in the new version of the manuscript.

**Minors:**

**2.1) From line 567, you explained why the newly designed method (HANDEL) performs better than SLALOM. Two reasons are provided for the better performance from HANDEL, including (1) regional database vs. global database; and (2) environmental parameters from model vs. from TBs. Which factor do you think is more important for the better performance from HANDEL?**

Thanks to the reviewer for pointing out this aspect. For sure, both factors are relevant. However, we think that the added value of the approach is the use of the differences between simulated and observed TBs. In Table 3 the statistical scores evaluated for the surface snowfall rate (SSR) detection module by using different predictor sets (for the test dataset) are reported. It can be observed that the use of the differences between measured and simulated clear-sky TBs gives better performances (the fourth row of Table 3 shows an HSS=0.69) with respect to the addition of environmental variables to the predictor set (the third row of Table 3 shows an HSS=0.68). This second approach is very similar to that used in SLALOM-CT, except for the fact that HANDEL-ATMS is trained over a "regional" database, so we can state that the added value of HANDEL-ATMS derives from the use of the $\Delta TB_{obs\text{-}sim}$ (differences between measured and simulated clear-sky TB).

To clarify this point, the following statement is reported in the text (lines 448-456):

*It is possible to see that the best performance is obtained when the predictor set is composed of ATMS $TB_{obs}$ and $\Delta TB_{obs-sim}$, (besides PESCA surface flag, the pixel elevation and the cosine of the viewing angle). In particular, it is notable the improvement of the detection capabilities with respect to a predictor set composed of ATMS $TB_{obs}$ and environmental parameters. On the other hand, the simultaneous use of both the $\Delta TB_{obs\text{-}sim}$ and the environmental parameters show scores almost equal to that obtained by using only $\Delta TB_{obs\text{-}sim}$. This indicates that the computation of the multi-channel clear-sky TBs at the time of the overpass through the estimation of the dynamic surface class emissivity spectra and its deviation from the measured TBs plays a fundamental role in snowfall retrieval. It provides essential information to the ANN to be able to exploit the subtle snowfall-related signal in ATMS measurements. This is the most innovative aspect of HANDEL-ATMS.*

which has been modified as (lines 449-459 in the revised manuscript):

*It is possible to see that the best performance is obtained when the predictor set is composed of ATMS $TB_{obs}$ and $\Delta TB_{obs-sim}$, (besides the PESCA surface flag, the pixel surface elevation, and the cosine of the viewing angle). In particular, it is notable the improvement of the detection capabilities with respect to a predictor set composed of ATMS $TB_{obs}$ and environmental parameters, **which is used in other approaches such as that of SLALOM-CT**. On the other hand, the simultaneous use of both the $\Delta TB_{obs\text{-}sim}$ and the environmental parameters show scores almost equal to that obtained by using only $\Delta TB_{obs\text{-}sim}$. This indicates that the computation of the multi-channel clear-sky TBs at the time of the overpass through the estimation of the dynamic surface class emissivity spectra and its deviation from the measured TBs plays a fundamental role in snowfall retrieval, **in particular in cold/dry environmental conditions**. It provides essential information to the ANN to be able to exploit the subtle snowfall-related signal in ATMS measurements. This is the most innovative aspect of the HANDEL-ATMS.*

**2.2) Figure 3 and the corresponding texts: can you explain in detail how you define the "pseudo- emissivity". Some of these emissivity values are greater than 1.**

Thanks to the reviewer for the suggestion. In lines 287-288, there is the following statement:

*23 GHz pseudo-emissivity (i. e. the ratio between an observed brightness temperature (TB) and a near surface*
*temperature value) -*
However, the text has been modified to make this definition clearer. In particular, lines 287-288 (lines 282-283 in
the revised manuscript) have been modified
from:
*23 GHz pseudo-emissivity (i. e. the ratio between an observed brightness temperature (TB) and a near surface*
*temperature value) - pem23).*
to
*23 GHz pseudo-emissivity (**pem23**) (i.e., the ratio between the 23 GHz observed TB and the near-surface*
*temperature value).*
and lines 300-302 (lines 294-297 in the revised manuscript) have been modified
from:
*Downstream of the sea ice/open water identification, information about sea ice characteristics is obtained from*
*the analysis of the two low-frequency pseudo-emissivity (pem23 and pem31), which are a good approximation of*
*sea-ice emissivity for low-frequency channels especially in cold and dry conditions.*
to:
*Downstream of the sea ice/open water identification, information about sea ice characteristics is obtained from*
*the analysis of the two low-frequency pseudo-emissivity values (pem23 and pem31) (defined as the ratio between*
*the observed TB and the near-surface temperature value) which can be considered a good approximation of sea-*
*ice emissivity for low-frequency channels especially in cold and dry conditions.*
**2.3) Figure 4. It seems that the emissivity at 165 GHz is too low over ocean. See below, it can be as low as**
**0.35 for 165 GHz. Can you please check.**

[Figure]

Thanks to the reviewer for the suggestion. The points highlighted in the plot correspond to $183.31 \pm 7$ GHz and
not to 165.5 GHz. The values highlighted are the emissivity for the "Ocean" class and the lower limit of the
"standard deviation" belt respectively. In the following figure, a comparison between the PESCA Ocean class
emissivity retrieved values for ATMS frequency channels and the emissivity spectrum derived from the TESSEM
model (*Prigent et al, 2017*) for Open Water is reported.

[Figure]

It is possible to observe that there is a good agreement up to 165.5 GHz, while at 183.31±7 GHz. On the contrary,
the 183.31±7 GHz mean emissivity value is lower with respect to that obtained by applying the emissivity model,
and it is characterized by high standard deviation. However, these results are preliminary, and a refinement process
has been carried out by clustering each PESCA surface class. The difference between the emissivity at 183.31±7
GHz derived from PESCA and the ones in TESSEM is due to the optical thickness of the atmosphere in the water
vapor absorption band. In the 183.31 GHz band, the atmosphere is opaque, due to water vapor absorption and a
direct estimate of the emissivity from the observed TBs presents some issues; also downstream of the refinement
process, the emissivity values obtained are lower than those expected. However, in these conditions the opacity
of the atmosphere guarantees the minor impact of the surface conditions on the upwelling radiation; in fact, despite
the emissivity underestimation at 183.31±7 GHz, the RMSE of the simulated clear sky TBs, as compared to the
observed ones, is very small (about 3.5 K).
REFERENCES:

Prigent, C., Aires, F., Wang, D., Fox, S., & Harlow, C.: Sea-surface emissivity parametrization from microwaves
to millimetre waves. *Quarterly Journal of the Royal Meteorological Society*, *143*(702), 596-605.
https://doi.org/10.1002/qj.2953, 2017.

**2.4) Figure 7 bottom panel. It shows that the HSS is smaller for TPW being 10 mm, compared with TPW**
**being 4 mm. Specifically, the HSS decreases from about 0.6 to about 0.4. I am surprised by this result. Can**
**you explain why? In contrast for snow water path (top panel), the HSS remains about 0.6.**

[Figure]

Thanks to the reviewer for the question. This behavior is due to the fact that there is not a perfect correspondence
between the snow water path flag and the snowfall rate flag derived from the CloudSat CPR 2C-Snow profile
product, and so there are observations (about 10 % of the SWP observations in the selected datasets) where the
presence of snow in the atmosphere is not matched by the presence of surface snowfall because of warmer near-
surface conditions. In the following Figures the histograms of SWP/SSR occurrences as a function of TPW and
$T_{2m}$ are reported. It is possible to observe that in moister/warmer environmental conditions there is a larger number
of SWP observations than SSR ones.

[Figure]

[Figure]

Generally, PMW measurements respond mostly to the snow in the atmospheric column than to snowfall at the
ground, so SWP statistical scores tend to improve with increasing TPW and $T_{2m}$ while SSR statistical scores show
a maximum for TPW between 3-4 mm (or $T_{2m}$ around 270 K) and then decrease in conditions where the mismatch
between SWP and SSR become significant. In lines 473-476 (lines 477-482 in the revised manuscript) there are
the following statements:
*It is possible to observe that in Figure 7 SSR detection capabilities show a maximum HSS value for TPW between*
*3 mm and 5 mm, and then there is a slight decrease due to the decrease of POD. A similar situation can be*
*observed in Figure 8, where HSS reaches a maximum between 250 K and 275 K, and it is lower than for SWP.*
***This is due to the fact that PMW measurements respond mostly to the snow in the atmospheric column and in***
***moister/warmer conditions the presence of snow in the atmosphere is not always linked to surface snowfall.***

In order to address the Reviewer's comment, a new Table has been added to the paper:

| Class | TPW (mm) | $T_{2m}$ (K) | # obs | % SWP obs | % SSR obs | SWP (kg m$^{-2}$) | SSR (mm h$^{-1}$) |
|---|---|---|---|---|---|---|---|
| Ocean | 6.2 | 273 | $3.9*10^5$ | 79 | 64 | 0.046 | 0.071 |
| New Sea Ice | 3.2 | 255 | $2.1*10^5$ | 38 | 38 | 0.033 | 0.050 |
| Broken Sea Ice | 5.2 | 266 | $1.4*10^5$ | 57 | 57 | 0.044 | 0.073 |
| Multilayer Sea Ice | 4.5 | 260 | $9.9*10^4$ | 43 | 43 | 0.033 | 0.051 |
| Land | 5.3 | 270 | $2.8*10^4$ | 43 | 41 | 0.043 | 0.068 |
| Perennial Snow | 1.6 | 248 | $3.6*10^5$ | 31 | 31 | 0.022 | 0.035 |
| Winter Polar Snow | 2.1 | 245 | $6.0*10^4$ | 32 | 32 | 0.033 | 0.048 |
| Deep Dry Snow | 3.8 | 261 | $1.1*10^5$ | 50 | 50 | 0.040 | 0.066 |
| Thin Snow | 4.5 | 267 | $1.8*10^4$ | 54 | 53 | 0.041 | 0.070 |
| Coast | 4.0 | 259 | $3.1*10^5$ | 47 | 46 | 0.043 | 0.068 |

**Table 2: Environmental Characteristics for each PESCA class (test dataset):  the number of occurrences, the**
**mean TPW and $T_{2m}$ value, the percentage of   SWP/SSR observations (over the total occurrences), and the**
**mean SWP and SSR values are shown**

and the following statements have been added to the text (line 342, lines 336-344 in the revised manuscript):
*In Table 2 the number of PESCA class occurrences, the percentage of snowfall observations, and the most*
*significant environmental characteristics in the ATMS-CPR coincident dataset are reported. It can be observed*
*that Land and Ocean classes are characterized by the warmest/moistest conditions and by the most intense*
*snowfall events (on average), while Perennial and Winter Polar Snow classes and New and Multilayer Sea Ice*
*classes are characterized by the coldest/driest environmental conditions and by the lightest snowfall events (on*
*average). Thin Snow and Broken Sea Ice classes show intermediate environmental conditions and snowfall*
*intensity values.* ***It is also interesting to highlight that a mismatch between the percentage of SWP and SSR***
***observations is observed mostly over the Ocean class and, less frequently other classes (Land, Thin Snow, and***
***Coast), where warmer and moister environmental conditions are found.***

**2.5) Fig. 10, As a comparison, can you provide a similar two-panel plot from SLALOM-CT?**

Thanks to the reviewer for the suggestion. These scatterplots have been already reported in the following article
by the same authors of the present paper:

Sanò, P., Casella, D., Camplani, A., D'Adderio, L. P., & Panegrossi, G., A Machine Learning Snowfall Retrieval
Algorithm for ATMS. Remote Sensing, 14(6), 1467, https://doi.org/10.3390/rs14061467, 2022.

Therefore, we decided not to include it in this paper.

**2.6) Are these results for all ATMSs (i.e., NPP, NOAA20, and NOAA21)?**

Thanks to the reviewer for the question. The study, currently, has been carried out over a dataset from 2014 to
2016, so only observations from ATMS onboard NPP were available. However, we are confident that HANDEL
can be used by exploiting the ATMS measurements provided by satellites following NPP.  A dedicated study is
being carried out to verify if HANDEL's performance remains consistent for the other satellites.
In lines 193-196 (lines 185-189 in the revised manuscript) there is the following statement:

*The present study is based on a coincidence dataset between CPR and SNPP ATMS observations between January*
*2014 and August 2016. The same dataset has been used for the development of SLALOM-CT (Sanò et al, 2022).*
*Each coincidence comes from observations from CloudSat CPR and ATMS - onboard SNPP - within a maximum*
*15-minute time window.*

However, to make this concept clearer, the text has been modified to:

*The present study is based on a coincidence dataset between CPR and ATMS observations between January 2014*
*and August 2016. The same dataset has been used for the development of SLALOM-CT (Sanò et al, 2022). Each*
*coincidence comes from observations from CloudSat CPR and ATMS within a maximum 15-minute time window.*
*In the period considered within the dataset, only the SNPP satellite was in orbit, so the dataset is composed only*
*of observations obtained from ATMS onboard this satellite.*

Reviewer 3

This AMT manuscript submission describes a new ATMS Machine Learning (ML) snowfall detection algorithm (HANDEL-ATMS) that is trained on ATMS-CloudSat observations and products. This algorithm can be considered as a new retrieval scheme with strong ties to a productive lineage of microwave retrieval algorithms from this research group. The current retrieval applies an updated methodology and exploits a different sensor (ATMS) to detect and quantify snowfall rates using cross-track microwave sounder observations compared to previous related retrievals developed by this group. HANDEL-ATMS is also specifically developed to improve snowfall detection and estimation at high latitudes.

Overall, the results presented in this study are meaningful to the microwave precipitation remote sensing community and deserve to be published. The authors demonstrate that this algorithm performs well under the typically challenging conditions (light snowfall rates, very dry atmospheric conditions, surface emissivity complications) that often occur at high latitudes. Key algorithm components that enable improved algorithm performance are also described and highlighted.

I recommend that the manuscript be published after the authors consider the minor comments listed below.

We would like to thank Reviewer #3 for his/her review of our paper and the important comments and suggestions provided. Please, find below our responses to the Reviewer's comments and the details on how we address them in the new version of the manuscript.

**3.1) Abstract: The first paragraph can be reduced considerably since it is covered exhaustively and effectively in the introduction. A possible way to reorganize the abstract is:**

**The High lAtitude sNow Detection and Estimation aLgorithm for ATMS (HANDEL-ATMS) is a new machine learning (ML)-based snowfall retrieval algorithm for Advanced Technology Microwave Sounder (ATMS) observations that is developed specifically to detect and quantify high latitude snowfall events that often form in cold, dry environments and produce light snowfall rates. ATMS and the future European MetOp-SG Microwave sounder offer good high latitude coverage and sufficient microwave channel diversity (23 to 190 GHz) that allows both surface radiometric properties to be dynamically characterized and the non-linear and sometimes subtle passive microwave response to falling snow to be detected. HANDEL-ATMS is based on a combined active-passive microwave observational dataset in the training phase, where each ATMS multichannel observation is associated with coincident (in time and space) CloudSat Cloud Profiling Radar (CPR) vertical snow profiles and surface snowfall rates. {The rest of the second abstract paragraph can follow.}**

**The above paragraph is only a suggestion and not mandatory. But it offers a way to distill and condense much of the introductory/background/motivation content into only 2-3 sentences.**

Thanks to the reviewer for the suggestion. The text in the Abstract (Lines 8-27, lines 8-17 in the revised manuscript) has been modified as suggested from:

*Snowfall detection and quantification are challenging tasks in the Earth system science field. Ground-based instruments have limited spatial coverage and are scarce or absent at high latitudes. Therefore, the development of satellite-based snowfall retrieval methods is necessary for the global monitoring of snowfall. Passive Microwave (PMW) sensors can be exploited for snowfall quantification purposes because their measurements in the high-frequency channels (> 80 GHz) respond to snowfall microphysics. However, the highly non-linear PMW multichannel response to snowfall, the weakness of snowfall signature and the contamination by the background surface emission/scattering signal make snowfall retrieval very difficult. This phenomenon is particularly evident at high latitudes, where light snowfall events in extremely cold and dry environmental conditions are predominant. Machine Learning (ML) techniques have been demonstrated to be very suitable to handle the complex PMW multichannel relationship to snowfall. Operational microwave sounders on near-polar orbit satellites such as the*

*Advanced Technology Microwave Sounder (ATMS), and the European MetOp-SG Microwave Sounder in the*
*future, offer a very good coverage at high latitudes. Moreover, their wide range of channel frequencies (from 23*
*GHz to 190 GHz), allows for the dynamic radiometric characterization of the surface at the time of the overpass*
*along with the exploitation of the high-frequency channels for snowfall retrieval. The paper describes the High*
*lAtitude sNow Detection and Estimation aLgorithm for ATMS (HANDEL-ATMS), a new machine learning-based*
*snowfall retrieval algorithm developed specifically for high latitude environmental conditions and based on the*
*ATMS observations.*
*HANDEL-ATMS is based on the use of an observational dataset in the training phase, where each ATMS*
*multichannel observation is associated with coincident (in time and space) CloudSat Cloud Profiling Radar (CPR)*
*vertical snow profile and surface snowfall rate.*
to:
*The High lAtitude sNow Detection and Estimation aLgorithm for ATMS (HANDEL-ATMS) is a new machine*
*learning (ML)-based snowfall retrieval algorithm for Advanced Technology Microwave Sounder (ATMS)*
*observations that is developed specifically to detect and quantify high latitude snowfall events that often form in*
*cold, dry environments and produce light snowfall rates. ATMS and the future European MetOp-SG Microwave*
*Sounder offer good high-latitude coverage and sufficient microwave channel diversity (23 to 190 GHz) that allows*
*both surface radiometric properties to be dynamically characterized and the non-linear and sometimes subtle*
*passive microwave response to falling snow to be detected. HANDEL-ATMS is based on a combined active-*
*passive microwave observational dataset in the training phase, where each ATMS multichannel observation is*
*associated with coincident (in time and space) CloudSat Cloud Profiling Radar (CPR) vertical snow profiles and*
*surface snowfall rates.*
**3.2) Lines 42-44: I suggest offering an appropriate reference that illustrates and quantifies the lack of**
**surface gauge coverage globally (e.g., Kidd et al. 2017).**
Thanks to the reviewer for the suggestion. Lines 42-44 (lines 32-34 in the revised manuscript) have been modified
from:
*However, global snowfall quantification is a challenging topic in weather sciences. Ground-based instruments*
*such as raingauges or snowgauges provide only punctual measurements which can not fully capture the spatial*
*variability of precipitation phenomena;*
to:
*However, global snowfall quantification is a challenging topic in weather sciences. Ground-based instruments*
*such as raingauges or snowgauges provide only punctual measurements which can not fully capture the spatial*
*variability of precipitation phenomena (**Kidd et al, 2017**);*
and the following reference has been added to the reference section:
*Kidd, C., Becker, A., Huffman, G. J., Muller, C. L., Joe, P., Skofronick-Jackson, G., & Kirschbaum, D. B.: So,*
*how much of the Earth's surface is covered by rain gauges?. Bulletin of the American Meteorological Society,*
*98(1), 69-78, https://doi.org/10.1175/BAMS-D-14-00283.1, 2017.*
**3.3) Lines 44-45: consider a more active writing style and shortening the sentence:"...the variability of**
**snowflake shape and density strongly influences particle fall speed and trajectory and therefore reduces**
**the gauge-based measurement accuracy of falling snow, especially compared to rain measurements (see**
**Skofronick-Jackson et al 2015)"**
Thanks to the reviewer for the suggestion. Lines 44-46 (lines 34-36 in the revised manuscript) have been modified
from:
*moreover, the variability of snowflake shape and density has a strong influence on their fall speed and trajectories*
*and therefore gauge-based measurements of falling snow result to be less accurate than for rain (see Skofronick-*
*Jackson et al, 2015).*
to:

*moreover, the variability of snowflake shape and density strongly influences particle fall speed and trajectory and*
*therefore reduces the gauge-based measurement accuracy of falling snow, especially compared to rain*
*measurements (Skofronick-Jackson et al, 2015).*
**3.4) Lines 107-110: similar to the previous comment. "Moreover, the algorithm also exploits an**
**observational dataset composed of ATMS multichannel observations and coincident (time and space)**
**CloudSat CPR vertical snow profiles and surface snowfall rates (hereafter the ATMS-CPR coincident**
**dataset)". I will refrain from offering further ways to condense content and provide a more active writing**
**style, but please know that I can provide further suggestions.**
Thanks to the reviewer for the suggestion. Lines 107-110 (lines 97-99 in the revised manuscript) have been
modified
from:
*Moreover, the algorithm is based on the exploitation of an observational dataset where each ATMS multichannel*
*observation is associated with coincident (in time and space) CloudSat CPR vertical snow profile and surface*
*snowfall rate (hereafter ATMS-CPR coincidence dataset).*
to:
*Moreover, the algorithm also exploits an observational dataset composed of ATMS multichannel observations*
*and coincident (time and space) CloudSat CPR vertical snow profiles and surface snowfall rates (hereafter the*
*ATMS-CPR coincident dataset).*
**3.5) Line 128: It might be worth mentioning explicitly here that the CPR may struggle with high snowfall**
**rates, but also note that CPR is uniquely suited to detect light snowfall rates that dominate high latitudes.**
**EDIT: The authors mention high snowfall rate underestimation in Line 183, which is great. But I still think**
**it is worthwhile to also highlight CloudSat's strength of detecting light snowfall - GPM, the only other**
**spaceborne radar, cannot be used for training since its detection limit is far too high to effectively detect**
**light snowfall. GPM's orbit also renders it largely useless to very high latitudes.**
Thanks to the reviewer for the suggestion. See the answer to question 3.6.
**3.6) Lines 189-191: This proves that I should read the entire article before commenting. My previous**
**comment has mostly been rectified by this content. Feel free to ignore it, or at the very least explicitly**
**highlight the further GPM drawbacks that make CPR the optimal training dataset for high latitude**
**snowfall applications.**
Thanks to the reviewer for the suggestion. The following statement has been added to the text (Line 191, lines
180-183 in the revised manuscript)
*These features appear to be an advantage compared to the GPM-Core Observatory (GPM-CO), which provides*
*observations only between 67 ° N and 67 ° S, and to the $K_u$- and $K_a$-band DPR has low sensitivity and is not*
*suitable to effectively detect light snowfall events (Casella et al, 2017).*
**3.7) Line 216: The parenthetical DARDAR reference is incomplete.**
Thanks to the reviewer for the comment. Lines 215-216 (lines 209-210 in the revised manuscript) have been
modified
from:
*The supercooled water information has been extracted from the DARDAR product (see DARDAR).*
to:
*The supercooled water information has been extracted from the DARDAR product (DARDAR, Delanoë &Hogan,*
*2010).*
and the following reference has been added to the reference section:

*Delanoë, J., and R. J. Hogan: Combined CloudSat-CALIPSO-MODIS retrievals of the properties of ice clouds. J. Geophys. Res., 115, D00H29, doi:10.1029/2009JD012346, 2010.*

**3.8) Lines 236-237: Rephrase slightly to "Moreover, clustering techniques have been used to characterize the background surface from a radiometric point of view."**

Thanks to the reviewer for the suggestion. Lines 236-237 (lines 230-231 in the revised manuscript) have been modified from:

*Moreover, clustering techniques have been used to characterize from a radiometric point of view the background surface.*

to:

*Moreover, clustering techniques have been used to characterize the background surface from a radiometric point of view.*

**3.9) Lines 286-287: The inputs listed in parentheses are somewhat confusing to read due to embedded parentheses.**

Thanks to the reviewer for the comment.

Lines 285-288 (lines 280-283 in the revised manuscript) have been modified from:

*It is based on a decision tree that makes use of a limited number of inputs (the ratio $TB_{23QV}/TB_{31QV}$ - **ratio**, the difference between $TB_{23QV}$ and $TB_{88QV}$ or Scattering Index - **SI**, 23 GHz pseudo-emissivity (i. e. the ratio between an observed brightness temperature (TB) and a near surface temperature value) - **pem23**).*

to:

*It is based on a decision tree that makes use of a limited number of inputs: the ratio between $TB_{23QV}$ and $TB_{31QV}$ (ratio), the difference between $TB_{23QV}$ and $TB_{88QV}$ or Scattering Index (SI), 23 GHz pseudo-emissivity (pem23) (i.e., the ratio between the 23 GHz observed TB and the near-surface temperature value).*

**3.10) Fig. 3: I initially thought the green line indicated in the first two figure panels was somehow related to the green discriminant line indicated in Fig. 2, but I think it is the 1:1 line. Consider either explicitly mentioning this in the figure caption, or change the color or linestyle of the 1:1 line in Fig. 3.**

Thanks to the reviewer for the suggestion.

The caption of Figure 3 has been modified from:

*Figure 3: Sea Ice detection and classification: relationship between 31 GHz Pseudo-Emissivity (y-axis) and 23 GHz Pseudo-Emissivity (x-axis). The color represents the mean AutoSnow sea ice percentage within each bin (top panel), the observation occurrence (middle panel), and the PESCA classification (Multi-Layer (ML), Broken and New sea ice) with the Nearest Neighbor markers (bottom panel).*

to:

*Figure 3: Sea Ice detection and classification: relationship between 31 GHz Pseudo-Emissivity (y-axis) and 23 GHz Pseudo-Emissivity (x-axis). The color represents the mean AutoSnow sea ice percentage within each bin (top panel), the observation occurrence (middle panel), and the PESCA classification (Multi-Layer (ML), Broken and New Sea Ice) with the Nearest Neighbor markers (bottom panel). The green continuous lines at the top and the center panels represent the bisector.*

**3.11) Line 339: remove the letter "e" after "constantly"**

Thanks to the reviewer for the comment. Line 339 (line 333 in the revised manuscript) has been modified from:

*and constantly e throughout the year,*

to:
*and constantly throughout the year,*
**3.12) Line 394: "represents" should be regular, not superscript, font size**
Thanks to the reviewer for the comment. Line 394 (line 398 in the revised manuscript) has been modified
from:
*where σ* *represents*
to:
*where σ represents*

**3.13) Line 462-463: This is somewhat of a general question, but is it worth comparing/contrasting high**
**latitude algorithm performance versus any other ATMS snowfall (or general precipitation) retrievals that**
**have been developed? Or other microwave retrievals? The statement provided in these lines provoked this**
**thought. Do other precipitation retrievals provide similar statistical results (POD > 0.8, FAR < 0.2) at high**
**latitudes?**
Thanks to the reviewer for the comment. See the answer to question 3.18.
**3.14) Lines 464-465 and Tables 3, 4, and 5 captions: I recommend explicitly advertising to readers what**
**validation dataset is used to generate the statistical scores as a function of TPW and T2m. I presume CPR**
**2C-SNOW not used for training?**
Thanks to the reviewer for the suggestion. Yes, the statistical scores have been calculated for the test dataset (see
Subsection 2.3, Lines 229-234, lines 223-228 in the revised manuscript).
Lines 461-462 (lines 465-467 in the revised manuscript) have been modified (by considering that a new table -
Table 2 - has been added)
from:
*In Table 4 the statistical scores of HANDEL-ATMS detection module performances are reported in terms of POD,*
*FAR and HSS*
to:
*In Table 5 the statistical scores of HANDEL-ATMS detection module performances are reported in terms of POD,*
*FAR, and HSS. These statistical scores - and the plot reported in the next figures - have been calculated for the*
*test dataset.*
The caption of Figure 7 has been modified
from:
*Figure 7: Dependence of HANDEL-ATMS SWP and SSR detection statistical scores on TPW. Each star represents*
*the statistical score value for different 1-mm t bin of TPW. The left y-axis reports POD, FAR and HSS values,*
*while the right y-axis reports the number of total and snowfall observations in the validation dataset.*
to:
*Figure 7: Dependence of HANDEL-ATMS SWP and SSR detection statistical scores on TPW calculated for the*
*test dataset. Each star represents the statistical score value for different 1-mm bin of TPW. The left y-axis reports*
*POD, FAR and HSS values, while the right y-axis reports the number of total and snowfall observations in the*
*test dataset.*
The caption of Figure 10 (Figure 11 in the revised manuscript) has been modified
from:
*Figure 10: 2D Histogram reporting HANDEL-ATMS SWP (left) and SSR (right) estimation (y-axis) and 2CSP*
*estimation (x-axis). The colorbar represents the number of observations for each HANDEL ATMS/2CSP bin. The*
*violet dashed line represents the bisector.*
to:

*Figure 11: 2D Histogram reporting HANDEL-ATMS SWP (left) and SSR (right) estimation (y-axis) and 2CSP*
*estimation (x-axis). The colorbar represents the number of observations for each HANDEL ATMS/2CSP bin (test*
*dataset). The violet dashed line represents the bisector.*
The caption of Figure 11 (Figure 12 in the revised manuscript) has been modified
from:
*Figure 11: Dependence of HANDEL-ATMS SWP and SSR estimation on TPW. Each star represents the value of*
*the statistical score for different 1-mm TPW bins. The left y-axis reports the RMSE and the mean intensity SWP*
*and SSR value for each 1-mm TPW bin, while the right y-axis reports the relative bias, calculated as the ratio*
*between the bias and the SWP/SSR mean value for each bin*
to:
*Figure 12: Dependence of HANDEL-ATMS SWP and SSR estimation on TPW calculated for the test dataset. Each*
*star represents the value of the statistical score for different 1-mm TPW bins. The left y-axis reports the RMSE*
*and the mean intensity SWP and SSR value for each 1-mm TPW bin, while the right y-axis reports the relative*
*bias, calculated as the ratio between the bias and the SWP/SSR mean value for each bin*
The caption of Table 3 (Table 4 in the revised manuscript) has been modified
from:
*Table 3: HANDEL-ATMS SSR Detection Performance: Statistical scores for different Predictor Sets*
to:
*Table 4: HANDEL-ATMS SSR Detection Performance: Statistical scores for different Predictor Sets. The*
*statistical scores have been calculated for the test dataset.*
The caption of Table 4 (Table 5 in the revised manuscript) has been modified
from:
*Table 4: HANDEL-ATMS detection Performance - SWP and SSR Detection Modules Statistical Scores*
to:
*Table 5: HANDEL-ATMS detection Performance - SWP and SSR Detection Modules Statistical Scores. The*
*statistical scores have been calculated for the test dataset.*
The caption of Table 5 (Table 6 in the revised manuscript) has been modified
from:
*Table 5: HANDEL-ATMS Estimation Performance - SWP and SSR Estimation Module Error Statistics*
to:
*Table 6: HANDEL-ATMS Estimation Performance - SWP and SSR Estimation Module Error Statistics. The error*
*statistics have been calculated for the test dataset.*
**3.15) Figs. 7, 8, and 9 and Line 467: Just to be certain that I am interpreting these figures correctly, are the**
**POD statistics valid for the entire distribution of snowfall rates and snow water paths in each 1 mm TPW**
**bin? It would be interesting to provide further context somewhere about how the snowfall rate and snow**
**water path distributions vary as a function of TPW. EDIT: Fig. 11 does illustrate SWP and SSR**
**distributions as a function of TPW. Maybe move Fig. 11 before current Figs. 7, 8, and 9 to provide more**
**context regarding mean SWP and SSR values before the POD and FAR statistics are provided?**
Thanks to the reviewer for the suggestion. Yes, your interpretation is correct. We think that moving the figure
should imply some more consistent changes in the section text since, following the algorithm flowchart, the
detection capabilities are analyzed first and then the estimation capabilities; however, a reference to Figure 11
(now 12) will be added.
Lines 466-467 (lines 471-473 in the revised manuscript) have been modified
from:
*This is due to the combined effect of a stronger scattering signal associated with more intense snowfall events -*
*linked to moister and warmer environmental conditions -*

to:
*This is due to the combined effect of a stronger scattering signal associated with more intense snowfall events -*
*linked to moister and warmer environmental conditions, as can be observed in Figure 12 and Table 2 –*

**3.16) Lines 479-481: This statement is exactly what I am referring to in the previous comment. 60% POD**
**for very light snowfall rates is excellent and should be appropriately highlighted. But I do not see how this**
**value is derived for a 0.001 mm h-1 snowfall rate based on Figs. 7, 8, and 9.**

Thanks to the reviewer for the comment. It is important to underline that detection and retrieval modules are based
on different neural networks; so, the detection modules manage to identify "snowfall" conditions also in presence
of very light snowfall events. In the plots below the dependence of HANDEL-ATMS snowfall detection
capabilities in function of SWP/SSR values retrieved by CPR 2CSP product is reported - the statistics is calculated
for snowfall observations, therefore only POD can be calculated.

[Figure]

**Figure 9: Dependence of HANDEL-ATMS SWP and SSR POD on SWP/SSR values. Each star represents the**
**statistical score value for different SWP/SSR bins. The left y-axis reports POD values, while the right y-axis**
**reports the number of snowfall observations in the test dataset. Only POD has been reported because the**
**index has been calculated for observations where CPR 2CSP detects the presence of SWP/SSR.**

This plot has been added to the paper, and Lines 481-484 (lines 485-489 in the revised manuscript) have been
modified:
From:
*Moreover, also considering very low SWP and SSR values (SWP ≈ 0.001 kg m⁻², SSR ≈ 0.001 mm h⁻¹), HANDEL-*
*ATMS manages to detect around 60 % of the snowfall events. Similar considerations can be done also for the*
*different background surfaces.*

to:

*In Figure 9 the dependence of HANDEL-ATMS snowfall detection statistical scores on SWP and SSR values retrieved by CPR 2CSP is reported. Only POD is reported because the statistics are calculated for snowfall observations only (2CSP SWP/SSR > 0 kg m-2/mm h-1). It is possible to observe that also considering very low SWP and SSR values (SWP ≈ 0.001 kg m-2, SSR ≈ 0.001 mm h-1), HANDEL-ATMS manages to detect around 60 % of the snowfall events.*

**3.17) Lines 487-488: Similar to the previous comment, POD > 0.7 and FAR < 0.25 for the Perennial Snow and Winter Polar Snow surface categories. These values are very impressive. But instead of generally stating that these values are impressive due to both the complicated backgrounds with variable surface emissivity and "low snowfall intensity", I recommend providing some basic quantitative guidance to bolster this analysis. A suggestion: either state what the mean or median snowfall rate is for each of these categories or provide snowfall rate distributions for various surface categories.**

Thanks to the reviewer for the comment. A new table has been added to the paper in order to properly address and emphasize the important aspects raised by the reviewer.

| Class | TPW (mm) | $T_{2m}$ (K) | # obs | % SWP obs | % SSR obs | SWP (kg m$^{-2}$) | SSR (mm h$^{-1}$) |
|---|---|---|---|---|---|---|---|
| Ocean | 6.2 | 273 | $3.9*10^5$ | 79 | 64 | 0.046 | 0.071 |
| New Sea Ice | 3.2 | 255 | $2.1*10^5$ | 38 | 38 | 0.033 | 0.050 |
| Broken Sea Ice | 5.2 | 266 | $1.4*10^5$ | 57 | 57 | 0.044 | 0.073 |
| Multilayer Sea Ice | 4.5 | 260 | $9.9*10^4$ | 43 | 43 | 0.033 | 0.051 |
| Land | 5.3 | 270 | $2.8*10^4$ | 43 | 41 | 0.043 | 0.068 |
| Perennial Snow | 1.6 | 248 | $3.6*10^5$ | 31 | 31 | 0.022 | 0.035 |
| Winter Polar Snow | 2.1 | 245 | $6.0*10^4$ | 32 | 32 | 0.033 | 0.048 |
| Deep Dry Snow | 3.8 | 261 | $1.1*10^5$ | 50 | 50 | 0.040 | 0.066 |
| Thin Snow | 4.5 | 267 | $1.8*10^4$ | 54 | 53 | 0.041 | 0.070 |
| Coast | 4.0 | 259 | $3.1*10^5$ | 47 | 46 | 0.043 | 0.068 |

**Table 2: Environmental Characteristics for each PESCA class (test dataset): the number of occurrences, the mean TPW and $T_{2m}$ value, the percentage of SWP/SSR observations (over the total occurrences) and the mean SWP and SSR values are shown**

The following statement has been added (Line 342. Lines 336-344 in the revised manuscript)

*In Table 2 the number of PESCA class occurrences, the percentage of snowfall observations, and the most significant environmental characteristics in the ATMS-CPR coincident dataset are reported. It can be observed that Land and Ocean classes are characterized by the warmest/moistest conditions and by the most intense snowfall events (on average), while Perennial and Winter Polar Snow classes and New and Multilayer Sea Ice classes are characterized by the coldest/driest environmental conditions and by the lightest snowfall events (on average). Thin Snow and Broken Sea Ice classes show intermediate environmental conditions and snowfall intensity values. It is also interesting to highlight that a mismatch between the percentage of SWP and SSR observations is observed mostly over the Ocean class and, less frequently other classes (Land, Thin Snow, and Coast), where warmer and moister environmental conditions are found.*

Moreover, Lines 487-491 (lines 492-497 in the revised manuscript) have been modified from:

*It can be observed that, also considering specifically the classes associated to extremely dry and cold environmental conditions such as Perennial Snow or Winter Polar Snow (see Camplani et al, 2021), where the detection is more problematic due to the uncertainties in the emissivity retrieval (see Table 2) , and to the low snowfall intensity, HANDEL-ATMS has good detection capabilities (POD and FAR values greater than 0.7 and less than 0.25, respectively, for both SWP and SSR).*

to:

*It can be observed that, also considering specifically the classes associated with extremely dry and cold environmental conditions such as Perennial Snow or Winter Polar Snow (see Camplani et al, 2021 and Table 2), where the detection is more problematic due to low snowfall intensity (see Table 2) and to the uncertainties in the emissivity retrieval (see Table 3), HANDEL-ATMS has good detection capabilities (POD and FAR values greater than 0.7 and less than 0.25, respectively, for both SWP and SSR).*

**3.18) Section 4.3: This section somewhat addresses my previous suggestion of comparing other ATMS or passive microwave retrievals to the HANDEL-ATMS results. Do other passive microwave SSR retrievals exist - even historical studies - that advertise much different statistical scores than the current study? I am trying to gain further context and encourage the authors to find ways to highlight how revolutionary HANDEL-ATMS is for high latitude snowfall rate retrievals.**

Thanks to the reviewer for the positive comment. An ATMS snowfall retrieval algorithm based on the CPR 2CSP product is described by *You et al, 2022*. This algorithm has been developed for snowfall retrieval over ocean, sea ice, and coastal areas and it is based on logistic regression methods. A general comparison between the two algorithms is not possible because they work over different environmental conditions (dry and cold environmental conditions typical of high latitude areas for HANDEL-ATMS, specific background surfaces for the You et al algorithm). However, it is interesting to observe that both the algorithms show higher statistical scores over open water (ocean) with respect to sea ice or a coast. Moreover, the You et al algorithm shows better performances in presence of higher SWP/SSR values. Other ATMS snowfall retrieval algorithms, such as *Kongoli et al, 2015* and *Meng et al, 2017* have been trained over a specific geographic area (the CONUS U. S.) which is not representative of the extreme high latitude environmental conditions which HANDEL-ATMS development has focused on, therefore a comparison could be not very significant. Algorithms that rely on other MW radiometers carried out by non-polar orbiting satellites, such as GMI onboard GPM-CO, do not retrieve snowfall at high latitudes, and so a direct comparison can not be carried out.

REFERENCES

Kongoli, C., Meng, H., Dong, J., & Ferraro, R.: A snowfall detection algorithm over land utilizing high-frequency passive microwave measurements—Application to ATMS. *Journal of Geophysical Research: Atmospheres*, *120*(5), 1918-1932, https://doi.org/10.1002/2014JD022427, 2015.

Meng, H., J. Dong, R. Ferraro, B. Yan, L. Zhao, C. Kongoli, N.-Y. Wang, and B. Zavodsky, A 1DVAR-based snowfall rate retrieval algorithm for passive microwave radiometers, J. Geophys. Res. Atmos., 122, 6520–6540, doi:10.1002/2016JD026325, 2017.

You, Y., Meng, H., Dong, J., Fan, Y., Ferraro, R. R., Gu, G., & Wang, L.: A Snowfall Detection Algorithm for ATMS Over Ocean, Sea Ice, and Coast. *IEEE Journal of Selected Topics in Applied Earth Observations and Remote Sensing*, *15*, 1411-1420, DOI:10.1109/JSTARS.2022.3140768, 2022

---

## Author Response (AR3)

**Author's response**

**Public justification (visible to the public if the article is accepted and published)**:

**Both reviewers have indicated that the revisions have satisfied all of their concerns. Referee #4 suggests that the information in lines 607-619 of the response may be valuable context in the discussion. Therefore, if the authors wish to include this content, I have given the option to include it as a minor revision which will have an extremely expedited editor review.**

We would like to thank the editor. Please, find below our responses to the Reviewers' comments and the details on how we address them in the new version of the manuscript.

**Reviewer 4**

**The authors have satisfactorily addressed the mostly minor comments that I raised during my initial review of this manuscript. I would be completely fine with publishing the manuscript in its current state. The only \*very minor\* additional step to consider - is it worth including some of the content contained in lines 607-629 of the response document somewhere in the manuscript (perhaps section 5 as discussion or perspective content?). I really like this content since it adds valuable context. But I do not regard this suggestion as mandatory. I leave it to the authors' and editor's discretion whether it should be included in the final revised manuscript.**

We would like to thank Reviewer #4 for his/her review of our paper and the suggestion provided.

In order to answer the reviewer's comment, the following statements have been added (Line 576, 576-589 in the revised manuscript):

*Recently, several MW-based snowfall retrieval algorithms have been developed, but HANDEL-ATMS is the only one tailored for high-latitude regions. Algorithms developed for the GMI onboard GPM-CO, based on machine learning techniques and on the use of CPR 2CSP as reference (e.g., Rysman et al 2018, Rysman et al 2019), do not retrieve snowfall at high latitudes, and therefore a direct comparison with HANDEL-ATMS can not be carried out. Other snowfall retrieval algorithms based on ATMS observations (e.g., Kongoli et al, 2015, Meng et al, 2017) are trained over specific geographical areas (the Continental US region) and are not representative of the extreme, high-latitude environmental conditions, therefore a comparison with HANDEL-ATMS could be not very significant. In another study by You et al, 2022 a retrieval algorithm for ATMS, trained using the CPR 2CSP product and based on logistic regression methods, provides snowfall retrieval only over specific background surfaces - ocean, sea ice, and coastal areas. However, it is interesting to observe a qualitative consistency with HANDEL-ATMS. The two algorithms show higher statistical scores over open water (ocean) with respect to sea ice or coast and better detection capabilities in presence of higher SWP/SSR values. A quantitative comparison between SLALOM-CT and HANDEL-ATMS is presented below, since both algorithms are based on a machine-learning approach and are trained on a global ATMS-CPR coincidence dataset.*

REFERENCES

Kongoli, C., Meng, H., Dong, J., & Ferraro, R.: A snowfall detection algorithm over land utilizing high-frequency passive microwave measurements—Application to ATMS. *Journal of Geophysical Research: Atmospheres*, *120*(5), 1918-1932, https://doi.org/10.1002/2014JD022427, 2015.

Meng, H., J. Dong, R. Ferraro, B. Yan, L. Zhao, C. Kongoli, N.-Y. Wang, and B. Zavodsky, A 1DVAR-based snowfall rate retrieval algorithm for passive microwave radiometers, J. Geophys. Res. Atmos., 122, 6520–6540, doi:10.1002/2016JD026325, 2017.

Rysman, J. F., Panegrossi, G., Sanò, P., Marra, A. C., Dietrich, S., Milani, L., & Kulie, M. S.: SLALOM: An all-surface snow water path retrieval algorithm for the GPM Microwave Imager. *Remote Sensing*, *10*(8), 1278, https://doi.org/10.3390/rs10081278, 2018.

Rysman, J. F., Panegrossi, G., Sano, P., Marra, A. C., Dietrich, S., Milani, L., Kulie, M. S., Casella, D., Camplani, A., Claud, C., & Edel, L.: Retrieving surface snowfall with the GPM Microwave Imager: A new module for the SLALOM algorithm. *Geophysical Research Letters*, *46*(22), 13593-13601, https://doi.org/10.1029/2019GL084576, 2019.

You, Y., Meng, H., Dong, J., Fan, Y., Ferraro, R. R., Gu, G., & Wang, L.: A Snowfall Detection Algorithm for ATMS Over Ocean, Sea Ice, and Coast. *IEEE Journal of Selected Topics in Applied Earth Observations and Remote Sensing*, *15*, 1411-1420, DOI:10.1109/JSTARS.2022.3140768, 2022